# Filament organization of the bacterial actin MreB is dependent on the nucleotide state

Vani Pande[1], Nivedita Mitra[2,3,4], Saket Rahul Bagde[1], Ramanujam Srinivasan[2,3,4], and Pananghat Gayathri[1]

**MreB, the bacterial ancestor of eukaryotic actin, is responsible for shape in most rod-shaped bacteria. Despite belonging to the actin family, the relevance of nucleotide-driven polymerization dynamics for MreB function is unclear. Here, we provide insights into the effect of nucleotide state on membrane binding of *Spiroplasma citri* MreB5 (ScMreB5). Filaments of ScMreB5[WT] and an ATPase-deficient mutant, ScMreB5[E134A], assemble independently of the nucleotide state. However, capture of the filament dynamics revealed that efficient filament formation and organization through lateral interactions are affected in ScMreB5[E134A]. Hence, the catalytic glutamate functions as a switch, (a) by sensing the ATP-bound state for filament assembly and (b) by assisting hydrolysis, thereby potentially triggering disassembly, as observed in other actins. Glu134 mutation and the bound nucleotide exhibit an allosteric effect on membrane binding, as observed from the differential liposome binding. We suggest that the conserved ATP-dependent polymerization and disassembly upon ATP hydrolysis among actins has been repurposed in MreBs for modulating filament organization on the membrane.**

## Introduction

The chromosomally encoded bacterial actin, MreB, plays a pivotal role in cell shape determination in bacteria (Shi et al., 2018; Errington, 2015). Genetic studies have shown that the deletion of *mreB* genes leads to loss of rod shape and eventual lysis (Kawai et al., 2009; Kruse et al., 2005; Bendezú and De Boer, 2008). MreB functions as a scaffold for the assembly of cell wall synthesis machinery and, thus, locally leads to cell wall insertion favoring a rod shape (van Teeffelen et al., 2011; Domínguez-Escobar et al., 2011; Garner et al., 2011). During growth, the short filaments of MreB align approximately perpendicular to the long axis of the rod-shaped cells (Garner et al., 2011; van Teeffelen et al., 2011). Their circumferential movement recruits the peptidoglycan synthesis machinery at uniformly distributed locations along the long axis, thus reinforcing the rod shape.

A characteristic feature of MreB filaments is their ability to bind to the lipid bilayer or monolayer in vitro (van den Ent et al., 2014; Salje et al., 2011; Maeda et al., 2011). Hence, MreB filaments are capable of sensing as well as generating membrane curvature in liposomes, independent of the peptidoglycan synthesis machinery (Shi et al., 2018; Salje et al., 2011; Garenne et al., 2020; Ursell et al., 2014). Experimental studies on *Escherichia coli* (EcMreB), *Caulobacter crescentus* (CcMreB), and *Thermatoga maritima* (TmMreB) MreBs have shown that MreB interacts with the cell membrane either via N-terminal amphipathic helix

in Gram-negative bacteria and/or a hydrophobic loop in subdomain 1A in Gram-positive bacteria (Salje et al., 2011).

MreB filaments possess an antiparallel double-protofilament assembly (van den Ent et al., 2014), as opposed to the parallel protofilament arrangement in most actin family members such as eukaryotic actin (Chou and Pollard, 2019; Fujii et al., 2010) and ParM (Gayathri et al., 2013), an actin-like protein in plasmid segregation. Biochemical studies of TmMreB have shown that it is an active ATPase (Bean and Amann, 2008). Light-scattering studies for MreBs have shown that it polymerizes in the presence of ATP, AMP-PNP (adenylyl-imidodiphosphate, nonhydrolyzable analog of ATP), GTP, or ADP (Bean and Amann, 2008; Nurse and Marians, 2013; Mayer and Amann, 2009; Gaballah et al., 2011). Additionally, *Bacillus subtilis* MreBs were shown to undergo nucleotide-independent polymerization (Mayer and Amann, 2009), and double-protofilament assembly was observed in the crystal structure of CcMreB without nucleotide (van den Ent et al., 2014). Therefore, the significance of nucleotide binding or hydrolysis for filament formation and dynamics in MreB function is ambiguous.

Although in vivo effects of ATP hydrolysis mutants of MreB have indicated a potential role for hydrolysis in MreB function (Kurita et al., 2019; Dye et al., 2011; Bratton et al., 2018), the exact role of hydrolysis-dependent filament dynamics is unknown. Mutational defects in the ATP-binding pocket of MreB altered

[1]Indian Institute of Science Education and Research, Pune, India; [2]School of Biological Sciences, National Institute of Science Education and Research, Bhubaneswar, India; [3]Homi Bhabha National Institutes, Training School Complex, Anushakti Nagar, Mumbai, India; [4]Centre for Interdisciplinary Sciences, National Institute of Science Education and Research, Bhubaneswar, India.

Correspondence to Pananghat Gayathri: gayathri@iiserpune.ac.in; Ramanujam Srinivasan: rsrini@niser.ac.in.

the localization of MreB filaments, cell morphology, and chromosome segregation in *B. subtilis* and *C. crescentus* (Gitai et al., 2005; Defeu Soufo and Graumann, 2006). Further, the spatial regulation of MreB filaments in response to cellular curvature has been hypothesized to depend on hydrolytic activity (Ursell et al., 2014; Dye et al., 2011). The observation of filament dynamics through an in vitro reconstitution approach has not been reported for MreB, probably owing to the challenges with imaging the short filaments using light microscopy.

Until recently, studies on MreB function, including cell shape regulation and maintenance, cell division, and motility, were reported only from cell-walled bacteria (Kawai et al., 2009; Dye et al., 2011; Defeu Soufo and Graumann, 2006; Mauriello et al., 2010), wherein it functions in conjunction with the peptidoglycan synthesis machinery. Therefore, how rod shape is mediated by MreB in bacteria without a cell wall remains enigmatic. Interestingly, many of the features of MreBs from cell-walled bacteria, such as the antiparallel double-protofilament assembly and membrane binding, are common to MreB from a wall-less bacterium too, as demonstrated by our study on MreB5, one of the five paralogs of MreB in wall-less helical bacterium *Spiroplasma citri* (Harne et al., 2020). The role of multiple (five to seven) paralogs of MreBs in these organisms remains poorly understood. Our work showed that among the five paralogs of MreB, MreB5 (ScMreB5) is essential for the helical shape and motility of *S. citri* (Harne et al., 2020).

Orientation of MreB filaments within cells has been proposed to be dependent on the differences between the principle curvatures, with a more ordered arrangement when the difference is higher, as in a narrow rod (Hussain et al., 2018). Mutations in MreB can result in cells of varying width (Ouzounov et al., 2016; Shi et al., 2017). The narrow diameter of *Spiroplasma* cells (∼100–150 nm; Shaevitz et al., 2005) compared with most other bacterial species makes the investigation of *Spiroplasma* MreB paralogs especially interesting. These aspects prompted us to carry out an in-depth study of ScMreB5. Characterization of an MreB that functions independently of cell wall synthesis machinery in a bacterium of unusually thin diameter will help in identifying the fundamental mechanism and conserved signatures for MreB function.

Here, we report the structural and biochemical characterization of ScMreB5^WT (WT). An ATPase-deficient mutant of ScMreB5, ScMreB5^E134A, shows defects in polymerization compared with ScMreB5^WT. We propose an additional novel role for the catalytic residue Glu134, which earlier has been implicated mostly in stimulating hydrolysis in most actin family members such as actin (Vorobiev et al., 2003), ParM (Gayathri et al., 2013), and MamK (Löwe et al., 2016). Our evidence suggests that Glu134 may assist in conformational changes during polymerization. Furthermore, through lipid specificity studies and mutational analysis, we show that the electrostatic interactions through positively charged residues contribute to membrane binding. The observations from liposome-binding studies of ScMreB5^E134A and nucleotide dependence of liposome binding provide novel insights into the role of ATP hydrolysis and its effect on conformational dynamics and membrane-binding properties essential for MreB function. The results also highlight the

conserved features of allostery and filament dynamics observed in both actin (Chu and Voth, 2005) and MreB, despite the differences in their protofilament organization.

## Results

### Crystal structure of ADP-bound ScMreB5 shows a conserved single-protofilament organization

We recently reported the crystal structure of ScMreB5 bound to AMP-PNP (Protein Data Bank [PDB] accession no. 7BVY; Harne et al., 2020). ScMreB5 also crystallized in the presence of ADP (PDB accession no. 7BVZ). The overall structure of ADP-bound ScMreB5 (ScMreB5–ADP) was very similar to AMP-PNP–bound ScMreB5 (ScMreB5–AMP-PNP), superimposing with an overall root mean square deviation (RMSD) of 0.87 Å (Fig. 1, A and B). Clear electron density was observed for the nucleotides in both ADP- and AMP-PNP–bound states (Fig. 1, C and D; data collection statistics in Table S1). The packing of ScMreB5–ADP molecules in the crystal structures revealed a single-protofilament assembly, similar to that of ScMreB5–AMP-PNP (Fig. 1 E; (Harne et al., 2020)). The longitudinal repeat distances of 51.1 Å for both these protofilaments were remarkably similar when compared with TmMreB and CcMreB subunit repeat distances of 51.1 Å (van den Ent et al., 2001), despite all four belonging to different space groups and packing environments. The hydrophobic loop in the IA domain, predicted to be the membrane-binding loop in TmMreB (Salje et al., 2011), was disordered (residues 93–97) in ScMreB5–ADP (Fig. 1 A). This loop was ordered in the ScMreB5–AMP-PNP structure (Fig. 1 B), probably owing to crystal packing differences.

An interesting observation in both the structures of ScMreB5 (ADP and AMP-PNP complexes) was the presence of strong electron density for a potassium ion (Fig. 1, C and D; and Fig. S1 A), positioned between α and β phosphates of ADP and AMP-PNP (Fig. 1, A–D; and Fig. S1 B). Thermal shift assays showed increased stability in the KCl buffer and upon addition of excess ADP or ATP (Fig. S1, C and D). We also observed that the purified ScMreB5 contains bound ADP (Fig. S1 E), providing a structural basis for increased stability of ScMreB5 in KCl buffer. The crystal structure of CcMreB has a water molecule at the position equivalent to the potassium ion binding site in ScMreB5 (Fig. S1 F). The residues Asp12 and Asn17, which are involved in potassium ion coordination in ScMreB5, are well conserved among MreBs from different bacteria (Fig. S2) and bind to the corresponding water molecule in CcMreB.

### ScMreB5 is an active ATPase

Residues in the nucleotide binding cleft of ScMreB5 are well conserved with respect to other MreBs and yeast actin (Figs. 1 F and S2). In ScMreB5, the water molecules that form the coordination sphere for Mg²⁺ are held together by the side chain carboxyl oxygens (Oδ1/Oδ2 or Oε1/Oε2) of Asp12, Asp156, and Glu134 (Fig. 1 G). Electron density for the catalytic water, typically situated at an in-line geometry with the γ-phosphate moiety, was absent in the structure of ScMreB5–AMP-PNP and hence not modeled. However, Glu134 and/or Thr161 might interact with the catalytic water, a hypothesis based on structure

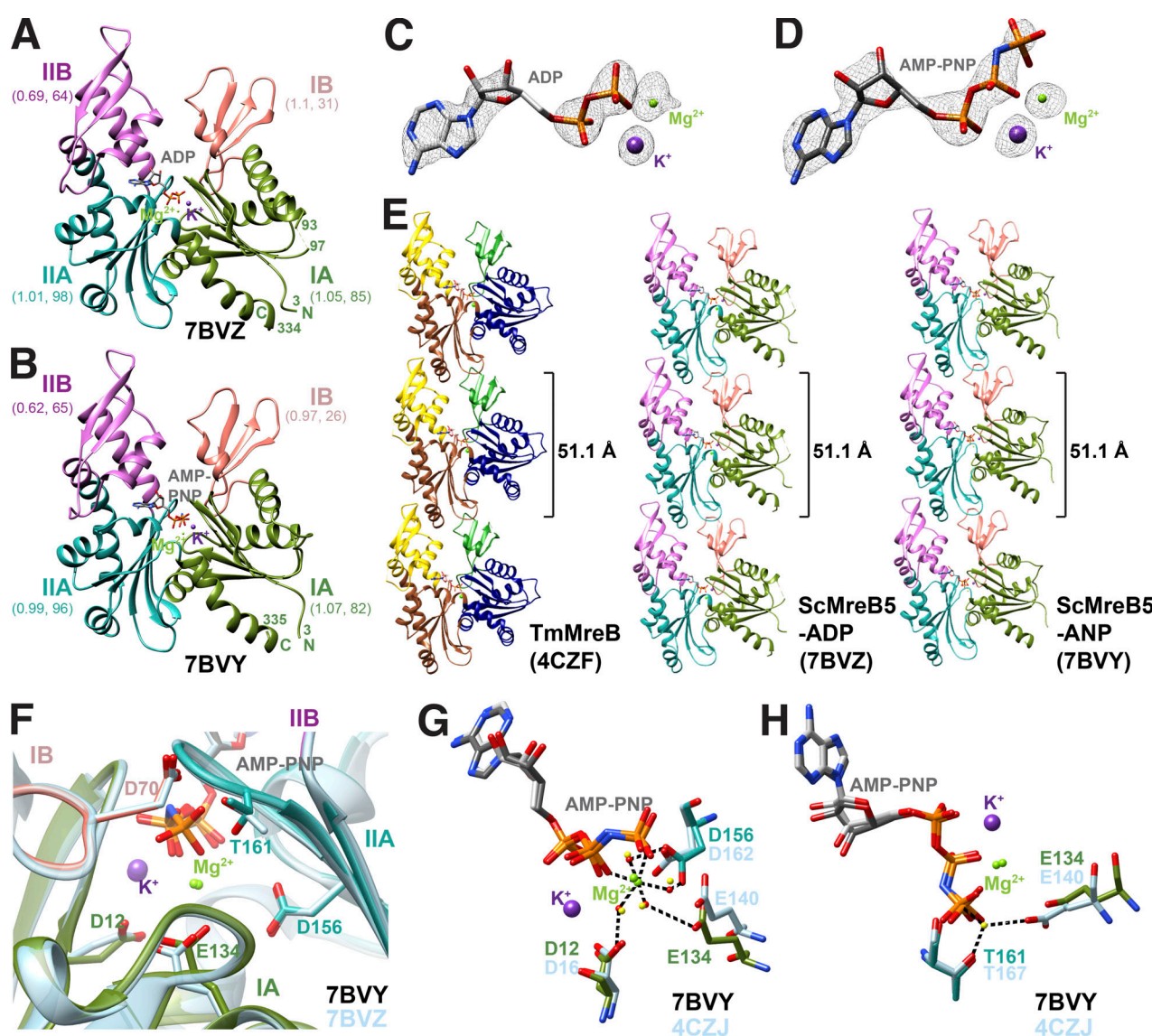

Figure 1. **ScMreB5 possesses a conserved protofilament arrangement and nucleotide-binding pocket. (A and B)** Crystal structures of ScMreB5 in ADP (PDB accession no. 7BVZ) and AMP-PNP (PDB accession no. 7BVY) bound states. The subdomains IA, IB, IIA, and IIB are colored and labeled. RMSD values of each subdomain upon superposition with the corresponding subdomains in CcMreB along with the number of Cα atoms superposed are given below the subdomain labels. N- and C-terminal ends are labeled N and C, respectively, and the terminal residue numbers are marked. The chain breaks in 7BVZ are also labeled by their residue numbers (93 and 97). **(C and D)** Electron density for the bound ADP and AMP-PNP with $Mg^{2+}$ and $K^+$ (composite omit map $F_o − F_c$ shown at 2.0 σ). **(E)** Protofilament structures of CcMreB (PDB accession no. 4CZF), ScMreB5 with bound ADP (PDB accession no. 7BVZ) and AMP-PNP (labeled as ANP in figure; PDB accession no. 7BVY). Both of the nucleotide-bound structures of ScMreB5 have the same subunit repeat as CcMreB (51.1 Å) in their protofilament assemblies. Individual chains are colored according to the subdomains. **(F)** Zoomed-in view of the residues at the nucleotide binding pocket. Residues of ScMreB5–AMP-PNP (PDB accession no. 7BVY; domain-wise colors) are shown superimposed with corresponding residues in ScMreB5–ADP (PDB accession no. 7BVZ; blue-gray). **(G)** Residues involved in $Mg^{2+}$ coordination in ScMreB5 (Asp156, Glu134, and Asp12). Distances for $Mg^{2+}$ coordination are marked by dotted lines for ScMreB5–AMP-PNP. **(H)** Residues adjacent to the γ-phosphate, Glu134 and Thr161, at the nucleotide-binding pocket. Distances with the catalytic water are marked by dotted lines for CcMreB structure.

superpositions with other MreBs and actin structures (Fig. 1, G and H; and Fig. 2, A and B; van den Ent et al., 2014; Vorobiev et al., 2003; Merino et al., 2018).

We carried out ATPase activity measurements of ScMreB5[WT] and active site residue mutants, ScMreB5[D12A], ScMreB5[D156A], ScMreB5[E134A], and ScMreB5[T161A], by measuring the released phosphate using a colorimetric assay. We also chose to mutate Asp70 (ScMreB5[D70A]; Fig. 1 F), which is well conserved in MreBs, whereas it is replaced by His73 in actin (Fig. 2 C). Studies on

actin have shown that His73 is important for polymerization and for regulating phosphate release upon ATP hydrolysis (Nyman et al., 2002). The ScMreB5 mutants were purified to homogeneity (Fig. S3 A) and were well folded (inferred based on elution in the monomeric fraction in size-exclusion chromatography; Fig. S3, B and C), and showed a decrease in ATPase activity, compared with WT (Fig. 2 D). $k_{obs}$ values for ScMreB5[WT] (0.15 ± 0.007 $min^{-1}$) and the mutants (ScMreB5[D12A], 0.02 ± 0.008; ScMreB5[D156A], 0.08 ± 0.014, ScMreB5[E134A], 0.01 ± 0.004;

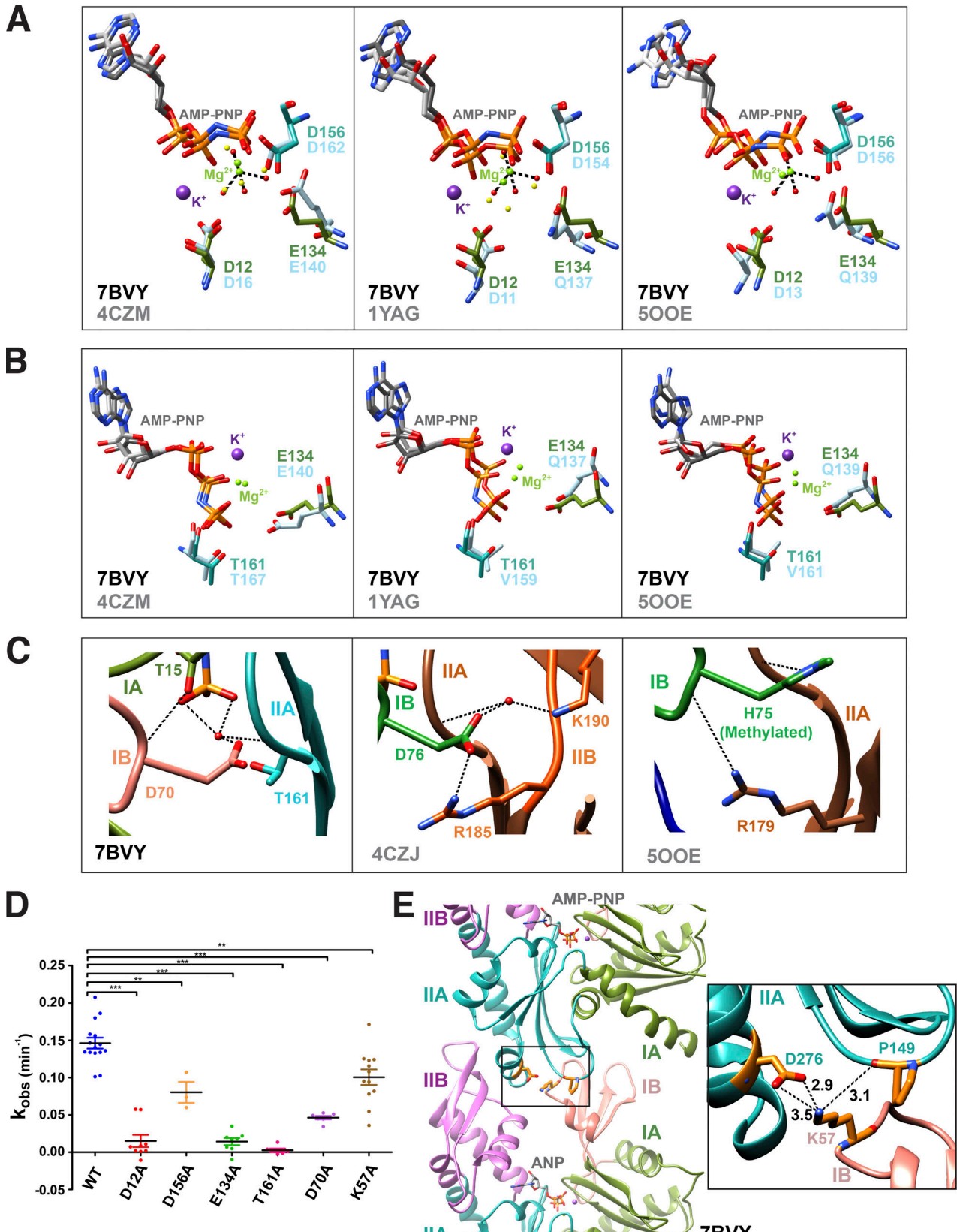

**Figure 2. ScMreB5 is an active ATPase.** Residues of ScMreB5 at the active site are compared with the monomeric CcMreB (PDB accession no. 4CZM), double-protofilament CcMreB (PDB accession no. 4CZJ), monomeric yeast actin (PDB accession no. 1YAG), and actin filament (PDB accession no. 5OOE) by superposing IIA domain of each structure onto IIA domain of ScMreB5–AMP-PNP structure (single protofilament conformation). The color of the residues are subdomain wise, for ScMreB5–AMP-PNP, IB (pink), IA (green), and IIA (sea green). For the other MreBs and actin structures, residues in subdomains are colored

light blue. All distances marked by black dotted lines are <3.5 Å. **(A)** Superimposed active site residues holding the $Mg^{2+}$ coordination sphere. Asp12, Glu134, and Asp156 residues of ScMreB5 are compared with the corresponding residues in monomeric CcMreB, monomeric actin, and actin filament. **(B)** Superimposed active site residues at the catalytic water interface. Glu134 and Thr161 residues of ScMreB5 are compared with the corresponding residues in monomeric CcMreB, monomeric actin, and actin filament. **(C)** Interacting interface for Asp70 of ScMreB5–AMP-PNP is shown with respect to corresponding residues present in double-protofilament CcMreB (PDB accession no. 4CZJ) and actin filament (PDB accession no. 5OOE). **(D)** ATPase activity characterization of ScMreB5. $k_{obs}$ (min$^{-1}$) for the ScMreB5$^{WT}$, active site mutants, and the polymerization mutant (ScMreB5$^{WT}$ [WT; $N$ = 3; $n$ = 15], ScMreB5$^{E134A}$ [E134A; $N$ = 3; $n$ = 8], ScMreB5$^{D12A}$ [D12A; $N$ = 2; $n$ = 7], ScMreB5$^{D156A}$ [D156A; $N$ = 2; $n$ = 3], ScMreB5$^{D70A}$ [D70A; $N$ = 2; $n$ = 7], ScMreB5$^{T161A}$ [T161A; $N$ = 1; $n$ = 8], and ScMreB5$^{K57A}$ [K57A; $N$ = 2; $n$ = 12]; $N$, number of independent protein purification batches; $n$, total number of repeats). The error bar denotes mean with SEM; unpaired $t$ test, two-tailed; ***, P < 0.0001; **, P = 0.001–0.002. 10 µM protein, 1 mM ATP, and 1 mM $MgCl_2$ were used in this assay. **(E)** Intra-protofilament polymerization interface of ScMreB5 (PDB accession no. 7BVY). Inset: Zoomed-in view of the interface showing the residue Lys57 of subdomain IB interacting with Asp276 and Pro149 of subdomain IIA. The distances in Å are labeled for the interactions.

---

ScMreB5$^{T161A}$, 0.0005 ± 0.001; and ScMreB5$^{D70A}$, 0.05 ± 0.002) are tabulated in Table S2. ScMreB5$^{WT}$ exhibited similar activity ($k_{obs}$ 0.15 ± 0.007 min$^{-1}$) as shown earlier for EcMreB ($k_{obs}$ 0.17 ± 0.01 min$^{-1}$; Nurse and Marians, 2013) and TmMreB ($k_{obs}$ 0.10 ± 0.01 min$^{-1}$; Esue et al., 2005).

Crystal structures of the various states of CcMreB showed that conformational changes upon polymerization affect the positioning of the catalytic residues at the active site (van den Ent et al., 2014). Hence, in addition to mutants of the ATP-binding pocket residues, we checked the ATPase activity of a polymerization interface mutant, ScMreB5$^{K57A}$. Lys57 is present at the intra-protofilament interface of ScMreB5 protofilament assembly (Fig. 2 E). This residue is conserved across different MreBs (Fig. S2). ATPase activity of ScMreB5$^{K57A}$ was lower compared with ScMreB5$^{WT}$ (Fig. 2 D and Table S2), indicating allosteric communication between the polymerization interface and the active site. Light scattering measurements, although performed at a lower concentration of protein compared with WT, show that ScMreB5$^{K57A}$ might indeed polymerize (Fig. S3, D and E). The mutation of a single residue at the interface might not abrogate polymerization completely but might result in a suboptimal interface. This could lead to a decrease in polymerized content or a suboptimal conformation of the ATPase active site within the polymers, thereby leading to a decrease in ATPase activity.

### ScMreB5$^{E134A}$ filaments exhibit impaired dynamics

We attempted to study the nucleotide dependence of filament formation for ScMreB5$^{WT}$ and its ATP hydrolysis mutant ScMreB5$^{E134A}$ by observing the presence of filaments in vitro using cryo-EM. ScMreB5$^{WT}$ in the presence of ATP (Fig. 3 A) and AMP-PNP (Fig. 3 B) formed a high density of double-protofilament assemblies having a sheet-like appearance of laterally associated filament bundles. We also observed filaments of ScMreB5$^{WT}$ in the presence of ADP (Fig. 3 C) and ScMreB5$^{E134A}$ in the presence of AMP-PNP (Fig. 3 D). However, very few sheet-like bundles were observed in ScMreB5$^{WT}$–ADP and ScMreB5$^{E134A}$–AMP-PNP compared with ScMreB5$^{WT}$–ATP and –AMP-PNP (Fig. 3, A–D). While filaments observed for ScMreB5$^{WT}$–ADP and ScMreB5$^{E134A}$–AMP-PNP demonstrated that ATP hydrolysis was not required for filament formation in vitro, the observation of very few sheet-like bundles suggested lower filament density or defective bundling or both.

With an aim to visualize the polymerization dynamics of ScMreB5 filaments, we expressed N-terminal GFP-fusion

constructs of ScMreB5$^{WT}$ and ScMreB5$^{E134A}$, respectively, in fission yeast and monitored their filament assembly. Dynamics of EcMreB polymerization in fission yeast with a similar N-terminal GFP fusion has been reported (Srinivasan et al., 2007). Although the N-terminal GFP-fusion did not adopt an orientational preference perpendicular to the long axis, as observed for *E. coli* or *Bacillus* MreBs in vivo (Domínguez-Escobar et al., 2011; Garner et al., 2011), it was functional for ATP hydrolysis (Table S2). Thus, it served as a useful system to observe the effect of the mutation on filament dynamics and bundling.

ScMreB5$^{WT}$ showed filaments extending across the cells that would eventually bundle up and orient along the long axis of the cells (Fig. 4, A and B; and Video 1). However, unlike ScMreB5$^{WT}$ filaments, the spatial organization of ScMreB5$^{E134A}$ filaments appeared to be different in yeast cells (Fig. 4 A). Differences in organization of filaments were more clearly visible by super-resolution imaging (3D structured illumination microscopy [3D-SIM]) of ScMreB5 filaments (Fig. 4 B and Video 1). Quantification of the spatial organization, by measuring anisotropy using FibrilTool (Boudaoud et al., 2014), and coefficient of variation (CV; Higaki et al., 2020), which is an indicator of cytoskeleton bundling, further confirmed that ScMreB5$^{E134A}$ exhibited differences in bundling of filaments (Fig. 4, C and D). However, the density of filaments (MreB polymer content per unit area of the cell) was not significantly different (Fig. 4 E). Moreover, a count of the number of cells with polymers showed that filaments were observed in very few cells expressing ScMreB5$^{E134A}$ in comparison with ScMreB5$^{WT}$ (Fig. 4 F). Quantification of the fluorescence intensity in cells with diffused fluorescence indicated that the average fluorescence intensity for ScMreB5$^{WT}$ was slightly lower than that of ScMreB5$^{E134A}$, suggesting a lower critical concentration for WT (Fig. 4, F and G).

Time-lapse imaging of polymerization of ScMreB5$^{WT}$ and ScMreB5$^{E134A}$ in yeast cells confirmed that polymerization and lateral association of filaments were efficient in ScMreB5$^{WT}$ compared with ScMreB5$^{E134A}$ (Fig. 5 A and Video 2). To visualize initiation of polymerization, cells were grown in the absence of the repressor (thiamine) for ≤28–30 h and then placed on agarose pads lacking thiamine, and random fields with cells exhibiting diffuse fluorescence were imaged. The time at which the cells were placed on agarose pads and first imaged was taken as $t$ = 0. Polymerization happens spontaneously, presumably within the cells that have sufficient monomers beyond the critical concentration for polymerization. An estimation of the time taken to initiate polymerization (from $t$ = 0 s) showed that

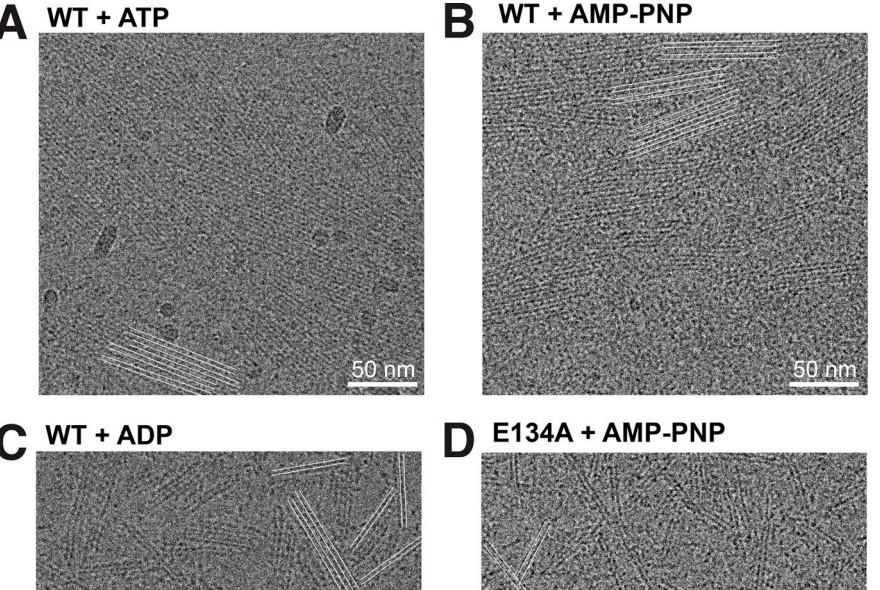

**A** WT + ATP

**B** WT + AMP-PNP

**C** WT + ADP

**D** E134A + AMP-PNP

50 nm

Figure 3. **ScMreB5 filaments form double-protofilament assemblies independent of nucleotide hydrolysis. (A–D)** Cryo-electron micrographs showing filaments of ScMreB5[WT] in the presence of 5 mM ATP and MgCl$_2$ (A); ScMreB5[WT] in the presence of 5 mM AMP-PNP and MgCl$_2$ (B); ScMreB5[WT] in the presence of 5 mM ADP and MgCl$_2$ (C); and ScMreB5[E134A] mutant (hydrolysis deficient) in the presence of 5 mM AMP-PNP and MgCl$_2$ (D). A few double protofilaments are highlighted by pairs of parallel white lines to enable easy visualization of the filament distribution. Concentration of protein used was 50 µM. Scale bar denotes 50 nm.

while cells expressing ScMreB5[WT] started to form polymers in 26.6 min (number of cells = 7, 95% confidence interval [CI] = 15.0), ScMreB5[E134A] started to assemble filaments much later, at 69.0 min (number of cells = 9, 95% CI = 11.5; Fig. 5, A and B). A time-course experiment and quantification of the percentage of cells exhibiting polymers in ScMreB5[WT] and ScMreB5[E134A] further confirmed the time lag in polymerization of ScMreB5[E134A] (Fig. 5 C). This was not due to differences in the expression levels of the mutant. Western blotting with anti-GFP antibodies and quantification of protein levels with tubulin as internal control show that both ScMreB5[WT] and ScMreB5[E134A] were expressed at similar levels (Fig. 5 D). Taken together, these results and the observation that filaments of ScMreB5[E134A] were seen in fewer cells compared with ScMreB5[WT] (Fig. 4 E) suggest a requirement for a higher concentration of monomers for polymerization of ScMreB5[E134A] (Fig. 4, F and G).

Lateral association of filaments in ScMreB5[WT] was often promoted by cell septation as the ingressing septa brought the filaments in close proximity (25 out of 43 cells). The difference in the spatial organization of ScMreB5[E134A] filaments was clearly seen in yeast cells undergoing cell division (Fig. 5 E and Video 3). Interestingly, in a few instances (4 of 43), ScMreB5[WT] filaments disassembled in the daughter cells immediately after a cytokinesis event (Fig. 5 F and Video 4). In rare events (2 out of 43 cells), filaments of ScMreB5[WT] appeared to undergo fragmentation and reannealing (Fig. 5 G and Video 5). However, we did not observe any such events of disassembly (out of 41 cells observed) of the ScMreB5[E134A] filaments, and they appeared stable. Although filament stabilization is a characteristic feature of ATPase-defective mutants of the actin family, we cannot

completely rule out that disassembly events were not observed in ScMreB5[E134A] due to experimental artifacts.

## Surface charge and active site mutation influences liposome binding of ScMreB5

Recently, we showed that ScMreB5 interacts with liposomes (Harne et al., 2020). We further explored the sequence determinants and lipid specificities that influence membrane binding of ScMreB5 in this study. Although ScMreB5 lacks a distinct amphipathic helix at its N- or C-terminal ends (Fig. 6, A and B), it possesses Ile95 and Trp96 in the hydrophobic loop of the IA domain (Fig. 6 C), which might act as membrane anchors (Fig. 6, C and D; de Jesus and Allen, 2013; Salje et al., 2011). Hence, we made single (ScMreB5[I95A] and ScMreB5[W96A]) and double (ScMreB5[I95A,W96A]) mutant constructs of these residues of ScMreB5[WT] (Fig. S3) and tested their binding using liposomes having lipid composition resembling the *S. citri* membrane (Davis et al., 1985). Before the addition of liposomes, the protein samples were spun at 21,500 *g* to ensure that any protein aggregates were removed. Pelleting assays of the reaction mix without liposomes served as negative controls for the liposome-binding experiments (Fig. S4 A). The control runs showed that the protein does not pellet on its own in the absence of liposome, irrespective of its polymerization state. In comparison to ScMreB5[WT], the mutations did not abrogate liposome binding significantly. Both the single and double mutant proteins were found in the pellet fraction (Fig. 6, E and F). This suggested that the hydrophobic loop might not serve as the sole membrane anchor for ScMreB5, contrary to what was observed for TmMreB (Salje et al., 2011).

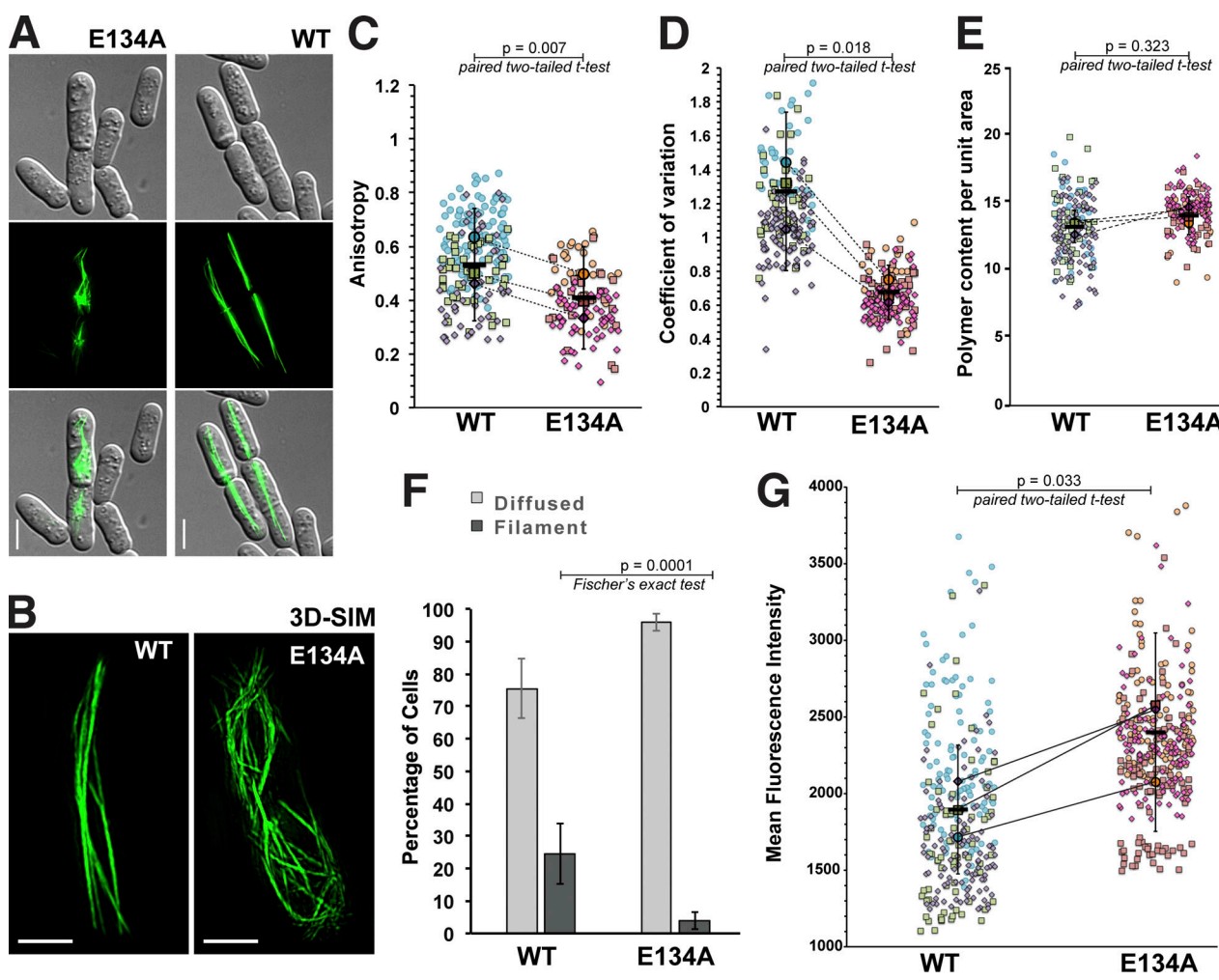

Figure 4. **Expression of GFP-tagged ScMreB5 in *S. pombe* cells reveals differences in filament assembly between the ATP hydrolysis mutant ScMreB5^E134A and ScMreB5^WT. (A)** ScMreB5^WT and ScMreB5^E134A with N-terminal tagged GFP expressed in *S. pombe* cells. Representative images are shown for the DIC images of the cell boundaries (top column), a GFP channel (middle column), and overlay (bottom column). **(B)** 3D-SIM images of ScMreB5^WT and the ScMreB5^E134A mutant are shown (related to Video 1). Filaments of ScMreB5^WT appear as bundles spanning from one end to the other of the cell, whereas ScMreB5^E134A filaments are not tightly bundled. **(C)** Plot showing the anisotropy, an indicator of parallelness of filament arrays, of ScMreB5^WT and the ScMreB5^E134A mutant. Plots are shown as superplots from three biological replicates (N = 3) with number of cells measured in each replicate ≥23 and ≤112. **(D and E)** Plots showing CV (D), a metric indicating the extent of filament bundling, and density (E), which is the amount of ScMreB polymers per unit area of cells, of ScMreB5^WT and the ScMreB5^E134A mutant. Plots are shown as superplots from three biological replicates (N = 3) with number of cells measured in each replicate being ≥23 and ≤93. **(F)** Plot comparing the percentage of cells showing either diffuse fluorescence or filaments for ScMreB5^WT and ScMreB5^E134A. Cells were grown in EMM without thiamine for 36–40 h. Cells expressing ScMreB5^E134A show more diffused fluorescence and very few filaments in comparison to ScMreB5^WT. The mean values (N = 3) are plotted, and the error bar denotes mean with SEM. Fisher's exact test, two-tailed. **(G)** Plot showing the mean fluorescence intensity of ScMreB5^WT and the ScMreB5^E134A mutant cells having diffuse fluorescence. The average diffuse fluorescence intensities in ScMreB5^E134A mutant cells are higher than in ScMreB5^WT, suggesting a higher critical concentration of polymerization for the ATPase mutant ScMreB5^E134A. Mean fluorescence intensities were calculated from sum intensity projections of cells with diffuse fluorescence. Plots are shown as superplots from three biological replicates (N = 3) with number of cells measured in each replicate being ≥23 and ≤112. For each replicate in C–E and G, ScMreB5^WT and the ScMreB5^E134A were grown at the same time to account for day-to-day variations in the growth of cultures at the single-cell level. Statistical significance was assessed by paired two-tailed Student's *t* test. P values were calculated using Microsoft Excel formula TTEST. The error bars shown are inferential and represent 95% CI. The mean values of each replicate in the superplot for ScMreB5^WT and ScMreB5^E134A are connected by dotted lines.

To decipher mechanistic details of ScMreB5–liposome interaction, we carried out a phospholipid specificity study. *S. citri* membrane consists of ∼38% phosphatidylglycerol (an anionic lipid) and ∼14% phosphatidylcholine (a neutral lipid; Davis et al., 1985). Hence, we tested whether liposome binding by ScMreB5 could be charge specific. ScMreB5^WT and the hydrophobic loop mutants did not bind to 100% 1,2-dioleoyl-sn-glycero-3-phosphocholine (DOPC) liposomes (Fig. 7, A and B), whereas binding

was observed with 100% 1,2-dioleoyl-sn-glycero-3-phospho-(1'-rac-glycerol) (DOPG, an anionic lipid) liposomes (Fig 7, A and B). Interestingly, MreB5s in spiroplasmas have a longer C-terminal end, which contains a stretch of lysines and arginines (Fig. 7 C). Based on the structures of MreBs, we know that both the N- and C-termini of the protein and the hydrophobic loop face the same side of the monomer and filament surface (Fig. 6 D), although the C-terminus is unstructured in the ScMreB5 crystal

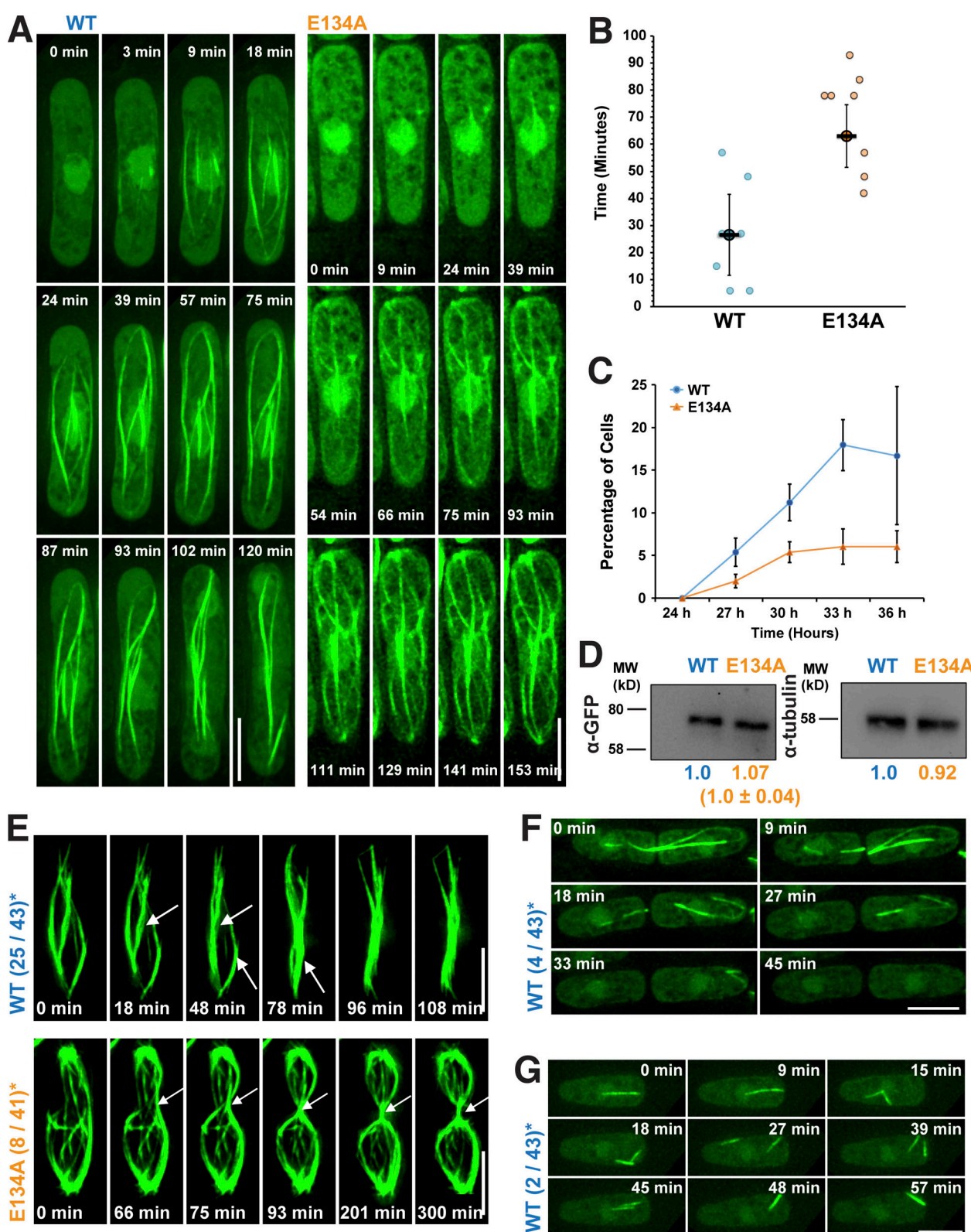

* Number of cells where events were observed / Total number of cells

Figure 5.   **Time-lapse imaging of GFP-tagged ScMreB5 in *S. pombe* cells for estimating the differences in filament assembly dynamics of the ATP hydrolysis mutant ScMreB5E134A. (A)** Time-lapse microscopy showing polymerization for ScMreB5WT and ScMreB5E134A (related to Video 2). Cells were grown for 28–30 h in the absence of thiamine, placed on an agarose pad without thiamine as mentioned in Materials and methods, and imaged at every 3-min time interval. The time at which the cells were placed on agarose pads and first imaged was taken as *t* = 0. **(B)** Plot showing the difference in time lag in the polymerization of ScMreB5E134A (number of cells = 7) compared with ScMreB5WT (number of cells = 9). In cells where polymerization was observed, it was seen

to initiate between 6 to 57 min, with a mean time of ∼26 min for ScMreB$^{WT}$ as mentioned in the text. For ScMreB$^{E134A}$ mutant, the time was significantly delayed, and polymerization was observed between 42 to 93 min, with a mean of 69 min. Error bars represent 95% CI; statistical significance was assessed using an unpaired two-tailed Student's $t$ test, and the P value (0.00044) was calculated using the formula TTEST in Microsoft Excel. **(C)** A time-course experiment showing the lag in polymerization of ScMreB5$^{E134A}$ compared with ScMreB5$^{WT}$. The percentage of cells (number of cells counted for each biological replicate was ≥1,239 and ≤1,792) having filaments were calculated for ScMreB5$^{WT}$ and ScMreB5$^{E134A}$ and plotted as a function of time. The mean from three biological replicates ($N$ = 3) is plotted, and the error bars are inferential and represent 95% CI. **(D)** A representative immunoblot (one of three repeats) showing similar levels of protein expression for ScMreB5$^{WT}$ and ScMreB5$^{E134A}$. ScMreB5$^{WT}$ and ScMreB5$^{E134A}$ were detected using anti-GFP antibodies conjugated to HRP, and β-tubulin was used a loading control. Band intensities were measured using Fiji, and the numbers below the GFP blot indicate values normalized with respect to the tubulin band. **(E)** Time-lapse microscopy showing bundling for ScMreB5$^{WT}$ and defect in bundling for ScMreB5$^{E134A}$ filaments (related to Video 3). White arrows indicate bundling events in ScMreB5 and, for ScMreB5$^{E134A}$, point to the site of septation and highlight the bundling events that happen at the time of cell division (8 cells out of 41). **(F)** Time-lapse microscopy of cells expressing ScMreB5$^{WT}$, which are undergoing division (related to Video 4). Filament disassembly can be observed after cell division. **(G)** Time-lapse microscopy of cells expressing ScMreB5$^{WT}$ shows fragmentation and reannealing or bundling of filaments (related to Video 5). Scale bar denotes 5 μm. For A, B, F, and G, cells were grown in EMM without thiamine for 24–32 h before imaging. Cells were placed on an EMM agarose pad lacking thiamine as described in Materials and methods and imaged at every 3-min time interval. For E–G, the number of cells in which such events were observed and the total number of cells are indicated in parentheses. Source data are available for this figure: SourceData F5.

structures (Fig. 1 A). The presence of positively charged residues suggested that a charge-based interaction might be mediated by the C-terminal tail. Hence, we designed a construct with the last 10 residues deleted (ScMreB5$^{ΔC10}$) and tested binding to 100% DOPG and DOPC liposomes (Figs. 7 B and S4 B). ScMreB5$^{ΔC10}$ showed an effect similar to the hydrophobic loop mutants.

To comparatively analyze liposome binding by the mutants, we chose a fixed concentration of liposomes based on a binding curve obtained for ScMreB5$^{WT}$ with increasing concentrations of liposomes (Fig. S4 C). 600 μM, a concentration just below saturation in the binding curve (Fig. S4 C), was maintained as a constant liposome concentration for further assays. Next, we repeated the pelleting assays for ScMreB5$^{WT}$ by varying ratios of DOPC:DOPG in the liposome preparation to tease out the contributing factors of lipid composition specificity (Fig. S4 D). Based on this, 80% DOPG at 600 μM of liposomes was used for the pelleting assays in further experiments (Fig. S4, C and D).

ScMreB5$^{ΔC10}$ and ScMreB5$^{I95A}$ exhibited reduced binding of the protein to 600 μM liposomes containing 80% DOPG (Fig. 7, D and E). From the above result, it appeared that membrane binding by ScMreB5 was driven by positively charged and hydrophobic residues on the membrane-binding surface. The surface potential of the membrane binding face of the modeled ScMreB5 double protofilament is also consistent with this hypothesis, which is either positively charged or hydrophobic (Fig. 7 F). Next, we explored whether the liposome interaction of ScMreB5$^{E134A}$ is affected by the mutation. Interestingly, a liposome binding assay of ScMreB5$^{E134A}$ showed a significant decrease compared with that of ScMreB5$^{WT}$ (Fig. 7, D and E), although the residue is situated away from the membrane-binding surface.

The effect of the E134A mutation on liposome binding prompted us to explore the interdependence between nucleotide state and liposome binding. We carried out liposome pelleting assays of ScMreB5$^{WT}$ upon addition of ADP, ATP, or AMP-PNP in the reaction mix. Pelleting assays of the reaction mix without liposomes showed that the protein does not pellet upon addition of nucleotide in the absence of liposome (Fig. S4 E). The observations from the pelleting assays with liposomes suggest that there is a differential binding for ScMreB5$^{WT}$ based on the nucleotide state (in the presence of ATP or ADP addition compared with AMP-PNP addition or in the absence of any nucleotide; Fig. 7, G and H), similar to the effect of the E134A mutation.

## Discussion

Among the actin filament family members, MreB filament is unique because of the antiparallel arrangements of the protofilaments, implying the absence of a kinetic or structural polarity between the two ends. Structures of MreB filaments have highlighted that the conformational changes accompanying filament formation are very similar to those observed in actin and ParM (van den Ent et al., 2014). The active site residues, including the residues coordinating Mg$^{2+}$ and those required for optimal orientation of the catalytic water, are highly conserved in MreBs and actins and across the Hsp70 superfamily members. Our ATPase activity measurements point out a role for these residues in ATP hydrolysis, emphasizing that ATP hydrolysis is an inevitable feature for MreB (as well as ScMreB5) function. Effects of the polymeric interface mutant ScMreB5$^{K57A}$ and the inter-subdomain contact mutant ScMreB5$^{D70A}$ on ATPase activity suggest allosteric communication between the ATP-binding pocket and the polymerization interface in MreB too, a conserved feature of many characterized actin family members (Chu and Voth, 2005; Vorobiev et al., 2003). An interesting observation from our biochemical and structural characterization is the identification of a potassium ion at the interface of the nucleotide and the protein, which probably stabilizes the bound nucleotide conformation of ScMreB5.

A thorough analysis of the reported crystal structures of CcMreB (van den Ent et al., 2014) showed us that the catalytic glutamate (Glu140 in CcMreB or Glu140$^{Cc}$; Glu134 in ScMreB5) functions as an interaction hub, forming a network of interactions with γ-phosphate of the nucleotide, catalytic water, and residues from all four MreB subdomains (Fig. 8, inset). The entire network of interactions (labeled i–vii in Fig. 8, inset) with all four subdomains was observed only in the double-filament conformation (PDB accession no. 4CZJ; Fig. 8) and not in the single-protofilament or monomeric states (PDB accession nos. 4CZI, 4CZF, or 4CZM in Fig. 8). The γ-phosphate of the nucleotide and Glu140$^{Cc}$ side chain play key roles in the network. Thus, the residue may act as the sensor for the ATP-bound state and trigger transition to the double-protofilament conformation—an

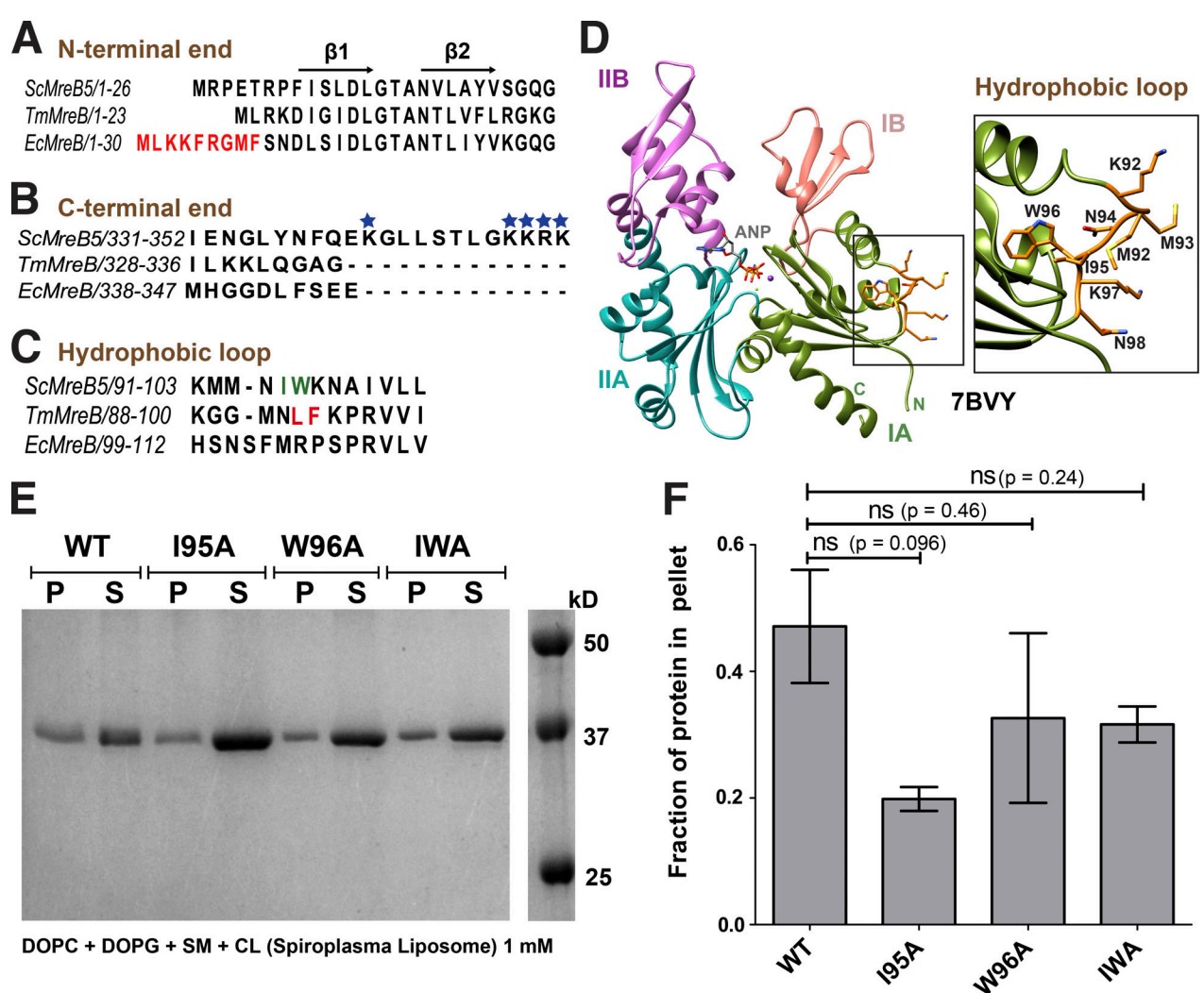

**Figure 6. ScMreB5 binds to liposomes. (A)** Sequence alignment of ScMreB5 with TmMreB and EcMreB, showing the absence of amphipathic helix at the N-terminus. Amphipathic helix of EcMreB is highlighted in red. Secondary structures are labeled on top of the alignment. **(B)** Sequence alignment of C-terminal region of ScMreB5 with TmMreB and EcMreB shows longer C-terminal tail enriched with positively charged residues (highlighted with blue stars). **(C)** Sequence alignment of ScMreB5 with TmMreB and EcMreB in the region of hydrophobic loop. The residues interacting with the membrane for TmMreB and predicted residues for ScMreB5 are highlighted in red and green, respectively. **(D)** Crystal structure of AMP-PNP–bound ScMreB5 (PDB accession no. 7BVY), with proposed membrane insertion loop (orange) in domain IA (green). Inset: Zoomed-in view of the loop. The N- and C-terminal ends of ScMreB5 are labeled N and C, respectively. **(E)** A representative 12% SDS-PAGE gel of liposome pelleting assay for comparing membrane binding of ScMreB5[WT] (denoted WT), with the hydrophobic loop mutants (single mutants ScMreB5[I95A] and ScMreB5[W96A] and double mutant ScMreB5[I95A, W96A], denoted as I95A, W96A, and IWA, respectively). P and S represent the pellet and supernatant fractions of the reaction. Concentrations of liposomes of composition mimicking *S. citri* lipids and protein used in the assay are 1 mM and 2 μM, respectively. **(F)** Plot showing relative intensities of the fraction of protein in the pellet corresponding to ScMreB5[WT] and hydrophobic loop mutants calculated from the SDS-PAGE gels (representative gel shown in E) from three independent experiments. The error bar denotes mean with SEM; unpaired two-tailed Student's *t* test; ns, P > 0.20). Source data are available for this figure: SourceData F6.

important feature of nucleotide state–dependent polymerization. Additionally, the requirement of Glu140[Cc] in providing an optimal active site geometry for ATP hydrolysis hints that the residue might also trigger efficient ATP hydrolysis, suggesting a crucial role in nucleotide-dependent polymerization dynamics.

The antiparallel double-protofilament structure of MreB highlights a repetitive arrangement within the filament, without a twist angle between monomers, ideally favoring a planar lipid membrane interaction (van den Ent et al., 2014; Salje et al., 2011). How these filaments align against curved membrane surfaces and/or bring about a curvature in liposomes is enigmatic. While

there are theoretical models on how this might be achieved (Wong et al., 2019), our study based on ScMreB5[E134A] and nucleotide dependence of liposome binding is indicative of the role of ATP-driven dynamics in polymerization and membrane binding of MreB. A hypothesis on how different nucleotide states could exhibit different modes of membrane binding by twisting of MreB filaments in the presence of a lipid bilayer was earlier put forward based on molecular dynamics simulations (Colavin et al., 2014; Shi et al., 2020). Impairment of membrane binding by ScMreB5[E134A] indicates that the conformational changes facilitated by Glu134 are required for liposome interaction. These effects could be due to a defective conformational

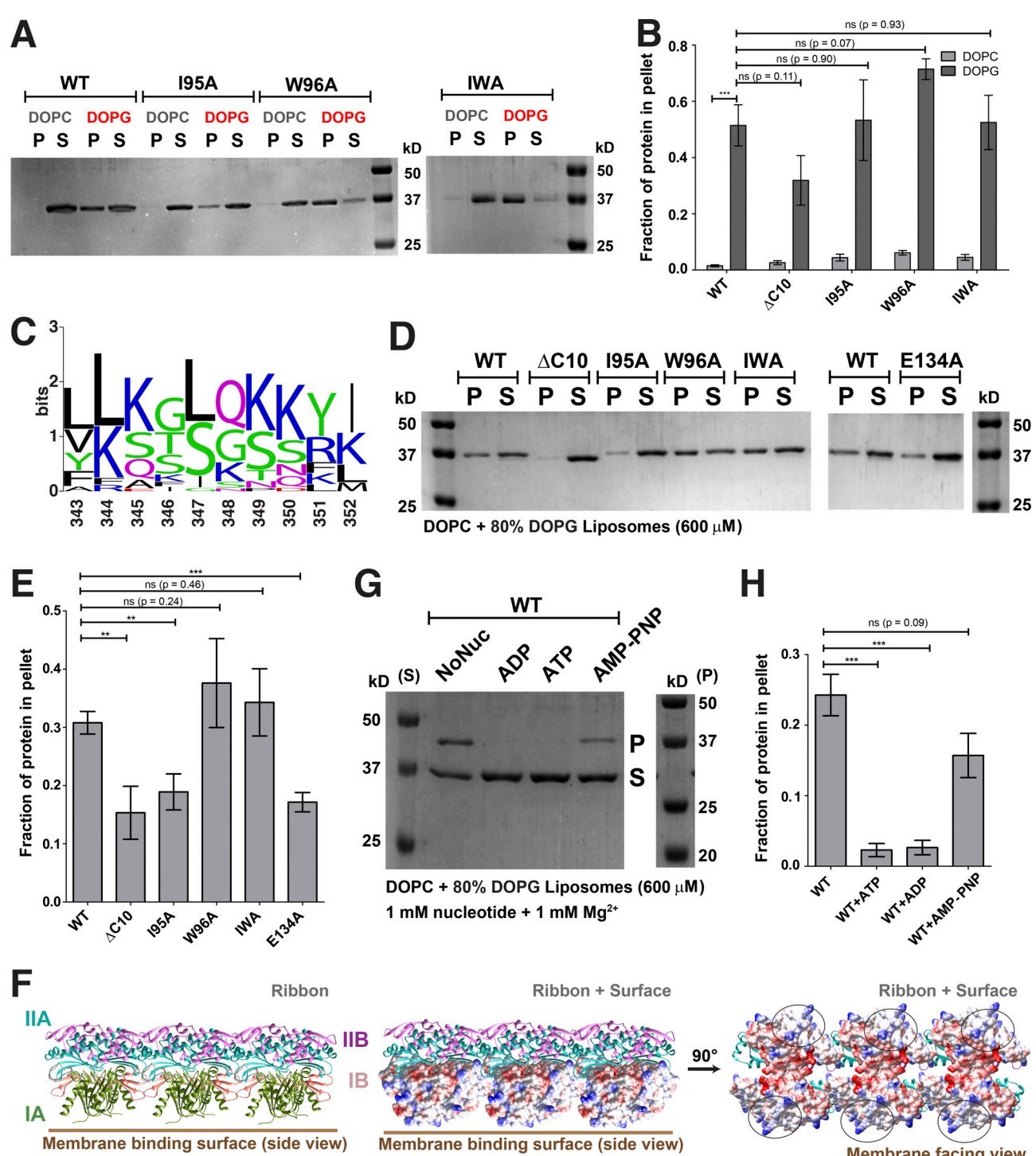

**Figure 7.  Membrane binding of ScMreB5 is modulated by lipid charge and conformational changes driven by Glu134. (A)** A representative 12% SDS-PAGE gel of liposome pelleting assay for determining membrane binding of ScMreB5[WT] and the hydrophobic loop mutants with a neutral lipid DOPC and an anionic lipid DOPG. P and S represent the pellet and supernatant fractions of the reaction. Concentrations of DOPG and DOPC liposomes and protein used in the assay are 1 mM and 2 µM, respectively. **(B)** Plot showing relative intensities of the fraction of protein in the pellet corresponding to ScMreB5[WT] and mutant constructs in the SDS-PAGE gels from five independent experiments. The binding is specifically observed for liposome composed of the anionic lipid DOPG for the ScMreB5[WT] as well as the mutants. Negligible binding is seen for both ScMreB5[WT] and the mutants for the liposome made from neutral lipid DOPC. The error bar denotes the mean with SEM; unpaired two-tailed Student's $t$ test; ns, $P > 0.05$; ***, $P < 0.0001$). **(C)** Weblogo of C-terminal end, showing the presence of lysine and arginine in *Spiroplasma* MreB5s. The numbering on the x axis is with respect to the last 10 residues of ScMreB5. **(D)** A representative 12% SDS-PAGE gel of liposome pelleting assay showing a decrease in binding for ScMreB5[WT] and mutants (ScMreB5[ΔC10], ScMreB5[I95A], ScMreB5[W96A], ScMreB5[I95A,W96A], and ScMreB5[E134A], denoted ΔC10, I95A, W96A, IWA, and E134A, respectively) at 20%:80% (DOPC:DOPG) liposome ratio. P and S represent the pellet and supernatant fractions of the reaction. Concentrations of liposomes and protein used in the assay are 600 and 2 µM, respectively. **(E)** Plot showing relative intensities of the fraction of protein in the pellet calculated from the SDS-PAGE gels from at least four independent experiments (representative image in D).

The error bar denotes the mean with SEM; unpaired two-tailed Student's *t* test; ns, P > 0.20; ***, P < 0.0001; **, P = 0.001–0.002). **(F)** Different views of membrane-binding face of double-protofilament ScMreB5. Electrostatic surface potential of the membrane-binding face (IA and IB subdomains; middle and right subpanels) of double protofilament of ScMreB5 is shown corresponding to the ribbon views of the double protofilament (left). Circled regions within the surface show the regions of positive and neutral charge for the membrane-binding face of the filament. The double protofilament of ScMreB5 was modeled using CcMreB double protofilament, PDB accession no. 4CZE. Subdomains IA and IB are colored pink and light green. **(G)** A representative 12% SDS-PAGE gel of liposome pelleting assay showing binding for ScMreB5[WT] in different nucleotide states at 20%:80% (DOPC:DOPG) liposome ratio. P and S represent the pellet and supernatant fractions of the reaction. Concentrations of liposomes and protein used in the assay are 600 and 2 µM, respectively. Concentration of nucleotides and $MgCl_2$ used are 1 mM each. **(H)** Plot showing relative intensities of the fraction of protein in the pellet in different nucleotide conditions calculated from the SDS-PAGE gel from at least four independent experiments (representative image in G). The error bar denotes the mean with SEM; unpaired two-tailed Student's *t* test; ns, P = 0.09; ***, P ≤ 0.0001). Source data are available for this figure: SourceData F7.

change cycle, a decrease in the filament content, or differences in lateral interactions between filaments.

Sheets of filaments with lateral interactions might be helpful in promoting binding and modulating the membrane curvature, as observed from in vitro experiments of MreB filaments bound to liposomes (Hussain et al., 2018; Salje et al., 2011) and synthetic reconstitution systems (Garenne et al., 2020). Different nucleotide states might possess distinct conformations or differential capabilities to form bundles or bind efficiently to specific curvatures, thus contributing to the sensing of membrane curvature by the MreB filaments. The effect of liposome binding and the accompanied conformational changes on the MreB filaments is not known; this interaction might also stimulate nucleotide

exchange and/or hydrolysis. It is possible that the filament conformations, spatial orientations, or bundling features of the different nucleotide states can (a) match the curvature of the liposomes, (b) remodel the liposomes to match the filament curvature, or (c) fall off in case of a curvature mismatch. The effects observed in this study are based on liposomes with protein added onto the exterior convex surface. The binding dependence on nucleotide state might have a different effect from a concave surface.

ScMreB5 is a major cytoskeletal protein that confers helical shape and facilitates motility in *S. citri* (Harne et al., 2020). Hence, the mechanistic basis of ScMreB5 function is of special interest to understand the minimal functional requirements of

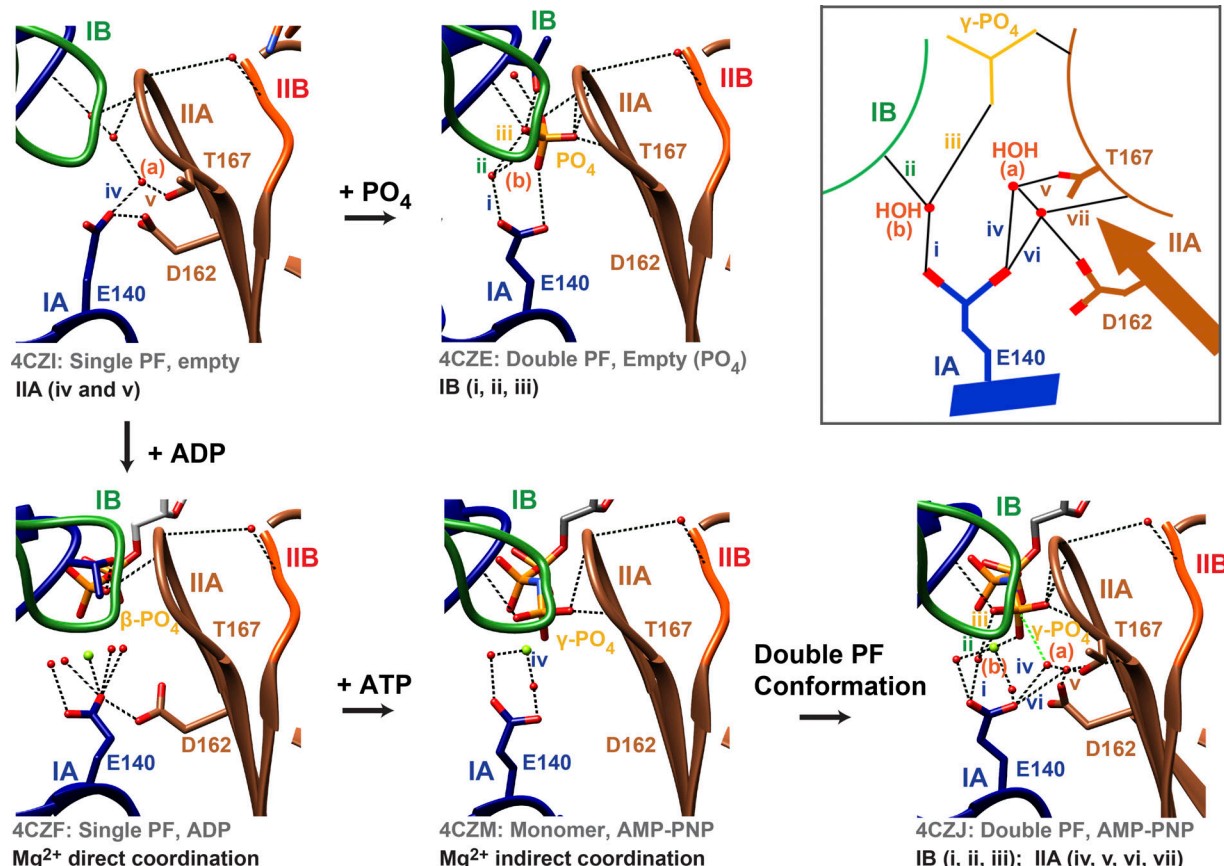

Figure 8.   **Mechanism of ATP dependence of MreB function.** Glu140[Cc] (CcMreB numbering) plays a pivotal role in ATP-dependent conformational change of MreB. Schematic (box): Glu140[Cc] in the double-protofilament AMP-PNP–bound state (PDB accession no. 4CZJ) holds the IB and IIA subdomain via water-mediated interactions, (a) and (b). These water-mediated interactions are not present for ADP single filament (PDB accession no. 4CZF) and AMP-PNP monomeric state (PDB accession no. 4CZM), where Glu140[Cc] functions only in $Mg^{2+}$ coordination.

MreBs in bacterial cell shape determination. In ScMreB5, we report a novel mode of membrane binding of MreBs assisted by surface charge-based interactions. Earlier studies on MreBs (*T. maritima* and *E. coli* MreBs) identified the membrane interaction to be abrogated completely by mutation of the hydrophobic loop or the N-terminal amphipathic helix (Salje et al., 2011). Our results show that this mode of interaction is not universal for MreBs. Our liposome binding studies with ScMreB5 indicate that charge-mediated interactions contribute to the membrane-binding features of ScMreB5. The lipid chemistry and the surface properties of ScMreB5 highlight organism-specific or paralog-specific modes of membrane binding based on sequence variation within a conserved filament architecture. These features can potentially modulate the energetics of filament interaction with the membrane, thereby affecting orientation of MreB filaments necessary for determination of cell diameter and shape. In the absence of membrane attachments facilitated by proteins related to peptidoglycan synthesis such as RodZ (van den Ent et al., 2010), a surface extensive interaction with the membrane might help in orienting the filaments in a cell-wall-less organism. Interestingly, RodZ plays an important role in circumferential movement of MreB by linking with the peptidoglycan synthesis machinery, and also in curvature-dependent localization of MreB (Morgenstein et al., 2015; Bratton et al., 2018). In the absence of RodZ and peptidoglycan synthesis in *Spiroplasma*, a novel mode of membrane binding involving an increased surface might be important for curvature sensing and remodeling.

Our studies suggest an allosteric effect of ATP binding and hydrolysis for efficient filament formation and membrane binding. The speed of processive movement of MreB filaments has been demonstrated to be independent of ATP hydrolysis (Garner et al., 2011). ATPase mutants of MreB possess localization defects in vivo, with highly localized filaments in certain areas of the cell, finally resulting in shape defects, as demonstrated for *B. subtilis* (Defeu Soufo and Graumann, 2006) and *C. crescentus* (Dye et al., 2011).

*Spiroplasma* MreB5 provides an excellent prototype for understanding not only the conserved features across all MreBs, but also the role of ATP hydrolysis in curvature sensing and plausible remodeling, especially in the absence of peptidoglycan synthesis, thereby providing insights into helicity generation and motility driven by the filaments. Our observations highlight that the fundamental features of MreB are conserved, independent of the involvement of the peptidoglycan synthesis machinery. Further studies of ScMreB5 in *S. citri* will ascertain the mechanistic basis of the role of ATP hydrolysis in motility and helicity.

## Materials and methods
### Cloning
*S. citri mreB5* gene was amplified from the genomic DNA of *S. citri* (21833; DSMZ). The amplified product was cloned into pHis17 vector (for the sequence, refer to Addgene catalog #78202) between NdeI and BamHI restriction sites by restriction-free cloning method (van den Ent and Löwe, 2006).

This resulted in a C-terminal hexahistidine-tagged construct with GSHHHHHH added after the last residue. Different single/double point mutants as well as the deletion construct were generated by restriction-free cloning using suitable primers. The clones and primers are listed in Table S3.

### Expression and purification of ScMreB5
Plasmid containing the gene of interest was transformed into *E. coli* BL21-AI cells. Cultures were grown in Luria Bertani medium supplemented with ampicillin (final concentration 100 µg/ml) at 37°C until $OD_{600}$ 0.8–1.0 was reached. The culture was induced with 0.05% arabinose (final concentration) and was further grown at 20°C for 12 h after induction. The same procedure was followed for expression of all the mutants. The culture was spun at 6,000 *g*, and the cell pellets were flash-frozen and stored at –80°C until further use.

For purification, a cell pellet from a 2-liter culture was thawed, and cells were homogenized in lysis buffer (50 mM Tris, pH 8, 200 mM NaCl, and 10% glycerol) and sonicated. The lysate was centrifuged at 44,082 *g* for 45 min at 4°C. Supernatant was loaded on a 5-ml Ni-NTA column (HisTrap; GE Healthcare) pre-equilibrated with buffer A (50 mM Tris, pH 8, and 200 mM NaCl). Hexahistidine tag present at the C-terminus of the protein facilitated binding to the Ni-NTA column. Bound protein was eluted using a step gradient of 5, 10, 20, 50, and 100% buffer B (50 mM Tris, pH 8, 200 mM NaCl, and 500 mM imidazole) with buffer A. Fractions containing purest protein were identified on a 12% SDS-PAGE gel. 1 mM ADP and 1 mM $MgCl_2$ (final concentrations) were added to those fractions to minimize protein precipitation. The protein was concentrated, and imidazole was removed in buffer exchange while concentrating. Finally, the protein was obtained in the buffer (50 mM Tris, pH 8, 50 mM NaCl, 1 mM ADP, and 1 mM $MgCl_2$). Protein was flash-frozen in aliquots and stored at –80°C until further use. This protein was used for thermal shift assay and crystallization of ScMreB5 with ADP.

After optimizing the purification protocol on the basis of thermal shift assay, purification with the optimized conditions was performed similarly as described above, with the following changes. In lysis buffer, buffer A, and buffer B, 200 mM NaCl was replaced with 300 mM KCl. Dialysis was performed after Ni-NTA elution for the fractions containing the purified protein against 50 mM Tris, pH 8, and 300 mM KCl. ADP and $MgCl_2$ were not added at any stage during purification. Analytical size exclusion was performed using a Superdex 75 or 200 (GE Life Sciences). All experiments other than thermal shift assay and crystallization of ADP-bound ScMreB5 were performed using the protein purified without extra addition of ADP and $MgCl_2$.

### Crystallization of ScMreB5–ADP
Approximately 1,000 conditions were screened from the commercially available crystallization screens (Molecular Dimensions; Hampton Research) using a drop size of 100 nl protein (5 mg/ml) and 100 nl crystallization condition. Initial hits obtained were further optimized. Well-diffracting, needle-shaped crystals were obtained in the condition containing 5 mg/ml protein (in 50 mM Tris, pH 8, 50 mM NaCl, 1 mM ADP, and

1 mM MgCl₂ buffer at the end of purification steps) crystallized with 2 mM ADP and 2 mM MgCl₂ in a condition containing 0.15 M Na-K phosphate and 16% PEG 3350, pH 7.8, by hanging drop method at a 1:1 ratio. Crystals were frozen in 20% glycerol as the cryoprotectant contained in the parent condition. Data collection and refinement statistics for ScMreB5–ADP are tabulated in Table S1.

The identity of the potassium ion bound to ScMreB5 in the crystal was established using x-ray fluorescence scanning using ScMreB5–AMP-PNP crystal. ScMreB5–AMP-PNP was crystallized with 2 mM AMP-PNP and 2 mM MgCl₂ in the buffer containing 0.15 M Na phosphate and 16% PEG 3350, pH 7.8 (Harne et al., 2020). To wash out any remnants of potassium ions in the cryoprotectant solution, crystals were picked up and successively transferred into three different drops of 20% glycerol cryoprotectant contained in the parent condition before freezing. X-ray fluorescence scanning was performed at the beamline I-04 Diamond Light Source, UK.

### Structure determination of ScMreB5–ADP
Data were collected at the home source, Rigaku MicroMax-007 HF. Crystal diffracted until 2.3 Å. Data reduction was performed using IMOSFLM (Battye et al., 2011) and scaling using AIMLESS (Evans and Murshudov, 2013), followed by molecular replacement using PHASER (McCoy et al., 2007) accessed through the CCP4 package (Potterton et al., 2018). Refinement was carried out using PHENIX package (Adams et al., 2010), and model building was done using Coot (Emsley et al., 2010). ScMreB5–ADP bound structure (PDB accession no. 7BVZ) was solved using CcMreB (PDB accession no. 4CZI) as model for molecular replacement. Composite omit maps for confirming the ligand densities were calculated using PHASER (McCoy et al., 2007).

### Thermal shift assay
For increasing the stability of the protein and optimizing purification, a thermal shift assay was performed (Ericsson et al., 2006). A final concentration of 2.6 µM of protein (ultracentrifuged at 100,000 g for 25 min at 4°C) in a 25-µl reaction was used in this assay. 2 µl of 50× SYPRO Orange (S5692; Sigma-Aldrich) was used as a fluorophore. The reaction was carried out in 96-well PCR plates (Bio-Rad) that were sealed with a microseal (Bio-Rad) after addition of all the components. The plate was spun for 30 s at 4°C before taking the readings. Bio-Rad CFX96 Real-Time System was used for measuring the $T_m$ (melting temperature) of the protein by monitoring the change in fluorescence of SYPRO Orange as the protein unfolded. The plate was first incubated in the machine at 4°C for 10 min. Subsequently, readings were taken from a temperature range of 4–90°C with an increase of 0.4°C every 20 s. Data for the change in SYPRO fluorescence emission as it bound to the hydrophobic pockets of the protein were collected using the flourescence resonance energy transfer channel (excitation at 470 nm and emission at 569 nm). The first derivative of the of the raw data ($-(dF)/dT$) was plotted with respect to temperature for a single repeat of the experiment ($N = 1$). Prism v5.00 for Windows (GraphPad) was used for plotting the graphs.

### HPLC run for detecting presence of bound nucleotide
20 mmol of purified ScMreB5$^{WT}$ was denatured at 75°C. The denatured protein was spun at 21,000 g for 20 min at 4°C. Supernatant was filtered with 0.22-µm cellulose acetate filter (Corning), and the sample was loaded onto the pre-equilibrated (with buffer A, 2 mM Tris, pH 8) DNAPac PA200 ion-exchange column (Thermo Fisher Scientific). The runs were performed with a linear gradient of 0–20% buffer B (2 mM Tris, pH 8, and 1.25 M NaCl) for three column volumes, 20–40% buffer B for three column volumes, and 100% buffer B for two column volumes. 40 mmol of filtered solutions of ATP or ADP was used as a standard in the run. The absorbance at 255 nm was plotted against the conductivity for all the runs in Prism v5.00 for Windows.

### Phosphate release assay for estimating ATPase activity
The release of inorganic phosphate during ATP hydrolysis was measured by malachite green assay (Feng et al., 2011). Protein was prespun at 22,000 g at 4°C for 20 min. Protein was added at a final concentration of 10 µM to the master mix of buffer containing ATP and MgCl₂ to achieve final concentrations of 1 mM ATP and 1 mM MgCl₂ in buffer A (300 mM KCl and 50 mM Tris, pH 8) and mixed. To stop the reaction for a 0 time point reading, 20 µl reaction was immediately mixed with 5 µl of 0.5 M EDTA in a 96-well plate. The rest of the master mix was incubated at 25°C for 60 min. After 60 min, the reaction was stopped with 0.5 M EDTA in the same manner. Simultaneously, phosphate standards were freshly diluted from 400 µM NaH₂PO₄ (S0751; Sigma-Aldrich). To measure the amount of phosphate release, malachite green solution was freshly prepared using 800 µl of 3.5 mM Malachite Green (38800; Sigma-Aldrich) dissolved in 3N H₂SO₄, 16 µl of 11% Tween 20 (P-1379; Sigma-Aldrich), and 200 µl of 7.5% wt/vol ammonium molybdate (277908; Sigma-Aldrich). 50 µl of malachite green solution was added to the stopped reactions and phosphate standards. Absorbance of Malachite Green was measured in Varioskan Flash (4.00.53) 5–8 min after addition at 630 nm wavelength.

For calculating the $k_{obs}$ (min⁻¹), first, the slope was calculated from the phosphate standards. The absorbance of protein containing reaction was calculated by subtracting the blank reaction absorbance (without protein). The amount of phosphate release (in µM) after 60 min was calculated by dividing the subtracted absorbance by the slope. To calculate the phosphate release per min ($k_{obs}$), the amount of phosphate release (in µM) from protein was divided by 60 min (µM min⁻¹) and then by 10 µM (min⁻¹).

Prism v5.00 for Windows was used for statistical analysis and plotting the graphs. Statistical significance was estimated by unpaired t test, two tailed. The data in the graph are expressed as mean ± SEM.

### Light scattering assay for monitoring polymerization
Protein was prespun at 22,000 g at 4°C for 20 min. Around 4 mg/ml of 300 µl of protein was injected into Superdex 75 (GE Life Sciences) that was pre-equilibrated with buffer A (50 mM Tris, pH 8, and 200 mM KCl). 200 µl of eluted monomeric fraction of protein was immediately taken for light scattering measurement. Measurement was performed in FluoroMax-4

(Horiba), with excitation and emission slit widths of 2 nm each and excitation and emission wavelength of 400 nm at 25°C. Readings were taken for 1,000 s for WT and 2,500 s for the polymerization mutant K57A. Simultaneously, protein estimation of the fraction subjected for light scattering was performed.

## Liposome preparation

All the lipids used in the experiments were purchased from Avanti Polar Lipids, namely, DOPC (850375C), DOPG (840475C), porcine brain sphingomyelin (860062C), and Cardiolipin (710335C). For performing pelleting assays with liposomes mimicking *S. citri* lipid composition, the following protocol (earlier reported in Harne et al., 2020) was used. A stock concentration of 2 mM lipids in chloroform was made for *S. citri* membrane mimic, only DOPG, only DOPC, and varying percentage ratios of DOPC:DOPG lipids. Chloroform solution of lipids was aliquoted in a clean test tube and dried. Dried lipids were resuspended in buffer A (50 mM Tris, pH 8, and 300 mM KCl) and 1 mM $MgCl_2$. This lipid solution was extruded through a 100-nm polycarbonate membrane (Avanti Polar Lipids) to get liposomes of 100-nm range. These liposomes were further used in the charge specificity and nucleotide-dependent liposome pelleting assays.

## Liposome pelleting assay

Protein was spun at 22,000 $g$ at 4°C for 20 min to remove any precipitation. From the supernatant, 2 µM (final concentration) protein was added to the reaction mixture of 100 µl containing buffer A (300 mM KCl, and 50 mM Tris, pH 8), 1 mM $MgCl_2$, and liposomes. This mixture was further incubated at 25°C for 15 min and spun at 100,000 $g$ for 25 min at 25°C. Supernatant was removed, and pellet was resuspended in 50 µl buffer A. Supernatant and pellet were mixed with 2× Laemmli buffer, and equal amounts of both were loaded onto the 12% SDS-PAGE gel. This protocol was followed for the following assays: (a) to compare the binding of ScMreB5^WT and mutants with liposome composition resembling *Spiroplasma* membrane composition at a concentration of 1 mM liposome; (b) to determine charge-based binding specificity of ScMreB5^WT and the mutants with liposomes prepared from 1 mM DOPG and 1 mM DOPC, respectively; (c) to determine the binding curves of ScMreB5^WT with increasing liposome concentration (0–1 mM); (d) to estimate the binding properties for ScMreB5^WT and mutants at 600 µM liposome concentration at varying ratios of DOPG and DOPC in the liposome mix; (e) to determine the binding specificity for ScMreB5^WT and the mutants at 600 µM (80% DOPG and 20% DOPC) liposome concentration; and (f) to determine the binding specificity for the WT in the presence of 1 mM ADP/ATP/AMP-PNP at 600 µM (80% DOPG and 20% DOPC) liposome concentration (here the protein was preincubated for 5 min at 25°C with the nucleotides before liposome addition).

The intensity analysis of the protein band was performed in ImageJ 1.52n (Rueden et al., 2017). For calculating the fraction of protein in the pellet, band intensity in the pellet fraction was divided by the sum of band intensities in pellet and supernatant. The data in the graph are expressed as the mean ± SEM. Statistical significance was estimated by unpaired $t$ test, two-tailed. Prism v5.00 for Windows was used for plotting the graphs.

## Sequence and structure analyses

For comparing ScMreB5 ADP- and AMP-PNP–bound structures as well as domain-wise comparison of ScMreB5-ADP and ScMreB5-AMP-PNP with CcMreB, UCSF Chimera v1.13.1 (Pettersen et al., 2004) was used. Each subdomain of ScMreB5 was individually superposed on CcMreB subdomains using Match Maker option in Chimera with default settings. RMSD values were obtained for pruned C-α atom pairs (generated by iteratively pruning C-α atom pairs until each atom pair was within a 2-Å distance cutoff).

For nucleotide-binding pocket residues comparison of ScMreB5 with different CcMreB and actin structures, C-α atom pairs of subdomain IIA of CcMreB and actin monomer were superposed over C-α atom pairs of ScMreB5 subdomain IIA using Match command in Chimera. The resulting superposed structures were then analyzed at the nucleotide-binding pocket region.

Sequence alignment of MreBs and actin, and their corresponding percentages of sequence identities (Fig. S2), were obtained using Clustal Omega (Sievers et al., 2011), and the figure was generated using ESPript v3.0 (Robert and Gouet, 2014).

For ScMreB5 C-terminus sequence conservation analysis, protein sequences of MreB5s of *Spiroplasma* species listed in Harne et al. (2020) were included for the sequence alignment. All the sequences were aligned using Clustal Omega (Sievers et al., 2011) with default settings. The alignment generated was analyzed and edited in JalView (Clamp et al., 2004). The C-terminus alignment conservation figure (Fig. 7 C) was generated using WebLogo (Crooks et al., 2004).

For generating the double-protofilament assembly model of ScMreB5–AMP-PNP, coordinates of the double-protofilament assembly of CcMreB (PDB accession no. 4CZJ) were generated by displaying and saving the coordinates of the symmetry mates using Coot (Emsley et al., 2010). Each subdomain of ScMreB5–AMP-PNP was saved as a separate PDB file through UCSF Chimera v1.13.1 (Pettersen et al., 2004). Each subdomain (IA, IB, IIA, and IIB) of ScMreB5–AMP-PNP was superposed on one of the protofilament monomers of the CcMreB filament using Match Maker option in Chimera with default settings. Similarly, ScMreB5 subdomains were superposed on the other two monomers of the same protofilament of CcMreB double-protofilament assembly. This generated a single protofilament of ScMreB5–AMP-PNP in a double-protofilament model. A second protofilament of ScMreB5 was also generated by repeating the same steps; this resulted in a double-protofilament model of ScMreB5–AMP-PNP. The electrostatic potential surface of this model was generated using Electrostatic Surface Coloring option in Chimera with default settings. All the figures related to crystal structure of ScMreB5 were prepared using UCSF Chimera v1.13.1 (Pettersen et al., 2004).

## Cryo-EM

For visualizing the filaments of ScMreB5^WT and the ATPase mutant ScMreB5^E134A, cryo-EM was carried out. Quantifoil Au 1.2/1.3 grids that were glow discharged for 90 s were used. Protein was ultracentrifuged at 100,000 $g$ for 25 min at 4°C. 5 mM nucleotide (AMP-PNP, ADP, or ATP) and 5 mM $MgCl_2$

(final concentrations) were added to a final concentration of 50 µM protein and incubated at 25°C for 10–15 min. 3 µl of the sample was put on the grid and incubated for 5–10 s before blotting for 3 s, followed by plunge-freezing into liquid ethane for vitrification using an FEI Vitrobot. For image acquisition, grids were mounted on a Triton-Krios 300 KeV electron microscope with Falcon-3 direct electron detector, and images were taken at a magnification of 59,000×. Images of the filaments were generated in ImageJ 1.52n (Rueden et al., 2017).

## Yeast strains and growth conditions

GFP-ScMreB5$^{WT}$ and GFP-ScMreB5$^{E134A}$ were expressed from the thiamine-repressible medium-strength *nmt41/42* promoter (Basi et al., 1993). *pREP41-GFP-ScMreB$^{WT}$* (*leu+*) and *pREP41-GFP-ScMreB$^{E134A}$* (*leu+*) expression vectors were transformed in *Schizosaccharomyces pombe* strain MBY192 (*h⁻ leu1-32 ura4-D18*; lab collection) using Li-acetate method (Keeney and Boeke, 1994). Yeast cells were grown on Edinburgh minimal medium (EMM) with the addition of specific supplements (histidine, uracil, and adenine). Cultures were grown in the absence of thiamine, as indicated below in EMM, to allow expression of the protein.

## Light microscopy

*S. pombe* strains carrying GFP-ScMreB5$^{WT}$ and GFP-ScMreB5$^{E134A}$ were grown in EMM broth for 24–32 h at 36°C in the absence of thiamine for the expression of protein and imaging. Cultures were intermittently diluted into fresh medium to maintain exponential growth. Cultures in early exponential phase (OD$_{600}$ of 0.2–0.4) were used for live cell imaging, and cultures in mid-exponential phase (OD$_{600}$ 0.8–1.0) were used for all other experiments. For counting of cells and comparison of number of cells with polymer bundles, cultures were grown for 36–40 h, unless indicated otherwise in the figure legends, with intermittent dilution into fresh medium to maintain exponential growth. At least 1,000 cells were counted for each sample, and the experiment was repeated at least three times. For live-cell microscopy, 1 ml cells after 24–32 h of growth in the absence of thiamine was pelleted down at 855 *g* for 2 min and reconstituted in 50–100 µl fresh medium. 2 µl cell suspension was then mounted on EMM agarose slides, and a coverslip was placed and sealed with VALAP. Further images were collected in Z-steps of 0.2 or 0.5 µm for time lapse at a fixed 3-min interval for 6–12 h by using an epifluorescence image restoration microscope (DeltaVision Elite) equipped with a 100×, 1.4-NA oil-immersion objective. Images were acquired using an interline CCD camera, Photometrics CoolSnap HQ2. GFP-ScMreB5 was imaged using excitation and emission filters of 475/28 nm and 525/48 nm, respectively. UltimateFocus was used to maintain the cells in focus during the entire duration of imaging. A constrained iterative deconvolution (Agard, 1984) was performed using SoftWorx software. 3D-SIM was performed using DeltaVision OMX-SR Blaze with cells mounted on an agarose pad as described above. Raw images were acquired using a 60×, NA 1.42 oil-immersion objective lens and a PCO Edge 4.2 sCMOS camera and reconstructed using the SI reconstruction module of SoftWorx software. All images were processed by using Fiji (v2.0.0-rc-69/1.52p; Schindelin et al., 2012).

## Quantitative analyses of ScMreB filaments

Cells were processed for imaging, and all images were acquired as described above. Fixed exposure time (0.25 s), binning (1 × 1), illumination, and camera gain (0.5) were used. These parameters were kept constant for all the images and replicates, with care taken so that none of the pixel intensities were saturated but thinner ScMreB filaments were still visible. Both differential interference contrast (DIC) and fluorescence channels were imaged, and z-stacks (0.2-µm step size) were obtained. Image restoration was carried out using SoftWorx with not more than 10 iterations by iterative constrained deconvolution (Agard, 1984), which has been validated for use in quantitative measurements (Swedlow et al., 2002). Maximum-intensity projections were created from these deconvolved images using SoftWorx and saved as 16-bit TIFF images. For estimation of the mean fluorescence intensity in cells, sum intensity projections were created, and the average background fluorescence outside of the cell was subtracted from the measured intensity values. The mean fluorescence intensity was calculated as the integrated density divided by the area of the cell. Cell outlines were drawn using the freehand tool in Fiji (Schindelin et al., 2012) on DIC images that had the best focus and stored as regions of interest (ROIs) using the ROI manager feature.

The CV of intensities, a metric for cytoskeletal bundling, was measured as described (Higaki et al., 2020). All images were analyzed using Fiji (v2.0.0-rc-69/1.52p; Schindelin et al., 2012) and preprocessed as described (Higaki, 2017), except that we did not skeletonize the images, as it has been found to be unnecessary when images are of high contrast (Henty-Ridilla et al., 2014). The regions outside of the ROIs were masked using the color picker tool, setting it black, and filling the inverse of the selected ROI. CV was obtained by running the macro to calculate cytoskeleton bundling indicators (Higaki et al., 2020). Anisotropy, which measures how well the filament arrays are ordered, was measured using the ImageJ plugin FibrilTool, which uses a concept based on nematic tensor (Boudaoud et al., 2014). An anisotropy value of 0 would be no order or isotropic distribution of filaments, and a value of 1 would imply perfectly ordered or parallel filament arrays. For measurement of density (amount of MreB polymers per unit area in the cell), images were preprocessed as mentioned above except that a skeletonization step as described (Higaki, 2017) was included.

## Immunoblotting

Yeast cultures expressing ScMreB$^{WT}$ or ScMreB$^{E134A}$ were grown for 30–36 h at 30°C in Erlenmeyer flasks to OD$_{600}$ 1.0 with intermittent subculturing. Whole-cell lysates of *S. pombe* were prepared by TCA precipitation. Briefly, 2 ml culture was harvested by centrifugation at 2,152 *g* for 5 min and resuspended in a freshly prepared mixture of 900 µl of 2 N NaOH and 100 µl β-mercaptoethanol solution. The suspension was vortexed and left on ice for 10 min. Subsequently, 200 µl of a 55% (wt/wt) TCA solution was added, vortexed, and kept on ice for further 10 min. The suspension was centrifuged at maximum speed (16,873 *g*)

for 15 min at 4°C. The pellet was resuspended in 50 µl high-urea buffer (8 M urea, 5% SDS, 200 mM Tris-Cl, pH 6.8, and 0.1 mM EDTA) containing 100 mM DTT, heated at 65°C for 15 min, and centrifuged briefly before subjecting to 12% SDS-PAGE. The proteins in the gel were transferred to a polyvinylidene difluoride membrane (Immobilon-P) using a semidry transfer apparatus (Amersham Life Sciences or Trans Blot Turbo Transfer System from Bio-Rad). Membranes were blocked using PBS containing 3% BSA and 0.05% Tween-20 or by completely drying as described in the method of rapid immunodetection (https://www.merckmillipore.com/) after the transfer. Subsequently, antibodies at appropriate dilutions were made in PBS containing 1% BSA and 0.05% Tween-20 and incubated for 1 h at room temperature. The blots were washed three times in PBS containing 0.05% Tween-20. Final detection was carried out using chemiluminescence HRP substrates (Immobilon Western Chemiluminescent HRP Substrate; WPKLS0500; Merck Millipore) and imaged using ChemiDoc XRS+ (Bio-Rad). GFP was detected using anti-GFP antibodies (B2) conjugated to HRP (sc-9996-HRP; Santa Cruz) at a dilution of 1:7,000. α-Tubulin was detected using anti–α-tubulin antibodies (DHSB 12G10) at a dilution of 1:5,000 and HRP-conjugated goat anti-mouse IgG antibodies (A4416; Sigma-Aldrich) at a dilution of 1:4,000. Band intensities were quantified using the built-in gel analysis feature in Fiji (v2.0.0-rc-69/1.52p; Schindelin et al., 2012) and normalized to tubulin, which was used as a loading control.

### Statistical analyses

For all the biochemical studies, statistical tests, SEM, and P values were calculated and plotted using GraphPad Prism. The details of each are mentioned in the figure legends as well as in the corresponding sections of Materials and methods. All the biochemical experiments were performed at least three times with technical replicates ($n$) and/or biological replicates ($N$, different batches of protein purification).

For all the yeast-based experiments, mean values, SD, SEM, 95% CI ($4 \times$ SEM for $N = 3$), and P values were calculated using Excel for Mac 2011 (v14.4.2) or Excel Office 365. Assessment of statistical significance, using parametric tests such as Student's $t$ test (two-tailed paired or unpaired as appropriate), and error bars are as stated in figure legends and are inferential (Cumming et al., 2007). We did not formally test for the normality of the data and assumed normal distribution. All experiments were repeated at least three times, and independently grown cultures from frozen glycerol stocks that were freshly streaked out each time on agar plates were considered as biological replicates ($N$). Prism v5.00 for Windows (GraphPad), Excel for Mac 2011 (v14.4.2), or Excel Office 365 was used for plotting the graphs. Superplots were plotted as described (Lord et al., 2020).

### Online supplemental material

Fig. S1 shows that ScMreB5 is stabilized by KCl and nucleotides. Fig. S2 shows that residues at nucleotide-binding pockets are well conserved in ScMreB5. Fig. S3 shows that ScMreB5$^{WT}$ and mutants are well folded. Fig. S4 shows that ScMreB5 binding to DOPG liposome is specific and concentration dependent. Video 1 shows 360° volume-rendering 3D-SIM images of ScMreB5$^{WT}$ and ScMreB5$^{E134A}$ filaments (corresponding frames shown in Fig. 4 B). Video 2 shows polymerization of ScMreB5$^{WT}$ and ScMreB5$^{E134A}$ in *S. pombe* cells (corresponding frames shown in Fig. 5 A). Video 3 shows ATP hydrolysis mutant ScMreB5$^{E134A}$ defects in bundling of filaments compared with ScMreB5$^{WT}$ filaments (corresponding frames shown in Fig. 5 E). Video 4 shows disassembly of ScMreB5$^{WT}$ filaments (corresponding frames shown in Fig. 5 F). Video 5 shows fragmentation and annealing of ScMreB5$^{WT}$ filaments (corresponding frames shown in Fig. 5 G). Table S1 lists data collection and refinement statistics of *S. citri* MreB5 bound to ADP. Table S2 lists $k_{obs}$ values of WT and mutants. Table S3 lists primers and clones.

## Acknowledgments

The authors thank Dr. Nimisha Sharma (Guru Gobind Singh Indraprastha University, New Delhi, India) for initial help with reagents for yeast growth and plasmids. Staff members at the Centre of Interdisciplinary Sciences, National Institute of Science Education and Research (NISER), for routine maintenance of OMX-SR super-resolution imaging facility; Ajay Kumar Sharma for initial help with 3D-SIM imaging; and Shrikant Harne, Indian Institute of Science Education and Research (IISER) Pune, for initial assistance with cloning, purification standardization, and sequence analysis for the project are acknowledged. The macromolecular crystallography facility at IISER Pune and synchrotron facilities at European Synchrotron Radiation Facility (ESRF), Grenoble, Diamond Light Source (MX22637) and Department of Biotechnology India for facilitating data collection at ID29, ESRF and the National Electron Cryomicroscopy Facility, Bangalore Life Sciences Cluster, India (K.R.Vinoth Kumar and Mamta Bangera) are acknowledged.

This work was initiated with the support from Department of Science and Technology INSPIRE Faculty Fellowship (IFA12/LSBM-52) and Innovative Young Biotechnologist Award (BT/07/IYBA/2013) to P. Gayathri. The work is currently funded by Department of Biotechnology Membrane Structural Biology Program grant (BT/PR28833/BRB/10/1705/2018) and IISER Pune to P. Gayathri, and Science and Engineering Research Board (EMR/2016/000487), Department of Biotechnology (BT/PR15183/BRB/10/1443/2015 and BT/INF/22/SP33046/2019), and intramural core funding from Department of Atomic Energy to R. Srinivasan. We also acknowledge fellowships from IISER Pune and Infosys Foundation (V. Pande), NISER, Department of Atomic Energy (N. Mitra), and INSPIRE (S.R. Bagde).

The authors declare no competing financial interests.

Author contributions: V. Pande designed and performed all experiments other than those mentioned below, analyzed the data, and wrote the manuscript; N. Mitra performed and analyzed the yeast microscopy experiments; S.R. Bagde standardized the initial purification of ScMreB5 and crystallized the ScMreB5–ADP complex; R. Srinivasan designed, analyzed, and supervised the yeast microscopy experiments; N. Mitra and R. Srinivasan wrote the sections pertaining to yeast experiments; P. Gayathri designed, conceptualized, and supervised the study and wrote the manuscript. All authors reviewed and provided input on the manuscript.

Submitted: 16 June 2021

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

*Provided online are Table S1, Table S2, and Table S3. Table S1 lists data collection and refinement statistics of **S. citri** MreB5 bound to ADP. Table S2 lists **k**$_{obs}$ values of WT and mutants. Table S3 lists primers and clones.*

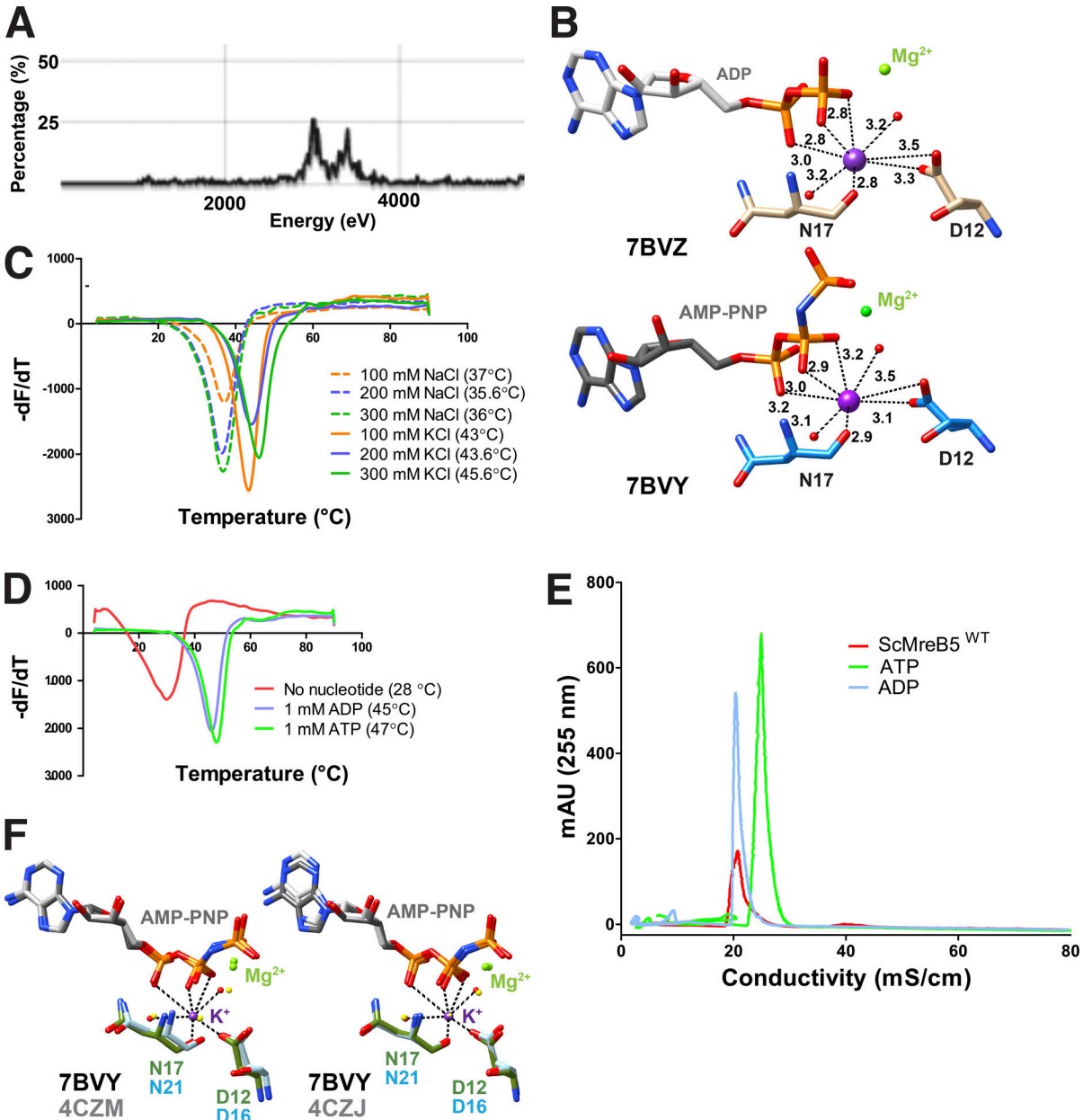

Figure S1.   **ScMreB5 is stabilized by KCl and nucleotides. (A)** X-ray fluorescence scan for ScMreB5-AMP-PNP crystals. **(B)** Coordination sphere of potassium in ScMreB5-ADP (top) and ScMreB5-AMP-PNP (bottom). **(C)** Melting curve for ScMreB5 showing $T_m$ for varying concentrations (orange, 100 mM; purple, 200 mM; and green, 300 mM) of NaCl (dotted line) and KCl (solid line). **(D)** Melting curve for ScMreB5 showing $T_m$ of ScMreB5 without any nucleotide (red), 1 mM ADP (blue), and 1 mM ATP (green). **(E)** HPLC run in DNAPac PA 200 column for 20 mmol of denatured ScMreB5$^{WT}$ supernatant fraction shows absorbance at 255 nm (milli absorbance unit [mAU]) of bound ADP peak compared to the standard ADP run. ADP and ATP standards are 40 mmol each. **(F)** Asp12 and Asn17 of ScMreB5 at the potassium-binding site are compared with the corresponding residue present in monomeric (PDB accession no. 4CZM) and double-protofilament CcMreB (PDB accession no. 4CZJ) by superposing IIA subdomain of CcMreBs onto IIA subdomain of ScMreB5–AMP-PNP structure (single protofilament conformation). The presence of a water molecule in the CcMreB at the potassium-equivalent position can be observed. The residues of ScMreB5 are colored domain-wise; those of CcMreB are light blue. The water molecules for ScMreB5 are red, and for CcMreBs, yellow.

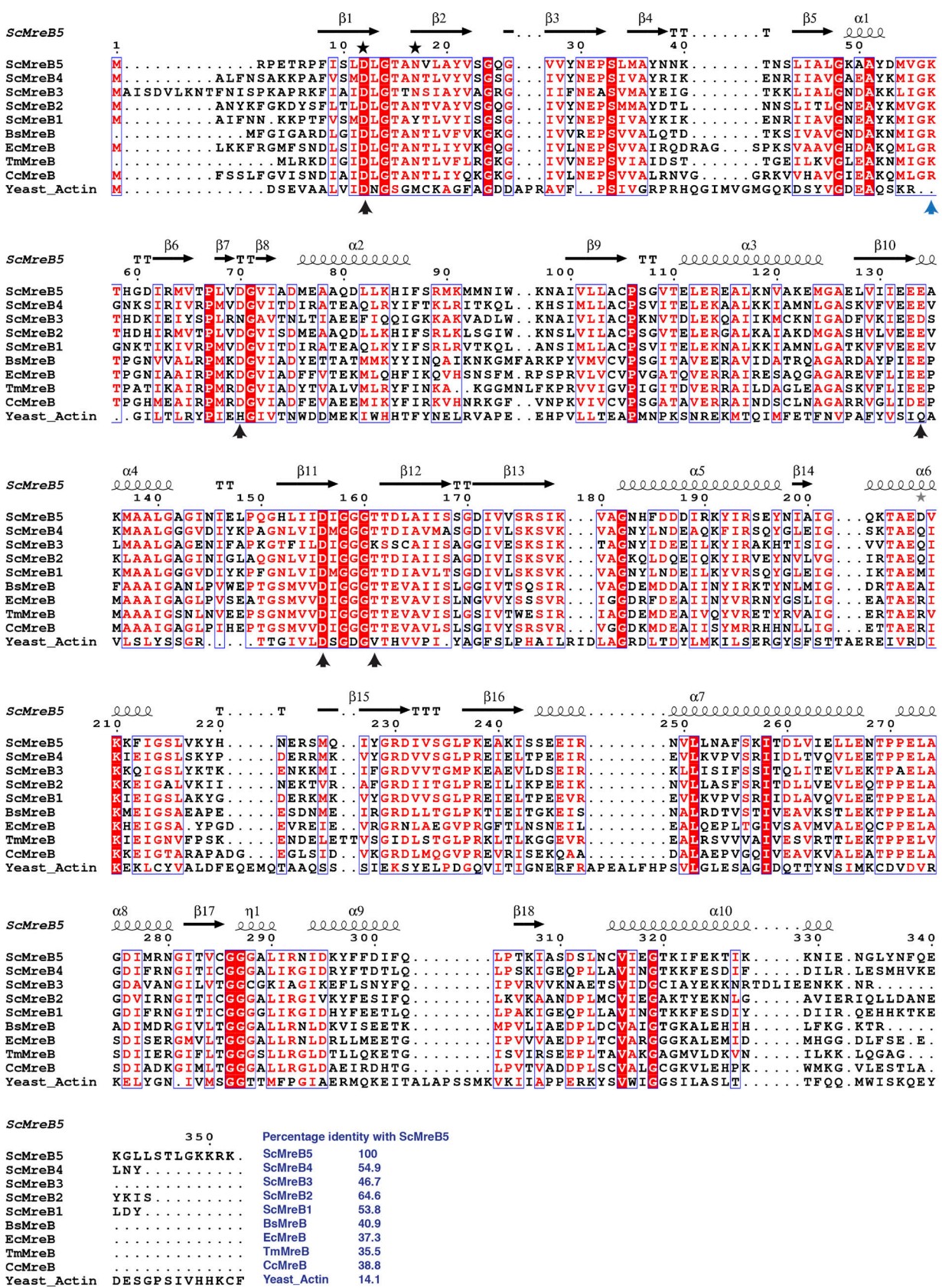

Figure S2. **Residues at nucleotide-binding pocket are well conserved in ScMreB5.** Sequence alignment of ScMreB5 (ScMreB5) with other *S. citri* MreBs (ScMreB1, ScMreB2, ScMreB3, and ScMreB4), CcMreB, TmMreB, EcMreB, *B. subtilis* MreB (BsMreB), and yeast actin. Residues involved in ATP hydrolysis (black arrow), K⁺ coordination (black star), and polymerization interface (blue arrow) are marked. Percentage identities of each sequence with ScMreB5 are marked at the end of the sequence alignment.

Figure S3. **ScMreB5<sup>WT</sup> and mutants are well folded. (A)** Representative gels of SDS-PAGE profile of purified protein samples of ScMreB5<sup>WT</sup> and the mutant constructs. **(B)** Analytical size-exclusion chromatography using Superdex 200 for ScMreB5<sup>WT</sup> (WT); ATPase mutants ScMreB5<sup>D12A</sup>, ScMreB5<sup>D70A</sup>, ScMreB5<sup>D156A</sup>, and ScMreB5<sup>T161A</sup> (D12A, D70A, D156A, and T161A); and membrane-binding mutants ScMreB5<sup>I95A</sup>, ScMreB5<sup>W96A</sup>, ScMreB5<sup>IWA</sup>, and ScMreB5<sup>ΔC10A</sup> (I95A, W96A, IWA, and ΔC10) in buffer A (300 mM KCl and 50 mM Tris, pH 8.0) shows a single peak corresponding to monomeric ScMreB5, molecular weight ~38 kD. Peaks (milli absorbance unit [mAU]) corresponding to monomeric protein are marked with asterisk (*). B inset: Calibration curve for size-exclusion chromatography for Superdex 200 using molecular weight standards. The theoretical and estimated molecular weights of ScMreB5<sup>WT</sup> monomer are mentioned. **(C)** Size-exclusion chromatography using Superdex 75 for ScMreB5 ATPase mutant ScMreB5<sup>E134A</sup> (E134A); polymerization mutant ScMreB5<sup>K57A</sup> (K57A), and ScMreB5<sup>WT</sup> (WT) in buffer A (300 mM KCl and 50 mM Tris, pH 8.0) shows a single peak corresponding to monomeric ScMreB5, molecular weight ~38 kD. Peaks corresponding to monomeric protein are marked with asterisk (*). C inset: Calibration curve for size-exclusion chromatography for Superdex 75 using molecular weight standards. The theoretical and estimated molecular weights of ScMreB5<sup>WT</sup> monomer are mentioned. **(D and E)** Intensity of light scattering measured for ScMreB5<sup>WT</sup> and ScMreB5<sup>K57A</sup> undergoing polymerization independent of nucleotide addition. Concentration of proteins ScMreB5<sup>WT</sup> and ScMreB5<sup>K57A</sup> monitored for light scattering are 35 and 5 µM, respectively. Source data are available for this figure: SourceData FS3.

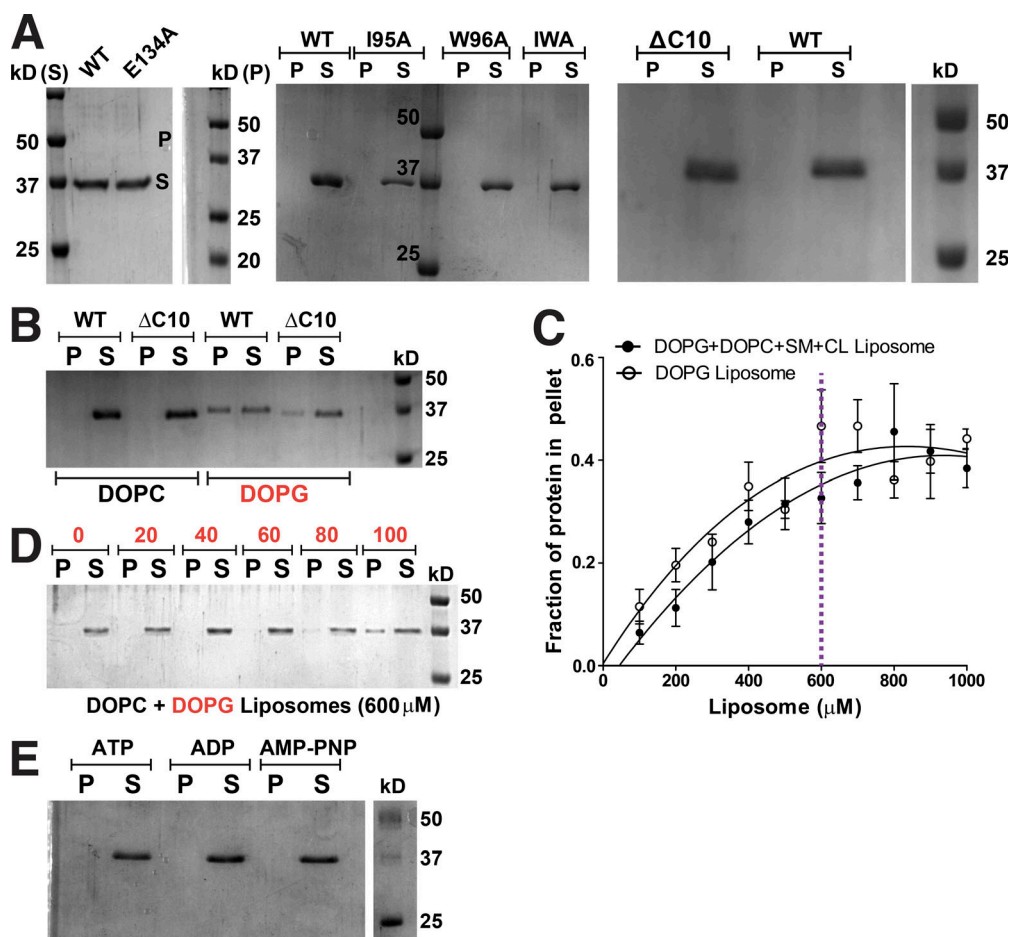

Figure S4. **ScMreB5 binding to DOPG liposome is specific and concentration dependent. (A)** A representative 12% SDS-PAGE gel of pelleting assay with WT, ATP hydrolysis mutant, and membrane-binding mutants showing that the protein does not pellet in the absence of liposomes at 100,000 $g$ spin. **(B)** A representative 12% SDS-PAGE gel of liposome pelleting assay with C-terminal deletion mutant (ΔC10) shows binding with the charged liposome, DOPG. Concentrations of DOPG and DOPC liposomes used in the assay are 1 mM each, and that of protein is 2 µM. **(C)** Liposome-binding curves showing the increase in the fraction of ScMreB5$^{WT}$ in the pellet (liposome-bound fraction) at 2-µM protein concentration, with increasing concentration of the liposomes mimicking *Spiroplasma* lipid composition and 100% DOPG liposomes. The purple dotted line marks the 600-µM liposome concentration chosen for further liposome-binding assays. **(D)** A representative 12% SDS-PAGE gel of liposome pelleting assay showing the binding specificity of ScMreB5$^{WT}$ by varying the DOPC and DOPG ratios at 600-µM liposome concentration. Protein in the pellet is observed at the higher DOPG percentages. **(E)** A representative 12% SDS-PAGE gel of pelleting assay with WT, in the presence of 1 mM ATP/ADP/AMP-PNP, showing that the protein does not pellet in the presence of nucleotides in the absence of liposomes at 100,000 $g$ spin. P and S represent the pellet and supernatant fractions of the protein. Source data are available for this figure: SourceData FS4.

Video 1. **360° volume-rendering 3D-SIM images of ScMreB5$^{WT}$ and ScMreB5$^{E134A}$ filaments.** 360° volume-rendering 3D-SIM images of ScMreB5$^{WT}$ and ScMreB5$^{E134A}$ filaments in fission yeast. 3D-SIM reconstruction of the images was carried out using SoftWorx software. 3D volume data was constructed using Fiji software. Maximum-intensity projection image is shown in Fig. 4 B. Scale bar represents 5 µm, and the frame rate is 10 frames per second (fps).

Video 2. **Polymerization of ScMreB5$^{WT}$ and ScMreB5$^{E134A}$ in *S. pombe* cells.** Time-lapse series showing the polymerization of GFP-ScMreB5$^{WT}$ and GFP-ScMreB5$^{E134A}$ in *S. pombe* cells observed by epifluorescence microscopy. Panels shown in Fig. 5 A (WT and E134A) were obtained from this time series. Images were deconvolved using SoftWorx software and are maximum-intensity projections from Z-stacks of 0.2 µm acquired at 3-min intervals. Scale bar represents 5 µm, and the frame rate is 10 fps.

Video 3.  **ATP hydrolysis mutant ScMreB5<sup>E134A</sup> shows defects in bundling of filaments compared with ScMreB5<sup>WT</sup> filaments.** Time-lapse microscopy showing bundling of GFP-ScMreB5<sup>WT</sup> filaments. GFP-ScMreB5<sup>WT</sup> filaments make lateral contacts and bundle. Septation is often seen to bring filaments together and act as a trigger for bundling as well. GFP-ScMreB5<sup>WT</sup> filaments fail to bundle. Although septation is often seen to bring filaments together, it fails to induce bundling of GFP- ScMreB5<sup>E134A</sup> filaments. Panels shown in Fig. 5 E were obtained from this time series. Images were deconvolved using SoftWorx software and are maximum-intensity projections from Z-stacks of 0.2 μm acquired at 3-min intervals. Scale bar represents 5 μm, and the frame rate is 10 fps.

Video 4.  **Disassembly of ScMreB5<sup>WT</sup> filaments.** Time-lapse series showing disassembly of GFP- ScMreB5<sup>WT</sup> filaments in *S. pombe* cells. Panels shown in Fig. 5 F were obtained from this time series. GFP-ScMreB5<sup>WT</sup> filaments are seen to disassemble or depolymerize, probably owing to dilution of protein concentration immediately upon cell division. Cells were observed by epifluorescence microscopy, and images were deconvolved using SoftWorx software. Images shown are maximum-intensity projections from Z-stacks of 0.2 μm acquired at 3-min intervals. Scale bar represents 5 μm, and the frame rate is 10 fps.

Video 5.  **Fragmentation and annealing of ScMreB5<sup>WT</sup> filaments.** Time-lapse series showing fragmentation and annealing of GFP-ScMreB5<sup>WT</sup> filaments in *S. pombe* cells. Panels shown in Fig. 5G were obtained from this time series. GFP-ScMreB5<sup>WT</sup> filaments are seen to fragment and re-anneal. Cells were observed by epifluorescence microscopy, and images were deconvolved using SoftWorx software. Images shown are maximum-intensity projections from Z-stacks of 0.2 μm acquired at 3-min intervals. Scale bar represents 5 μm, and the frame rate is 10 fps.

