## [Peer Review File · The Journal of Cell Biology]

Filament organization of the bacterial actin MreB is dependent on the nucleotide state

Vani Pande, Nivedita Mitra, Saket Bagde, Ramanujam Srinivasan, and Pananghat Gayathri

Corresponding Author(s): Pananghat Gayathri, Indian Institute of Science Education and Research Pune and Ramanujam Srinivasan, National Institute of Science Education and Research

Review Timeline:

Submission Date:	2021-06-16
Editorial Decision:	2021-07-14
Revision Received:	2021-11-01
Editorial Decision:	2021-12-16
Revision Received:	2022-01-24
Editorial Decision:	2022-02-03
Revision Received:	2022-02-09

Monitoring Editor: Kenneth Yamada

Scientific Editor: Dan Simon

Transaction Report:

DOI: <https://doi.org/10.1083/jcb.202106092>

Revision 0

Review #1

1. How much time do you estimate the authors will need to complete the suggested revisions:

Estimated time to Complete Revisions (Required)

(Decision Recommendation)

Cannot tell / Not applicable

2. Evidence, reproducibility and clarity:

Evidence, reproducibility and clarity (Required)

In this paper, Pande et al. investigate the in vitro properties of the prokaryotic actin-like protein MreB, purifying one of its counterparts (MreB5) from Spiroplasma, a clever approach given that there are major difficulties in working with MreB proteins extracted from bacterial model systems such as E. coli or B. subtilis. The paper first resolves the

structure of MreB5 and shows that it adopts the expected MreB fold. The authors then proceed in investigating the function of nucleotide-binding and in particular the role of ATP hydrolysis, a still much debated aspect of MreB function. Combining Cryo-Electron microscopy, reconstitution in fission yeast, and lipid interaction assays, the authors propose that ATP-binding in MreB polymers promotes a global MreB polymer conformational change that facilitates its binding to negatively charged lipids.

****Major comments:****

The paper makes an ambitious claim that if correct would significantly advance our understanding of MreB polymer biochemistry in cells. However, while there are many potentially interesting observations in the manuscript they are still quite indirect and some potentially artefact-prone to truly support the strong conclusions.

The main difficulty of the paper is the absence of robust biochemical assay to quantify polymer dynamics and their nucleotide dependence in vitro and correlate such dynamics to ATP hydrolysis measurements and lipid-binding. In absence of such assay protein aggregation artefacts cannot be excluded when site-directed mutants are constructed, greatly limiting the conclusions. Some examples are provided below to explain these limitations.

- In Figure 1 I, ATPase activity is measured in various mutants but given that each mutant may form distinct polymer concentrations it is really difficult to disentangle if the decrease in the ATPase activity is linked to a polymerization issue or to a hydrolysis issue or to both of them. For example, the K57A interface mutant is said to no longer polymerize (is this true?) but hydrolyze ATP. Does this mean polymerization is not required for ATP hydrolysis?

- The capacity of MreB to form filaments bundle independently of the nucleotide state is clearly presented by cryo-EM. However, this technique is not quantitative and cannot appropriately show that lower filament densities are obtained in the presence of ADP or when the E134A mutant is in presence of AMP-PNP is not ideal. Of note, the nucleotide state is not crucial for polymerization even for eukaryotic actin. It is possible that the the critical concentration for polymerization that is lowered in the presence of ATP. Could titration experiments be performed to show whether the polymer assembly is changed depending on concentration and ATP or ADP availability?

- The fission yeast system has been used in the past and provides an interesting perspective to look at polymers but there are a number of limitations with the authors interpretation. As presented, the results are mostly presented as example snapshots and lack overall quantification and

statistics. It is not currently clear that bundling is affected in the E134A mutant as compared to the WT. Surely the density, length and spatial distributions of the filaments could be quantified across yeast cells for the various MreBs. It would also be interesting to test the various mutants of this study in the yeast system, perhaps as a way to characterize their filament-forming properties in a quantitative way. Why does the yeast system not capture the expected MreB membrane interactions?

- The sedimentation assay is an interesting first attempt to look at MreB membrane interactions, but it is rather low resolution and not free of potential artefacts. This could be especially worrisome if the tested MreB mutants have slower filament forming dynamics or form aggregates which might alter their own sedimentation properties. The authors use this assay to propose their conformational change hypothesis, but again not knowing the exact polymerization properties of the mutants is problematic. For example, if the I and W mutant tend to have slower polymerization than the WT, this could explain why they are found less in the pellet in Figure 4 E-F.

- Page 12, the authors suggest "an allosteric communication between the nucleotide pocket and the membrane binding interface of MreB". This is an attractive idea but how is the membrane interaction affected with various nucleotides?

- As presented the Figure 6 model is difficult to understand and it is not presented in details in the discussion section.

****Minor comments:****

- Could the authors provide in the SI an SDS-PAGE figure of the purified protein

- Could the authors indicate what the arrows point to in Figure 3E

- Page 12, the authors mention "Figure 5 F-G". These panels are absent from the Figure.

- Page 27, the reference "12" is mentioned twice (12 and 13)

3. Significance:

Significance (Required)

****Significance:****

In recent years a number of important discoveries have been made with regards to the in vivo function of the bacterial MreB cytoskeleton, the main function of which is now thought to form a dynamic polymeric scaffold to orient cell wall complexes around the cell cylinder. However, the exact mechanisms driving MreB polymer dynamics and in particular the function of ATP hydrolysis have remained a major black box due to difficulties in developing robust biochemical approaches to study MreB in vitro. In this context, demonstrating that nucleotide binding affects the conformation of the polymers in such a way that they will tend to bind the bacterial membrane would be a major step toward understanding how MreB polymers form spontaneously against curved membrane as has been described by a number of laboratories.

While the data presented goes in this direction, it does not formally prove this mechanism which limits its current significance for people working on the bacterial cell envelope and more generally on bacterial cell biology.

Review #2

1. How much time do you estimate the authors will need to complete the suggested revisions:

Estimated time to Complete Revisions (Required)

(Decision Recommendation)

Between 1 and 3 months

2. Evidence, reproducibility and clarity:

Evidence, reproducibility and clarity (Required)

****Summary:****

MreB is a critical protein for rod shape in many bacteria species. MreB is thought to regulate cell shape through the organization of the cell wall synthesis enzymes, yet wall-less Spiroplasma depend on MreB to regulate their shape. MreB forms polymers on the cytoplasmic face of the plasma membrane and can hydrolyze ATP. Although the exact role of ATP hydrolysis is not known it has been suggested to be involved in structural reorganization of MreB monomers which could affect polymerization and membrane binding. This study uses Spiroplasma to study the role of MreB ATP hydrolysis on filament organization and membrane binding through crystallization, cryo-EM, lipid-binding assays, and mutational analysis.

****Major Comments:****

Overall the work is well-done and the authors conclusions are grounded. However, some additional controls and quantification would be useful.

1) Do the authors know that their mutant MreB proteins are properly folded?

2) Figure 3: Since the authors expressed E. coli MreB in yeast with an N-term fusion it has been shown that the observed structures in E. coli with this fusion are artifacts (Swulius and Jensen, J Bac, 2012) as well as being non-functional, which is why most researchers have switched to a functional sandwich fusion. Do the authors have any evidence that their N-terminal fusion is functional? Do the authors know that their mutant protein is being expressed in the yeast at the same levels? Perhaps a western blot or total cell fluorescence can be measured. This could explain why less cells form structures.

It would also be nice to know how many cells were observed (or percent of cells) with the phenotypes in E, F, and G. Also what is the time lag in 3D between the formation of polymers in WT and mutant expressing cells.

The authors suggest that higher concentration of E134A monomers is needed to get polymerization in yeast. This could be easily tested by overexpressing their constructs. Again quantification would be needed.

3) The authors should explain the differing results in fig 4-5 more. Quantification of Fig. 5A would be useful. It appears that W96A has a much higher concentration in the pellet fraction when mixed with 100% DOPG as opposed to fig 4. Also why do the authors believe that the difference in I95A in fig 4 is not statistically significant but is in fig 5 D is important. What about the fact that W96A was less than WT in Fig 4 but more in both fig 5?

There is no fig 5F-G

4) The authors use of language is too harsh in the discussion.

Please remove proves from the end of the first paragraph.

Please make statements that are hypothesis or models known. For ex: 'Thus, the residue MAY act as the sensor..', 'Thus, WE PROPOSE Glu140 drives the conformational switch...'

The authors state that ATP hydrolysis is essential for a conformational switch yet the crystal structures of MreB with ADP or AMP-SNP look remarkably similar and there is no crystal structure for the E134A mutant, so it seems unclear how they can make such a direct statement about the role of ATP binding.

****Minor comments:****

In the future it would be beneficial to put line numbers even if it is not required. Also either indent new paragraphs or add an extra space

Line 3: 'non-spherical bacteria' is confusion here. Do you mean rod shaped bacteria or bacteria that did not become round when MreB is lost? Please reword.

Gram should be capitalized.

Ouzounov et al. Biophys J 2016 is a missing reference for MreB and cell width

Figure 1I and table 3 do not seem to match. For example, E134A says its Kobs is .05 in the table and D70A is .02, yet in the figure D70 is clearly above E134A

Can these MreB mutants be expressed in vivo?

3. Significance:

Significance (Required)

****Significance****

MreB is found in many rod-shaped organisms. It is thought to control rod shape through the organization of cell wall synthesis enzymes. It is therefore quite interesting that MreB is found in wall-less organisms and

is essential for their shape. Spiroplasma makes an interesting model organism to study MreB assembly and membrane binding as it allows one to ignore cell wall synthesis.

A more detailed discussion on how these results mesh with results from walled organisms would be useful and help expand the interest of this study to a wider audience. For example, the authors reference the molecular dynamic simulations done by Colavin et al but do not really explain how the author's results help to interpret or modify the computational results. Or the authors mention RodZ but do not reference Morgenstein et al , or Bratton et al, which examine MreB in cells lacking RodZ and therefore MreB-wall communication. I believe there will be interest for those who study MreB or cell shape, as well as Spiroplasma

My expertise is in E coli cell shape and the roles of MreB/RodZ

Review #3

1. How much time do you estimate the authors will need to complete the suggested revisions:

Estimated time to Complete Revisions (Required)

(Decision Recommendation)

Less than 1 month

2. Evidence, reproducibility and clarity:

Evidence, reproducibility and clarity (Required)

****Summary:****

The manuscript "Filament dynamics driven by ATP hydrolysis modulates membrane binding of the bacterial actin MreB" gives several important and novel insights into the mechanism of polymerization of MreB polymerizes onto membranes. This work examine how the nucleotide state of the polymer affects this association. To gain these insights, this work uses structural biology, liposome association assays, and expression of MreB inside orthologous (eukaryotic) hosts to gain insights into the underlying mechanism of these membrane-associated polymers. While these studies were conducted on a MreB from a cell wall-less bacteria, I expect many of the observations found in this work to be applicable to other MreB filaments examined in more standard model systems

To highlight some of the (many) findings in this work:

1) This work remedies an important, long-standing deficit in the MreB field, how the nucleotide affects MreBs ability to polymerize onto membranes. Not only do they demonstrate that ScMreB5 filaments assemble independent of ATP hydrolysis, but they also show that the ability of MreB to hydrolyze ATP controls the rate of filament formation, the lateral association of filaments, and filament disassembly. This result itself that hydrolysis affects filament disassembly is a giant leap forward in our understanding of this polymer, one that will influence many future studies of MreB in many organisms.

2) This careful structural work finally nails down the role of the E134 residue in attacking the gamma phosphate on ATP, a long-standing hypothesis in the actin field, one that was not testable with eukaryotic actin. Furthermore, they very nicely show that this residue serves as an "interaction hub," connecting the nucleotide, catalytic water, and residues

from MreB sub-domains, thereby communicating the nucleotide state to the rest of the monomer, and thus affecting filament structure.

3) Unlike the MreB of *E. coli* and *C. crescentus*, the MreB of *Spiroplasma citri* charged surface on one side, and thereby are dependent on the charged nature of the membrane. Similarly, this work examines what residues help this MreB bind to the membrane, as it lacks an N-terminal amphipathic helix. Surprisingly, restudies within the "hydrophobic loop" do not appear to be involved in membrane association, an interesting point of data for other groups studying MreB inside gram-positive bacteria.

****Major comments:****

Most of the critical conclusions of the paper are very convincing and well backed by the data, but a few minor points in the discussion require some re-evaluation, as detailed below. Throughout this work, the data and methods are very clear, and this work would be easily reproducible. Likewise, the key experiments are well replicated, including using protein from multiple purifications to validate the lipid-binding assays.

There are a few statements in the text and discussion that, in light of past data, invoke questionable models and thus require rephrasing. On some points, showing a bit more data (if data is available) would help bolster their arguments.

1) The authors often state "modulating the membrane curvature." It must be noted that, thus far, MreB has not been observed to modulate membrane curvature in vivo, and this conclusion might be premature. As little is known about the: A) energetics of MreB binding to membranes, B) rigidity of MreB filaments or the membrane, and C) pressure drop across the membrane, it is not clear if, inside the cell, if 1) MreB filaments deform to the membrane, or 2) if the membrane deforms to the filaments. Notably, no freeze-fracture study of bacterial membranes in *E. coli* or *B. subtilis* has seen any local membrane deformations, so it is likely the filaments deform to the membrane.

2) In the discussion, they state - "We envisage a mechanism in which bundles of ATP-bound MreB filaments sense an optimal curvature for binding, remodel the membrane and hydrolyze ATP, and then exchange ADP with ATP and bind to the adjacent region with favorable curvature for binding, thus resulting in a processive motion."

This model is very suspect given the existing data in the field, as all experiments thus far indicate MreB processive motion is not driven by

polymer dynamics, but the activity of the associated cell wall synthesis enzymes: 1) When MreB motion is halted by antibiotics in *B. subtilis*, little to no filament polymerization dynamics are observed even at long timescales (Domínguez-Escobar et al., 2011)., and 2) GFP-MreB containing the E148A mutation moves around the cell width at the same rate as WT MreB (Garner et al., 2011).

3) In the discussion, the authors state: "Thus, ATP hydrolysis can modulate filament length and bundling, and consequently the orientation of MreB filaments on the cell membrane depending on the curvature." Given they have not yet examined if ScMreB5 is curved when bound to liposomes, much less seen that the filaments orient around rod-shaped cells (in fact, they see the opposite, as noted below), this statement appears to be highly speculative and should be rephrased or removed.

4) Similar to (3) above, the expression of ScMreB5 inside pombe cells gives an unexpected result for MreB, all the images provided suggest that ScMreB5 prefers to orient along the long axis of rods, more similar to what the Gladfelter lab observed with septins (Bridges et al., 2016), rather than what Hussain et al. observed with MreB. While this in no way impacts the findings in the paper, the authors may want to address this discrepancy in the discussion and perhaps revise any statements regarding orientation along curvature to be more cautious given their data.

5) While this paper nicely demonstrates that 1) ScMreB5 filament structure when bound to flat membranes and 2) that ScMreB5 binds to liposomes, it leaves the reader wondering if ScMreB5 filaments also are curved when bound to deformable liposomes. While not necessary, having EM images of ScMreB5 bound to liposomes would resolve this question and perhaps give further insight into the "long axis" filament alignment seen inside pombe.

****Minor comments:****

I found the text and figures incredibly clear and concise, and the methods well composed and detailed. If I may offer a few minor corrections and suggestions:

1) The authors state, "Orientation of MreB filaments within cells has been proposed to be dependent on the cell diameter," citing Hussain et al. Importantly, this was not the conclusion of that study, Hussain observed that MreB filaments orient inside rods of any width (even up to 5uM). Rather, the conclusion of that study was not that filaments orient dependent of cell diameter, but rather the "difference between principle curvatures."

2) The authors state, "Assisting in conformational changes during polymerization is an additional novel role proposed for the catalytic residue, which has always been implicated only in stimulating hydrolysis in most actin family members such as actin, ParM, and MamK." Given this paper very nicely elucidates the role of E148 in coordinating the water that attacks the gamma phosphate, it seems critical to cite the original study that proposed the equivalent residue in eukaryotic actin is responsible for ATP hydrolysis - "The structure of nonvertebrate actin: Implications for the ATP hydrolytic mechanism" by Vorobiev (Vorobiev et al., 2003)

3) It might help assist the reader in understanding the experiments in 4E-F if the authors added a small note to the legend or text that these experiments had a "pre-clearing" step (spinning the protein alone to remove any aggregates). Currently, this important detail is only mentioned in the methods.

3. Significance:

Significance (Required)

****Section B.****

While MreB has been studied in different bacteria for over 20 years, the field, thus far has lacked an understanding of how the different nucleotide states of MreB polymer affect its dynamics and membrane association. This work gives large advances in our understanding of how the associated nucleotide not only affects membrane binding but also filament dynamics. Impressively, this work approaches these problems using elegant structural studies, identifying key residues involved not only in nucleotide hydrolysis but how these residues communicate the identity of the bound nucleotide to the rest of the filament, affecting how filaments associate with the membrane. As noted above, while these studies were conducted on a MreB from a cell wall-less bacteria, I expect many of the observations found in this work to be applicable to other MreB filaments examined in more standard model systems. Generally, I expect this work to be of interest to not only the bacterial cell biology field but also researchers in the eukaryotic actin community.

My expertise lies in bacterial cell biology and biophysics. Specifically, I focus on studying bacterial polymers, examining not only their polymerization dynamics in vitro but also their in vivo motions and how these dynamics affect their associated biological function inside the cell.

****Referee Cross-commenting****

I think this is a great paper, and have no huge issues (save the need to rephrase a couple of factual errors in citing other papers).

Reviewer #1 (Evidence, reproducibility and clarity (Required)):

In this paper, Pande et al. investigate the in vitro properties of the prokaryotic actin-like protein MreB, purifying one of its counterparts (MreB5) from *Spiroplasma*, a clever approach given that there are major difficulties in working with MreB proteins extracted from bacterial model systems such as *E. coli* or *B. subtilis*. The paper first resolves the structure of MreB5 and shows that it adopts the expected MreB fold. The authors then proceed in investigating the function of nucleotide-binding and in particular the role of ATP hydrolysis, a still much debated aspect of MreB function. Combining Cryo-Electron microscopy, reconstitution in fission yeast, and lipid interaction assays, the authors propose that ATP-binding in MreB polymers promotes a global MreB polymer conformational change that facilitates its binding to negatively charged lipids.

****Major comments:****

The paper makes an ambitious claim that if correct would significantly advance our understanding of MreB polymer biochemistry in cells. However, while there are many potentially interesting observations in the manuscript they are still quite indirect and some potentially artefact-prone to truly support the strong conclusions.

The main difficulty of the paper is the absence of robust biochemical assay to quantify polymer dynamics and their nucleotide dependence in vitro and correlate such dynamics to ATP hydrolysis measurements and lipid-binding. In absence of such assay protein aggregation artefacts cannot be excluded when site-directed mutants are constructed, greatly limiting the conclusions. Some examples are provided below to explain these limitations.

Authors' response:

For all the site directed mutants, we have checked for protein aggregation by performing size exclusion chromatography. In order to look at the aggregation of all the mutants used in the assays, analytical size exclusion chromatography has been performed for all of them. All the proteins elute as a monomer, similar to the wildtype (**Response Fig. 1A, B**; to be included as Supplementary Fig S4A, B). The protein fraction in the void is negligible for all these mutants, as can be seen in the figure panels.

- In Figure 1 I, ATPase activity is measured in various mutants but given that each mutant may form distinct polymer concentrations it is really difficult to disentangle if the decrease in the ATPase activity is linked to a polymerization issue or to a hydrolysis issue or to both of them. For example, the K57A interface mutant is said to no longer polymerize (is this true?) but hydrolyze ATP. Does this mean polymerization is not required for ATP hydrolysis?

Authors' response:

We have currently not disentangled whether the mutants affect polymerization, hydrolysis or both. The hydrolysis rate is indeed expected to be a cumulative effect of polymerization and catalytic activity for most characterized filament forming proteins of the actin family.

In the case of K57A mutation, the residue does not lie at the ATP binding site, but at a probable polymerization interface (as seen from the ScMreB5 structure). Hence, the decrease in activity was hypothesized to be due to compromised polymerization.

Based on time-dependent light scattering measurements performed with the monomeric protein (fraction eluted after size exclusion chromatography), we see that the K57A mutant can potentially polymerize within a few minutes of elution, similar to the wild type (**Response Fig. 2**, to be included as Supplementary Fig. S5). However, the decreased ATPase activity might suggest a conformation of filaments that is not optimal for ATPase activity.

For all the ATPase mutants of residues at the nucleotide-binding site, our main focus was to check if hydrolysis rate was affected. Though these residues have been implicated to play a role in hydrolysis based on comparison with other actins, a mutational study with biochemical characterization of all the relevant residues included in our manuscript is not available till date for MreBs.

To clarify this further to the readers, **we have included the following sentences to the revised version of the manuscript, and include the light scattering data** for the wild type and K57A mutant as a Supplementary figure (**Response Fig. 2**, Fig. S5).

“Light scattering measurements show that ScMreB5^{K57A} might form polymers similar to the wild type (Fig. S5). The mutation of a single residue at the interface might not abrogate polymerization completely, but might lead to a sub-optimal conformation of the ATPase active site within the polymers. The abnormal interface might lead to a decrease in ATPase activity.”

[page 8, lines 9-13]

- The capacity of MreB to form filaments bundle independently of the nucleotide state is clearly presented by cryo-EM. However, this technique is not quantitative and cannot appropriately show that lower filament densities are obtained in the presence of ADP or when the E134A mutant is in presence of AMP-PNP is not ideal. Of note, the nucleotide state is not crucial for polymerization even for eukaryotic actin. It is possible that the the critical concentration for polymerization that is lowered in the presence of ATP. Could titration experiments be performed to show whether the polymer assembly is changed depending on concentration and ATP or ADP availability?

Authors' response:

We agree that cryo-EM technique is not quantitative. However, for performing titration experiments, polymerization kinetics experiments using light scattering has not been responsive to the addition of nucleotides for ScMreB5.

ScMreB5 (WT) polymerizes independent of the addition of nucleotide in vitro immediately (within 10 minutes) post elution of the monomer fraction from size exclusion chromatography (**Response Fig. 2**, to be included as Supplementary Fig. S5). Once polymerized, depolymerization of the filaments has not been observed based on light scattering. The addition of nucleotide (ADP, ATP or AMPPNP) at plateau stage also does not have an effect on the light scattering.

Dilution of the protein to very low concentrations (with an aim of working below the critical concentration of spontaneous association) did not solve the challenge, probably because of sensitivity issues at low concentrations. The scattering signal

from low concentrations of protein was at noise level and were not reliable. Since the experiment had to be performed within a few minutes of elution after size exclusion chromatography (SEC), it was difficult to ensure the exact concentration for different repeats for polymerization assay post SEC. Because of these reasons, we were not able to perform the assays to estimate critical concentrations for ScMreB5 in the presence of ATP/ADP/AMP-PNP.

Hence, we shifted to yeast expression to observe polymerization.

- The fission yeast system has been used in the past and provides an interesting perspective to look at polymers but there are a number of limitations with the authors interpretation. As presented, the results are mostly presented as example snapshots and lack overall quantification and statistics. It is not currently clear that bundling is affected in the E134A mutant as compared to the WT. Surely the density, length and spatial distributions of the filaments could be quantified across yeast cells for the various MreBs.

Authors' response:

We agree that quantification of the yeast data will bolster the evidence for bundling defects of E134A mutant.

We propose to quantify the bundling of E134A mutant using the intensity-metric method of coefficient of variation (*Higaki, T., Akita, K. & Katoh, K. Coefficient of variation as an image-intensity metric for cytoskeleton bundling. Sci Rep 10, 22187 (2020). <https://doi.org/10.1038/s41598-020-79136-x>*; demonstrated for evaluating actin bundling in images obtained by wide-field microscopy). We have already quantified a set of images (**Response Fig. 3**, to be included as panels in Supplementary Fig. S6 and/or Fig. 3) and will carry out the same procedure for other replicates.

For the spatial distribution, we have quantified the anisotropy of WT and E134A filaments using the FibrilTool (*Boudaoud, A., Burian, A., Borowska-Wykręt, D. et al. FibrilTool, an ImageJ plug-in to quantify fibrillar structures in raw microscopy images. Nat Protoc 9, 457–463 (2014). <https://doi.org/10.1038/nprot.2014.024>*), which again quantifies bundling and parallelness of filaments against the long-axis of the fission yeast cells (to be included as panels in Supplementary Fig. S6 and/or Fig. 3).

These results will be included in the revised version of the manuscript, after quantifying the data from more repeats. A representative plot with the current data has been included in **Response Fig. 3**.

It would also be interesting to test the various mutants of this study in the yeast system, perhaps as a way to characterize their filament-forming properties in a quantitative way.

Authors' response:

We do plan to test the yeast system for characterization of the various mutants. However, we would like to maintain the focus of this manuscript on the E134A mutation, for which we have data for a complete mechanism because of a structural explanation, biochemical data on ATPase activity and liposome binding experiments.

Why does the yeast system not capture the expected MreB membrane interactions?

Authors' response:

The main objective of the yeast expression in this study was to capture the polymerization dynamics, which was technically challenging to do in vitro due to reasons mentioned above.

We (Srinivasan, et al, 2007) and others (Karczmarek A, et al, 2007) have earlier found that the N-terminal GFP fusion in *E. coli* MreB (EcMreB) abolishes membrane binding and induces bundling, likely because the N-terminal amphipathic helix of EcMreB faces the same side of the membrane binding loop. An EcMreB sandwich construct localises to the membranes in yeast (Srinivasan, unpublished results). We think the same might be true for ScMreB5 as well since both the N-terminus and C-terminus of ScMreB5, and the proposed hydrophobic loop all face the membrane-binding surface. Currently, we wanted a system to observe the filament dynamics independent of membrane binding, and hence utilised the N-terminal GFP fusion strain.

Now, we have generated a ScMreB sandwich construct, similar to the one constructed for *E. coli* MreB and plan to express the same in fission yeast and test its localisation and assembly properties.

Results pertaining to these experiments will be included in the revised version.

We have added the sentence below in the Results section of the revised submission of the manuscript to clarify that the N-terminal-GFP fusion does not bind the membrane.

“The N-terminal GFP-fusion probably prevented the ScMreB5^{WT} filaments from binding to the membrane, and adopting any orientational preference according to the membrane curvature.”

[page 9, lines 9-11]

- The sedimentation assay is an interesting first attempt to look at MreB membrane interactions, but it is rather low resolution and not free of potential artefacts. This could be especially worrisome if the tested MreB mutants have slower filament forming dynamics or form aggregates which might alter their own sedimentation properties.

Authors' response:

ScMreB5 filaments do not sediment on their own upon polymerization (ADP, ATP or AMP-PNP addition) except in the presence of liposomes (**Response Fig. 4**, to be included as Supplementary Fig. S7).

The aggregation of wildtype and the mutants does not occur at the given concentration of the protein. The respective control runs (without the liposomes) have been performed for the all the nucleotide states for wild type and mutants (representative data in **Response Fig. 4**). Moreover, prior to the addition of liposomes, the protein samples were spun at 22,000 xg to ensure that any protein aggregates were removed (also see Reviewer 3, point no. 3). Hence, we conclude

that the observation of protein in the pellet fraction is indeed driven by liposome-protein interactions.

We have included this detail in the revised submission of the manuscript in the results section also. To quote:

“Pelletting assays of the reaction mix without liposomes also served as negative controls for the liposome-binding experiments (Fig. S7). Prior to the addition of liposomes, the protein samples were spun at 22,000 xg to ensure that any protein aggregates were removed.”

[page 10, lines 20 - 23]

We also demonstrate that the pelleting effect is dependent on the liposome composition and concentration (no protein in pellet fraction in the presence of 100 % DOPC liposomes; Fig. 5A), which can be fitted to a binding curve (**Supplementary Fig. S4** in original manuscript). This experiment also supports that the observed pelleting is a feature of lipid-protein interaction, and not due to the aggregation of the protein alone.

- The authors use this assay to propose their conformational change hypothesis, but again not knowing the exact polymerization properties of the mutants is problematic. For example, if the I and W mutant tend to have slower polymerization than the WT, this could explain why they are found less in the pellet in Figure 4 E-F.

Authors' response:

ScMreB5 filaments do not sediment on their own upon polymerization (ADP, ATP or AMP-PNP addition) except in the presence of liposomes (**Response Fig. 4**, to be included as Supplementary Fig. S7).

Nonetheless, the lower binding of mutants could be potentially due to any of the following three reasons:

- a) The mutation of a residue on the membrane-binding interface that directly interacts with the liposome will affect binding.
- b) MreB upon polymerisation binds to membranes and thus lower levels of polymerization could lead to less binding to the liposomes.
- c) Geometry of the filaments and their higher organization such as bundles could affect the liposome binding in response to the curvature.

For I95, W96 and Δ C10 mutants, since the mutations are present at the membrane binding interface, possibly, the direct binding interface (point (a) above) might be affected. A decrease in polymer content for these constructs could imply an allosteric effect of membrane binding on polymerization or vice versa because the membrane-binding interface is away from the polymerization interface.

For the E134A mutant, the decrease in binding could be due to the reduced polymerization and bundling leading to reduced binding to the liposomes.

- Page 12, the authors suggest "an allosteric communication between the nucleotide pocket and the membrane binding interface of MreB". This is an attractive idea but how is the membrane interaction affected with various nucleotides?

Authors' response:

In the original manuscript, the idea was proposed as a hypothesis based on the observations from liposome binding properties of the E134A mutant. This prompted us to carry out liposome binding assays in the presence of various nucleotides namely ADP, ATP and AMP-PNP. Our preliminary results show us that there is a differential binding for the wild type protein too based on the nucleotide state (**Response Fig. 5**).

We see a decrease in binding for wild type ScMreB5 upon addition of ADP and ATP, similar to that observed for the E134A mutant. The decreased binding could be because of the reduced bundling in ADP and ATP filaments compared to that of AMP-PNP (EM filament images in Fig. 2 of the manuscript) . Thus, we hypothesize that the filament geometry in the ADP state is not compatible with binding to the liposomes of the curvature used in our assays.

Based on the differential pelleting observed in the ATP, ADP and AMP-PNP states and without nucleotide addition (**Response Fig. 5**), we confirm that the membrane interaction is indeed dependent on the nucleotide state. The different nucleotide states might possess different conformations or different capabilities to form bundles, or bind efficiently to different curvatures, thus contributing to the sensing of membrane curvature by the MreB filaments. It is possible that in the absence of additional nucleotide, the filaments are capable of bending to the liposome curvature, ADP-bound filaments can bind efficiently only to liposomes that match their curvature, while the bundles of filaments formed in the presence of AMP-PNP can remodel the shape of the liposomes upon binding. Indeed, data from electron cryotomography of liposomes in the presence of AMP-PNP (Salje, et al, 2011) and excess of ATP (Hussain, et al, 2018; note that regions on the liposome with bundles of filaments are remodelled while the areas with single filaments bound retain the spherical shape) seem to be consistent with this hypothesis.

Further experiments with liposome of different radii and the effect of mutations and nucleotide states are currently under progress. We plan to include this data as part of a future manuscript on the mechanistic basis of curvature sensing by MreB.

- As presented the Figure 6 model is difficult to understand and it is not presented in details in the discussion section.

Authors' response:

This has been rephrased in the revised submission of the manuscript to clarify the point further by including references to the relevant figure panels. To quote:

“The entire network of interactions (labeled as i – vii in Fig. 6, inset) with all the 4 subdomains was observed only in the double filament conformation (PDB ID 4CZJ; Fig. 6) and not in the single protofilament or monomeric states (PDB IDs 4CZI, 4CZF or 4CZM in Fig. 6).”

[page 13, lines 10-13]

****Minor comments:****

- Could the authors provide in the SI an SDS-PAGE figure of the purified protein

Authors' response:

SDS-PAGE profiles of all purified proteins used in this study are shown in **Response Fig. 1C** and will be included in the Supplementary data of the revised manuscript.

- Could the authors indicate what the arrows point to in Figure 3E.

Authors' response:

This information has now been included in the figure legend of the revised submission of the manuscript.

“White arrows highlight the bundling events, which happen only at the site of cell division for ScMreB5^{E134A}.”

[page 37, lines 14-15]

- Page 12, the authors mention "Figure 5 F-G". These panels are absent from the Figure.

Authors' response:

The error has been corrected in the revised submission of the manuscript. Reference to Figure 5F-G in the text has been replaced Figure 5C-D.

“Interestingly, liposome binding assay of ScMreB5^{E134A} showed a significant decrease compared to that of ScMreB5^{WT} (Fig. 5C, D),”

[page 12, lines 10-12]

- Page 27, the reference "12" is mentioned twice (12 and 13)

Authors' response:

The error has been corrected in the reference list of the revised submission of the manuscript. *[pages 27-31]*

Reviewer #1 (Significance (Required)):

****Significance:****

In recent years a number of important discoveries have been made with regards to the in vivo function of the bacterial MreB cytoskeleton, the main function of which is now thought to form a dynamic polymeric scaffold to orient cell wall complexes around the cell cylinder. However, the exact mechanisms driving MreB polymer dynamics and in particular the function of ATP hydrolysis have remained a major black box due to difficulties in developing robust biochemical approaches to study MreB in vitro. In this context, demonstrating that nucleotide binding affects the conformation of the polymers in such a way that they will tend to bind the bacterial membrane would be a major step toward understanding how MreB polymers form spontaneously against curved membrane as has been described by a number of laboratories.

While the data presented goes in this direction, it does not formally prove this mechanism which limits its current significance for people working on the bacterial cell envelope and more generally on bacterial cell biology.

Authors' response:

We thank the reviewer for providing inputs to improve the manuscript. We hope the suggested modifications to the manuscript help in addressing the concerns raised and provides support for the mechanism.

Reviewer #2 (Evidence, reproducibility and clarity (Required)):

****Summary:****

MreB is a critical protein for rod shape in many bacteria species. MreB is thought to regulate cell shape through the organization of the cell wall synthesis enzymes, yet wall-less Spiroplasma depend on MreB to regulate their shape. MreB forms polymers on the cytoplasmic face of the plasma membrane and can hydrolyze ATP. Although the exact role of ATP hydrolysis is not known it has been suggested to be involved in structural reorganization of MreB monomers which could affect polymerization and membrane binding. This study uses Spiroplasma to study the role of MreB ATP hydrolysis on filament organization and membrane binding through crystallization, cryo-EM, lipid-binding assays, and mutational analysis.

****Major Comments:****

Overall the work is well-done and the authors conclusions are grounded. However, some additional controls and quantification would be useful.

1) Do the authors know that their mutant MreB proteins are properly folded?

Authors' response:

Analytical size exclusion chromatography runs for the mutants have been performed which shows that all the mutants elute as monomer (**Response Fig. 1A and 1B**), indicating they are well folded.

2) Figure 3: Since the authors expressed *E. coli* MreB in yeast with an N-term fusion it has been shown that the observed structures in *E. coli* with this fusion are artifacts (Swulius and Jensen, J Bac, 2012) as well as being non-functional, which is why most researchers have switched to a functional sandwich fusion. Do the authors have any evidence that their N-terminal fusion is functional?

Authors' response:

The N-terminal fusion in *E. coli* MreB is likely non-functional due to the loss of the essential membrane binding activity, because the N-terminal amphipathic helix of *E. coli* MreB faces the same side of the membrane binding loop. Nonetheless, the N-terminal fusion in *E. coli* MreB is fully functional for polymerisation and ATP-binding (polymerisation is inhibited by A22) (Srinivasan, 2007). Further, our unpublished results using the the functional sandwich fusion of EcMreB in yeast show that it localises efficiently to membranes and assembles into short polymers or patches on membranes.

Thus, in the heterologous yeast expression studies, while the functional sandwich construct captures membrane binding of MreB effectively, the N-terminal fusion turns out to be a better indicator for bundling. Therefore, we resorted to the use of the N-terminal GFP fusion to ScMreB for our expression studies in fission yeast and assess bundling.

Based on our yeast imaging experiments, the N-terminal fusion construct appears to be functional for polymerization, and depolymerization too. Comparison with the E134A mutant suggests that the WT protein can undergo depolymerization while the E134A (ATPase deficient mutant) cannot. This suggests that the protein might be a functional ATPase.

To ascertain the activity and functionality of N-terminal GFP fusion in ScMreB, we plan to purify the N-terminal GFP tagged ScMreB and test its polymerisation and ATPase activity *in vitro* as well.

Results pertaining to these will be included in the revised manuscript.

Do the authors know that their mutant protein is being expressed in the yeast at the same levels? Perhaps a western blot or total cell fluorescence can be measured. This could explain why less cells form structures.

Authors' response:

This is a very valid point that could potentially alter our interpretations. A preliminary Western blot analysis using the same OD of cells indicated that the expression levels of WT ScMreB and E134A mutant were similar. However, we plan to perform Western blots from multiple cultures and quantitate the levels as a fraction of total cell lysate loaded and/ or include loading controls and provide statistical parameters. This should address the issues of differential expression of the two proteins.

The data from the Western blots and the quantification of the protein levels will be included in the revised manuscript.

It would also be nice to know how many cells were observed (or percent of cells) with the phenotypes in E, F, and G. Also what is the time lag in 3D between the formation of polymers in WT and mutant expressing cells.

Authors' response:

We have counted the cells in which we observed the phenotypes reported in E, F and G and will include this information in the revised manuscript. For the time lag between the assembly of polymers in WT and mutant, we plan to perform a time-course experiment and represent the percentage of cells having polymers as a function of time after induction of expression. We believe this would be more appropriate than measuring time difference from time-lapse images, wherein the zero time point is ill defined.

This information and the results from new experiments will be included in the revised manuscript.

The authors suggest that higher concentration of E134A monomers is needed to get polymerization in yeast. This could be easily tested by overexpressing their constructs. Again quantification would be needed.

Authors' response:

We agree that this will again bolster our claim on the critical concentration of polymerisation being elevated by the E134A mutation. We will express the mutant from a stronger version of the same promoter (*nmt1*) and make a quantitative comparison and provide statistical measure of the number of cells with polymers between the medium strength promoter (*nmt41*) currently used and the stronger promoter (*nmt1*).

This information and the results from new experiments will be included in the revised manuscript.

3) The authors should explain the differing results in fig 4-5 more. Quantification of Fig. 5A would be useful. It appears that W96A has a much higher concentration in the pellet fraction when mixed with 100% DOPG as opposed to fig 4. Also why do the authors believe that the difference in I95A in fig 4 is not statistically significant but is in fig 5 D is important. What about the fact that W96A was less than WT in Fig 4 but more in both fig 5?

Authors' response:

Fig 4 (panels E and F) shows the binding of wildtype and the mutants with the liposomes that mimic the whole cell membrane composition of *Spiroplasma citri*. Here, the binding of the wild type and the membrane binding mutants appear to be similar. However, we had used a liposome concentration of 1 mM in these experiments, and we wanted to ensure that we were not comparing the binding curves at the saturated phase. Hence, we shifted to carrying out liposome binding assays at a constant concentration of liposomes, but varying the content of DOPG, which is a better comparison of the composition effect without changing the liposome amounts.

For Fig 5, to decipher the exact mechanism of how MreB5 is binding to the membrane, we compared the binding with the negatively charged lipid (DOPG, 38% in *Spiroplasma* lipid composition) and a neutral lipid (DOPC, 14% in *Spiroplasma* lipid composition). DOPG is present in the highest composition whereas DOPC in the lowest percentage in the *Spiroplasma* composition. Thus, we took the two extreme lipids in terms of charge and percentage to see the binding effect.

Quantification of Fig. 5A has been provided in the Supplementary figure Fig. S4A of the original manuscript.

The difference in liposome binding of W96A between Fig 4 (E and F) and Fig. 5A arises because both these experiments are different in terms of the liposome composition. A similar effect has been observed for I95A mutant as well (Fig 4 and 5D). In the Fig. 5D experiment, the composition of liposome (80% DOPG and 20% DOPC) brought out the difference in terms of liposome binding, which was earlier masked when 100% DOPG and/or *Spiroplasma* composition was used.

The W96A pellet fraction in Fig. 4 has higher error when compared to Fig 5, hence, statistically, binding for W96A and wildtype cannot be claimed to be different.

There is no fig 5F-G

Authors' response:

Reference to Fig 5F-G has been corrected as Fig. 5C,D in the revised submission of the manuscript.

4) The authors use of language is too harsh in the discussion. Please remove proves from the end of the first paragraph. Please make statements that are hypothesis or models known. For ex: 'Thus, the residue MAY act as the sensor..', 'Thus, WE PROPOSE Glu140 drives the conformational switch...'

Authors' response:

These sentences have been rephrased accordingly in the revised submission of the manuscript.

“Thus, the residue might act as the sensor for the ATP-bound state and trigger the double protofilament conformation”

[page 13, lines 14-15]

“Thus, we propose that Glu140 drives the conformational switch between the ATP-bound double protofilament state and the ADP state incompatible with the double protofilament conformation.”

[page 13, lines 19-20]

The authors state that ATP hydrolysis is essential for a conformational switch yet the crystal structures of MreB with ADP or AMP-PNP look remarkably similar and there is no crystal structure for the E134A mutant, so it seems unclear how they can make such a direct statement about the role of ATP binding.

Authors' response:

The crystal structures for ScMreB5 complexed with ADP and AMPPNP, respectively, are both single protofilament states, and were not captured in the double protofilament conformation. Hence, the structural basis has been explained by carefully analysing the CcMreB structures, the only MreB where double protofilament, single protofilament and monomeric conformations have been captured.

A crystal structure of E134A may not be as informative because the side chain will be missing. It can be explained best only based on a structure with intact glutamate side chain and bound ATP.

****Minor comments:****

In the future it would be beneficial to put line numbers even if it is not required. Also either indent new paragraphs or add an extra space

Authors' response:

We have made these changes in the revised submission of the manuscript.

Line 3: 'non-spherical bacteria' is confusion here. Do you mean rod shaped bacteria or bacteria that did not become round when MreB is lost? Please reword.

Authors' response:

We have deleted the words in the revised submission of the manuscript. Please refer *page 3, line 4*.

Gram should be capitalized.

Authors' response:

The required changes have been made in the revised submission of the manuscript. Please refer *page 3, lines 16, 17*.

Ouzounov et al. Biophys J 2016 is a missing reference for MreB and cell width

Authors' response:

This reference has been cited appropriately in the revised submission of the manuscript. To quote the first instance of citing:

“Mutations in MreB can result in cells of varying width (25).”

[page 5, lines 3 - 4]

Figure 1I and table 3 do not seem to match. For example, E134A says its Kobs is .05 in the table and D70A is .02, yet in the figure D70 is clearly above E134A

Authors' response:

The change will be made in the updated table S3 in the revised submission of the manuscript. We thank the reviewer for pointing out this error.

Can these MreB mutants be expressed in vivo?

Authors' response:

We are in the process of expressing these mutants in *Spiroplasma* ASP-I (a strain deficient in MreB5, refer our previous work Harne, et al, 2020, Current Biology).

However, this work will not be part of the current manuscript because it will take about a year or more to generate the mutant strains, complete the experiments, and characterize the mutants to obtain mechanistic insights. We look forward to these results and plan to publish this as a separate manuscript in future.

Reviewer #2 (Significance (Required)):

****Significance****

MreB is found in many rod-shaped organisms. It is thought to control rod shape

through the organization of cell wall synthesis enzymes. It is therefore quite interesting that MreB is found in wall-less organisms and is essential for their shape. *Spiroplasma* makes an interesting model organism to study MreB assembly and membrane binding as it allows one to ignore cell wall synthesis.

A more detailed discussion on how these results mesh with results from walled organisms would be useful and help expand the interest of this study to a wider audience. For example, the authors reference the molecular dynamic simulations done by Colavin et al but do not really explain how the author's results help to interpret or modify the computational results.

Authors' response:

We have rewritten the text in the discussion to make the interpretation more clear in the context of our results, in the revised submission of the manuscript. To quote:

“While there are theoretical models on how this might be achieved (36), our study based on ScMreB5^{E134A} provides insights into the role of ATP-driven dynamics in polymerization and membrane binding of MreB. A hypothesis on how different nucleotide states could exhibit different modes of membrane-binding by MreB was earlier put forward based on molecular dynamics simulations (37). Impairment of membrane binding by ScMreB5^{E134A} indicates that the conformational change facilitated by Glu134 on sensing gamma phosphate of ATP might be required for efficient liposome interaction, possibly mediated by the filaments bundling through lateral interactions.”

[page 14, lines 3 - 11]

Or the authors mention RodZ but do not reference Morgenstein et al , or Bratton et al, which examine MreB in cells lacking RodZ and therefore MreB-wall communication. I believe there will be interest for those who study MreB or cell shape, as well as *Spiroplasma*

Authors' response:

We have rewritten the text in the discussion to make the interpretation more clear in the context of our results, in the revised submission of the manuscript. To quote:

“In the absence of membrane attachments facilitated by the peptidoglycan synthesis related proteins such as RodZ (38), a surface extensive interaction with the membrane might help in orienting the filaments in a cell-wall-less organism. Interestingly, RodZ plays an important role in circumferential movement of MreB by linking with the peptidoglycan synthesis machinery, and also in curvature dependent localization of MreB (39, 40). In the absence of RodZ and peptidoglycan synthesis in *Spiroplasma*, a novel mode of membrane binding involving an increased surface might be important.”

[page 15, lines 8 - 12]

We thank the reviewer for the suggestions for improvement. We will include these suggestions and rewrite the discussion accordingly in the revised manuscript. We hope that the revisions will address the concerns of the reviewer.

My expertise is in E coli cell shape and the roles of MreB/RodZ

Reviewer #3 (Evidence, reproducibility and clarity (Required)):

****Summary:****

The manuscript "Filament dynamics driven by ATP hydrolysis modulates membrane binding of the bacterial actin MreB" gives several important and novel insights into the mechanism of polymerization of MreB polymerizes onto membranes. This work examine how the nucleotide state of the polymer affects this association. To gain these insights, this work uses structural biology, liposome association assays, and expression of MreB inside orthologous (eukaryotic) hosts to gain insights into the underlying mechanism of these membrane-associated polymers. While these studies were conducted on a MreB from a cell wall-less bacteria, I expect many of the observations found in this work to be applicable to other MreB filaments examined in more standard model systems.

To highlight some of the (many) findings in this work:

- 1) This work remedies an important, long-standing deficit in the MreB field, how the nucleotide affects MreBs ability to polymerize onto membranes. Not only do they demonstrate that ScMreB5 filaments assemble independent of ATP hydrolysis, but they also show that the ability of MreB to hydrolyze ATP controls the rate of filament formation, the lateral association of filaments, and filament disassembly. This result itself that hydrolysis affects filament disassembly is a giant leap forward in our understanding of this polymer, one that will influence many future studies of MreB in many organisms.
- 2) This careful structural work finally nails down the role of the E134 residue in attacking the gamma phosphate on ATP, a long-standing hypothesis in the actin field, one that was not testable with eukaryotic actin. Furthermore, they very nicely show that this residue serves as an "interaction hub," connecting the nucleotide, catalytic water, and residues from MreB sub-domains, thereby communicating the nucleotide state to the rest of the monomer, and thus affecting filament structure.
- 3) Unlike the MreB of *E. coli* and *C. crescentus*, the MreB of *Spiroplasma citri* charged surface on one side, and thereby are dependent on the charged nature of the membrane. Similarly, this work examines what residues help this MreB bind to the membrane, as it lacks an N-terminal amphipathic helix. Surprisingly, residues within the "hydrophobic loop" do not appear to be involved in membrane association, an interesting point of data for other groups studying MreB inside gram-positive bacteria.

****Major comments:****

Most of the critical conclusions of the paper are very convincing and well backed by the data, but a few minor points in the discussion require some re-evaluation, as detailed below. Throughout this work, the data and methods are very clear, and this work would be easily reproducible. Likewise, the key experiments are well replicated, including using protein from multiple purifications to validate the lipid-binding assays. There are a few statements in the text and discussion that, in light of past data, invoke questionable models and thus require rephrasing. On some points, showing a bit more data (if data is available) would help bolster their arguments.

- 1) The authors often state "modulating the membrane curvature." It must be noted

that, thus far, MreB has not been observed to modulate membrane curvature in vivo, and this conclusion might be premature. As little is known about the: A) energetics of MreB binding to membranes, B) rigidity of MreB filaments or the membrane, and C) pressure drop across the membrane, it is not clear if, inside the cell, if 1) MreB filaments deform to the membrane, or 2) if the membrane deforms to the filaments. Notably, no freeze-fracture study of bacterial membranes in *E. coli* or *B. subtilis* has seen any local membrane deformations, so it is likely the filaments deform to the membrane.

Authors' response:

As the reviewer rightly points out, while there are instances of MreB changing the membrane curvature of deformable liposomes (Hussain, et al, 2018; Salje, et al, 2011), there is no evidence of MreB modulating the membrane curvature in vivo. Hence, we have rephrased the relevant sentences in the revised submission of the manuscript, and mentioned about localising to a definite membrane curvature within the cell, and modulating of membrane curvature only in the context of liposomes.

2) In the discussion, they state - "We envisage a mechanism in which bundles of ATP-bound MreB filaments sense an optimal curvature for binding, remodel the membrane and hydrolyze ATP, and then exchange ADP with ATP and bind to the adjacent region with favorable curvature for binding, thus resulting in a processive motion."

This model is very suspect given the existing data in the field, as all experiments thus far indicate MreB processive motion is not driven by polymer dynamics, but the activity of the associated cell wall synthesis enzymes: 1) When MreB motion is halted by antibiotics in *B. subtilis*, little to no filament polymerization dynamics are observed even at long timescales (Domínguez-Escobar et al., 2011)., and 2) GFP-MreB containing the E148A mutation moves around the cell width at the same rate as WT MreB (Garner et al., 2011).

Authors' response:

Though the role of polymer dynamics of MreB in processive motion has not been established, the experiments carried out with ATPase mutants (E134 equivalent) in MreBs of *Bacillus*, *E. coli* and *Caulobacter* indeed show that there are shape defects. Though the rate of processing movement remains the same, the localization of MreB filaments is defective, leading to overall shape defects.

This has been elaborated in the revised submission of the manuscript, as follows:

"Though the speed of processive movement of MreB filaments has been demonstrated to be independent of ATP hydrolysis (9), ATPase mutants of MreB display localization defects in vivo, and have shape defects as demonstrated for *Bacillus subtilis* (22) and *Caulobacter crescentus* (20). We envisage a mechanism in which bundles of ATP-bound MreB filaments sense an optimal curvature for binding, recruit peptidoglycan machinery to remodel the membrane, possibly hydrolyse ATP during the process, exchange ADP with ATP and bind to the adjacent region with favourable curvature for binding, thus resulting in a processive motion."

[page 15, line 21 onwards]

3) In the discussion, the authors state: "Thus, ATP hydrolysis can modulate filament length and bundling, and consequently the orientation of MreB filaments on the cell membrane depending on the curvature." Given they have not yet examined if ScMreB5 is curved when bound to liposomes, much less seen that the filaments orient around rod-shaped cells (in fact, they see the opposite, as noted below), this statement appears to be highly speculative and should be rephrased or removed.

Authors' response:

We agree that this is highly speculative based on the existing data in the manuscript. Hence, we have deleted this sentence from the abstract in the revised submission.

[page 2, line 16]

4) Similar to (3) above, the expression of ScMreB5 inside pombe cells gives an unexpected result for MreB, all the images provided suggest that ScMreB5 prefers to orient along the long axis of rods, more similar to what the Gladfelter lab observed with septins (Bridges et al., 2016), rather than what Hussain et al. observed with MreB. While this in no way impacts the findings in the paper, the authors may want to address this discrepancy in the discussion and perhaps revise any statements regarding orientation along curvature to be more cautious given their data.

Authors' response:

The construct used in the yeast experiments does not reflect any of the properties related to orientation along the axis, probably because of the loss of membrane localisation, given that the GFP is positioned at the potential membrane-binding interface.

Now, we have generated a ScMreB sandwich construct, similar to the one constructed for *E. coli* MreB and plan to express the same in fission yeast and test its localisation and assembly properties.

Our proposed experiments with the sandwich construct might give us insights on orientation, if the MreB binding is compatible with the curvature and membrane lipid composition of fission yeast.

These points will be addressed in the discussion in the revised manuscript, after recording the observations of the proposed sandwich construct experiments.

5) While this paper nicely demonstrates that 1) ScMreB5 filament structure when bound to flat membranes and 2) that ScMreB5 binds to liposomes, it leaves the reader wondering if ScMreB5 filaments also are curved when bound to deformable liposomes. While not necessary, having EM images of ScMreB5 bound to liposomes would resolve this question and perhaps give further insight into the "long axis" filament alignment seen inside pombe.

Authors' response:

As mentioned above, the long axis filament alignment might be the result of the absence of membrane binding of the N-terminal GFP construct. Nonetheless, our

proposed experiments with the sandwich construct might give us insights on orientation, if the MreB binding is compatible with the curvature and liposome composition of fission yeast to an extent. Further, as in response to Point 3, we have now deleted the speculative statement pertaining to MreB filament orientation and membrane curvature.

We have not performed EM experiments with liposomes with the ScMreB5. Unfortunately, we will not be able to include these experiments in the current manuscript because of restricted access to the electron microscope, especially due to the current pandemic situation when access to national facilities will be available only after an indefinite period. We propose to carry out these experiments in future and include it in a follow-up study.

****Minor comments:****

I found the text and figures incredibly clear and concise, and the methods well composed and detailed. If I may offer a few minor corrections and suggestions:

1) The authors state, "Orientation of MreB filaments within cells has been proposed to be dependent on the cell diameter," citing Hussain et al. Importantly, this was not the conclusion of that study, Hussain observed that MreB filaments orient inside rods of any width (even up to 5µM). Rather, the conclusion of that study was not that filaments orient dependent of cell diameter, but rather the "difference between principle curvatures."

Authors' response:

We have rephrased the relevant sentence in the revised submission of the manuscript, as follows:

"Orientation of MreB filaments within cells has been proposed to be dependent on the differences between the principle curvatures, with a more ordered arrangement when the difference is higher as in a narrow rod (8)."

[page 5, lines 2 -4]

2) The authors state, "Assisting in conformational changes during polymerization is an additional novel role proposed for the catalytic residue, which has always been implicated only in stimulating hydrolysis in most actin family members such as actin, ParM, and MamK." Given this paper very nicely elucidates the role of E148 in coordinating the water that attacks the gamma phosphate, it seems critical to cite the original study that proposed the equivalent residue in eukaryotic actin is responsible for ATP hydrolysis - "The structure of nonvertebrate actin: Implications for the ATP hydrolytic mechanism" by Vorobiev (Vorobiev et al., 2003)

Authors' response:

We had already cited the relevant reference in the original manuscript in the Results section while mentioning the role of Glu134 equivalent residues.

"However, Glu134 and/or Thr161 might interact with the catalytic water, a hypothesis based on structure alignments with other MreB and actin structures in their double protofilament conformations (Fig. 1H; Fig. S3 A-B) (10, 14, 30).

[page 7, lines 10 – 13]

We have included this reference in the Introduction section too in the revised submission of the manuscript (ref. 14).

“Assisting in conformational changes during polymerization is an additional novel role proposed for the catalytic residue, which has always been implicated only in stimulating hydrolysis in most actin family members such as actin (13, 14), ParM (15) and MamK (28).”

[page 5, lines 13 – 16]

3) It might help assist the reader in understanding the experiments in 4E-F if the authors added a small note to the legend or text that these experiments had a "pre-clearing" step (spinning the protein alone to remove any aggregates). Currently, this important detail is only mentioned in the methods.

Authors' response:

We have included this detail in the revised submission of the manuscript in the results section also. To quote:

“Pelletting assays of the reaction mix without liposomes also served as negative controls for the liposome-binding experiments (Fig. S7). Prior to the addition of liposomes, the protein samples were spun at 22,000 xg to ensure that any protein aggregates were removed.”

[page 10, lines 20 - 23]

Reviewer #3 (Significance (Required)):

****Section B.****

While MreB has been studied in different bacteria for over 20 years, the field, thus far has lacked an understanding of how the different nucleotide states of MreB polymer affect its dynamics and membrane association. This work gives large advances in our understanding of how the associated nucleotide not only affects membrane binding but also filament dynamics. Impressively, this work approaches these problems using elegant structural studies, identifying key residues involved not only in nucleotide hydrolysis but how these residues communicate the identity of the bound nucleotide to the rest of the filament, affecting how filaments associate with the membrane. As noted above, while these studies were conducted on a MreB from a cell wall-less bacteria, I expect many of the observations found in this work to be applicable to other MreB filaments examined in more standard model systems. Generally, I expect this work to be of interest to not only the bacterial cell biology field but also researchers in the eukaryotic actin community.

My expertise lies in bacterial cell biology and biophysics. Specifically, I focus on studying bacterial polymers, examining not only their polymerization dynamics in vitro but also their in vivo motions and how these dynamics affect their associated biological function inside the cell.

****Referee Cross-commenting****

I think this is a great paper, and have no huge issues (save the need to rephrase a couple of factual errors in citing other papers).

Authors' response:

We thank the reviewer for the suggestions for improvement. We will include these suggestions and rewrite the discussion accordingly in the revised manuscript. We hope that the proposed plan for revision will address the concerns raised by the reviewers.

A**B****C**
Response Fig. 1. ScMreB5 mutants are monomeric and well folded

- A. Analytical size exclusion chromatography using Superdex 200 for ScMreB5 (WT) , ATPase mutants (D12A, D70A, D156A, T161A) and membrane binding mutants (I95A, W96A, IWA, Δ C10) in Buffer A (300 mM KCl, 50 mM Tris, pH= 8.0) shows a single peak corresponding to monomeric ScMreB5 molecular weight of approximately 38 kDa.
- B. Size exclusion chromatography using Superdex-75 for ScMreB5 ATPase mutant (E134A), polymerization mutant (K57A) and wild type in Buffer A (300 mM KCl, 50 mM Tris, pH= 8.0) shows a single peak corresponding to monomeric ScMreB5 molecular weight of approximately 38 kDa.
- C. Representative gels of SDS-PAGE profile of purified protein samples of ScMreB5 and the mutant constructs.

This will be included as Supplementary Figure Fig. S4 in the revised manuscript.

Response Fig. 2. ScMreB5 WT and mutants undergo polymerization independent of nucleotide addition, as monitored through right-angle light scattering

A. Polymerization of ScMreB5^{WT} at 35 μ M concentration.

B. Polymerization of ScMreB5^{K57A} at 5.1 μ M concentration.

All these readings were taken at right angle light scattering measured using a fluorimeter at 400 nm excitation and 400 emission wavelength, immediately after elution from size exclusion chromatography.

This will be included as Supplementary Figure Fig. S5 in the revised manuscript.

Response Figure 3. Quantification of localization features of MreB5 filaments in fission yeast.

- A. Anisotropy values measured using the FibrilTool for the ScMreB5 (WT) and the ATPase mutant (E134A) shows that the E134A mutant is less bundled and poorly oriented along the long axis of the yeast cells. The open circle indicates the mean and the black bar represents the 95 % CI. ScMreB5^{WT} – 0.63 ± 0.02 95% CI, (n=112) and ScMreB5^{E134A} – 0.50 ± 0.03 95% CI, (n=32).
- B. Coefficient of Variation of Intensity (CV) measured using FIJI shows that WT ScMreB5 filaments exhibit higher bundling as compared to the ATPase mutant (E134A). The open circle indicates the mean and the black bar represents the 95 % CI. ScMreB5^{WT} – 1.14 ± 0.04 95% CI, (n=112) and ScMreB5^{E134A} – 0.95 ± 0.05 95% CI, (n=32).

This will be part of Supplementary Figure Fig. S6 in the revised manuscript.

Response Figure 4. ScMreB5 WT and mutants do not pellet in the absence of liposomes and/or nucleotides

- A. A representative 12% SDS-PAGE gel of pelleting assay with ScMreB5 WT and ATPase mutant (E134A) shows that the protein does not pellet in the absence of liposomes. ScMreB5 WT protein also does not pellet in the absence of liposomes and in presence of ATP, ADP (left) and AMP-PNP (right). Nucleotide concentration used is 1 mM.
- B. A representative 12% SDS-PAGE gel of pelleting assay with membrane binding loop mutants shows that the protein does not pellet in the absence of liposomes.
- C. A representative 12% SDS-PAGE gel of pelleting assay with C-terminal deletion mutant (Δ C10) shows that the protein does not pellet in the absence of liposomes.

These assays were performed at a protein concentration of 2 μ M and spun at 100,000 xg for 25 mins at 25°C. P and S represent the pellet and supernatant fractions of the protein. For the gels in panel A, the pellet fraction has been loaded in the same gel after 10 min of starting the electrophoresis run with the samples of the supernatant fraction. The marker on the left was loaded along with the pellet samples, while the marker on the right was loaded along with the supernatant samples.

This will be part of Supplementary Figure Fig. S7 in the revised manuscript.

Response Figure 5. Nucleotide dependence of liposome interaction of ScMreB5

A. A representative 12% SDS-PAGE gel of pelleting assay with WT and ATPase mutant (E134A) shows that the amount of pellet fraction is dependent on the nucleotide. NoN denotes no additional nucleotide, while ADP, ATP and AMP-PNP denote addition of 2 mM of respective nucleotides to 2 μ M protein in the reaction mixture.

B. Quantification of the intensity of the pellet fractions from 3 different repeats of the experiment.

These assays were performed at a protein concentration of 2 μ M and spun at 100,000 \times g for 25 mins at 25°C. P and S represent the pellet and supernatant fractions of the protein. For panels A and B, the pellet fraction has been loaded in the same gel after 10 min of starting the electrophoresis run with the samples of the supernatant fraction. The marker on the left was loaded along with the supernatant samples, while the marker on the right was loaded along with the pellet samples.

July 14, 2021

Re: JCB manuscript #202106092T

Dr. Pananghat Gayathri
Indian Institute of Science Education and Research Pune
Indian Institute of Science and Research
Pune 411008
India [IN]

Dear Dr. Gayathri,

Thank you for submitting your manuscript entitled "Filament dynamics driven by ATP hydrolysis modulates membrane binding of the bacterial actin MreB." Please accept our apologies for the delay in the processing of your manuscript. We have now had a chance to assess the comments of the Review Commons referees and agree with their assessment that your study represents a significant advance in our understanding of MreB. We invite you to submit a full revision as outlined in your response to the reviewer comments.

GENERAL GUIDELINES:

Text limits: Character count for an Article is < 40,000, not including spaces. Count includes title page, abstract, introduction, results, discussion, acknowledgments, and figure legends. Count does not include materials and methods, references, tables, or supplemental legends.

Figures: Articles may have up to 10 main text figures. Figures must be prepared according to the policies outlined in our Instructions to Authors, under Data Presentation, <https://jcb.rupress.org/site/misc/ifora.xhtml>. All figures in accepted manuscripts will be screened prior to publication.

*****IMPORTANT:** It is JCB policy that if requested, original data images must be made available. Failure to provide original images upon request will result in unavoidable delays in publication. Please ensure that you have access to all original microscopy and blot data images before submitting your revision. ***

Supplemental information: There are strict limits on the allowable amount of supplemental data. Articles may have up to 5 supplemental figures. Up to 10 supplemental videos or flash animations are allowed. A summary of all supplemental material should appear at the end of the Materials and methods section.

As you may know, the typical timeframe for revisions is three to four months. However, we at JCB realize that the implementation of measures to limit spread of COVID-19 also pose challenges to scientific researchers. Therefore, JCB has waived the revision time limit. We recommend that you reach out to the editors to discuss an appropriate time frame for resubmission. Please note that your revised manuscript will be sent for re-review to the same reviewers that evaluated it for Review Commons. Additionally, JCB policy is that papers are generally considered through only one major revision cycle, so any revised manuscript will likely be either accepted or rejected.

Thank you for this interesting contribution to Journal of Cell Biology. You can contact us at the journal office with any questions, cellbio@rockefeller.edu or call (212) 327-8588.

Sincerely,

Kenneth Yamada, MD, PhD
Editor
The Journal of Cell Biology

Dan Simon, PhD
Scientific Editor
The Journal of Cell Biology

Reviewer #1 (Evidence, reproducibility and clarity (Required)):

In this paper, Pande et al. investigate the in vitro properties of the prokaryotic actin-like protein MreB, purifying one of its counterparts (MreB5) from Spiroplasma, a clever approach given that there are major difficulties in working with MreB proteins extracted from bacterial model systems such as E. coli or B. subtilis. The paper first resolves the structure of MreB5 and shows that it adopts the expected MreB fold. The authors then proceed in investigating the function of nucleotide-binding and in particular the role of ATP hydrolysis, a still much debated aspect of MreB function. Combining Cryo-Electron microscopy, reconstitution in fission yeast, and lipid interaction assays, the authors propose that ATP-binding in MreB polymers promotes a global MreB polymer conformational change that facilitates its binding to negatively charged lipids.

****Major comments:****

The paper makes an ambitious claim that if correct would significantly advance our understanding of MreB polymer biochemistry in cells. However, while there are many potentially interesting observations in the manuscript they are still quite indirect and some potentially artefact-prone to truly support the strong conclusions.

The main difficulty of the paper is the absence of robust biochemical assay to quantify polymer dynamics and their nucleotide dependence in vitro and correlate such dynamics to ATP hydrolysis measurements and lipid-binding. In absence of such assay protein aggregation artefacts cannot be excluded when site-directed mutants are constructed, greatly limiting the conclusions. Some examples are provided below to explain these limitations.

Authors' response:

Necessary control experiments for ruling out protein aggregation artefacts have been performed, as detailed below and in the rest of the response.

For all the site directed mutants, the protein samples have been through a step of size exclusion chromatography during purification and the eluted monomeric fractions were used in the assays.

In order to look at the aggregation state of mutants, analytical size exclusion chromatography has been performed for all of them. All mutants elute as a monomer, similar to the wildtype. These profiles have been included as Supplementary **Fig. S3 B, C** in the revised manuscript. The protein fraction in the void is negligible for all these mutants, as can be seen in the figure panels.

- In Figure 1 I, ATPase activity is measured in various mutants but given that each mutant may form distinct polymer concentrations it is really difficult to disentangle if the decrease in the ATPase activity is linked to a polymerization issue or to a hydrolysis issue or to both of them. For example, the K57A interface mutant is said to no longer polymerize (is this true?) but hydrolyze ATP. Does this mean polymerization is not required for ATP hydrolysis?

Authors' response:

We have currently not disentangled whether the mutants affect polymerization, hydrolysis or both. The hydrolysis rate is indeed expected to be a cumulative effect of polymerization and catalytic activity for most of the characterized filament forming proteins of the actin family.

ATPase mutants of *residues at the nucleotide-binding site*: Though these residues have been implicated to play a role in hydrolysis based on comparison with other actins, a mutational study with biochemical characterization of all the relevant residues is not available till date for MreBs.

In the case of K57A mutation, the residue does not lie at the ATP binding site, but at the *polymerization interface* (as seen from the ScMreB5 structure). Hence, the decrease in activity was hypothesized to be due to compromised polymerization.

Based on time-dependent light scattering measurements performed with the monomeric protein (fraction eluted after size exclusion chromatography), we see that there is an increase in scattering intensity within a few minutes of elution (included as Supplementary Fig. S3 D, E in the revised manuscript), similar to the wild type. Hence, the K57A mutant can potentially polymerize. However, the decreased ATPase activity might suggest a conformation of filaments that is not optimal for ATPase activity, or a decrease in the polymerized content due to a sub-optimal interface.

To clarify this further to the readers, we have included the following sentences to the revised version of the manuscript, along with the light scattering data in Supplementary Fig. S3 D, E.

“Light scattering measurements show that ScMreB5^{K57A} might polymerize (Fig. S3, D and E). The mutation of a single residue at the interface might not abrogate polymerization completely, but might result in a sub-optimal interface. This could lead to a decrease in polymerized content or a sub-optimal conformation of the ATPase active site within the polymers, thereby leading to a decrease in ATPase activity.”

[page 9, lines 5 - 10]

The data is now part of Fig. 2, and not Fig. 11, in the revised version of the manuscript.

The light scattering data is shown in Supplementary Fig. S3 D, E of the revised manuscript.

- The capacity of MreB to form filaments bundle independently of the nucleotide state is clearly presented by cryo-EM. However, this technique is not quantitative and cannot appropriately show that lower filament densities are obtained in the presence of ADP or when the E134A mutant is in presence of AMP-PNP is not ideal. Of note, the nucleotide state is not crucial for polymerization even for eukaryotic actin. It is possible that the the critical concentration for polymerization that is lowered in the presence of ATP. Could titration experiments be performed to show whether the polymer assembly is changed depending on concentration and ATP or ADP availability?

Authors' response:

For performing titration experiments, polymerization kinetics experiments using light scattering has not been responsive to the addition of excess nucleotides for ScMreB5.

The purified ScMreB5^{WT} comes bound to ADP. We have included this data now in Supplementary **Fig. S1 E**. Addition of EDTA to obtain nucleotide-free protein results in precipitation. The protein does not appear to be stable enough for biochemical assays in the absence of the bound nucleotide.

ScMreB5^{WT} polymerizes independent of the addition of excess of nucleotide in vitro immediately (within 10 minutes) post elution of the monomer fraction from size exclusion chromatography (Supplementary **Fig. S3 D**). Once polymerized, depolymerization of the filaments has not been observed (based on light scattering). The addition of excess nucleotide (ADP, ATP or AMP-PNP) at plateau stage also does not have an effect on light scattering.

Since the experiment has to be performed within a few minutes of elution after size exclusion chromatography (SEC), it was difficult to ensure the exact concentration for different repeats for polymerization assay post SEC. Dilution of the protein to lower concentrations (with an aim of working below the critical concentration of spontaneous association) did not solve the challenge probably because of sensitivity issues at low concentrations. The scattering signal from low concentrations of protein was at noise level and were not reliable. Because of these reasons, we were not able to perform quantitative assays to estimate critical concentrations for ScMreB5 in the presence of ATP/ADP/AMP-PNP.

Hence, we shifted to yeast expression to observe polymerization. Microscopy-based approaches, though not as quantitative as an in vitro approach, overcome one of the disadvantages of interpreting a light scattering assay – direct visualization of the filaments help in distinguishing between individual filaments, bundling, disassembly or protein aggregation events. These are indistinguishable in a light scattering experiment and often leads to potential artefactual interpretations of light scattering data.

- The fission yeast system has been used in the past and provides an interesting perspective to look at polymers but there are a number of limitations with the authors interpretation. As presented, the results are mostly presented as example snapshots and lack overall quantification and statistics. It is not currently clear that bundling is affected in the E134A mutant as compared to the WT. Surely the density, length and spatial distributions of the filaments could be quantified across yeast cells for the various MreBs.

Authors' response:

We have quantified and presented statistics (as detailed below) in the revised manuscript.

We have quantified the bundling of E134A mutant using the intensity-metric method of coefficient of variation (Higaki, T., Akita, K. & Katoh, K. *Coefficient of variation as an image-intensity metric for cytoskeleton bundling. Sci Rep* **10**, 22187 (2020). <https://doi.org/10.1038/s41598-020-79136-x>; demonstrated for evaluating actin bundling in images obtained by wide-field microscopy). We have included this data along with the microscopy images in Fig. 4 of the revised version (**Fig. 4D**).

For the spatial distribution, we have quantified the anisotropy of WT and E134A filaments using the FibrilTool (Boudaoud, A., Burian, A., Borowska-Wykręć, D. et al. *FibrilTool, an ImageJ plug-in to quantify fibrillar structures in raw microscopy images. Nat Protoc* **9**, 457–463 (2014). <https://doi.org/10.1038/nprot.2014.024>), which again quantifies bundling and parallelness of filaments against the long-axis of the fission yeast cells (included in **Fig. 4C** of the revised version).

The quantification approaches indeed support differences in bundling between the wild type and E134A mutant, at equivalent levels of protein expression (as evidenced from Western blot analysis included in **Fig. 5 D** of the revised manuscript).

Description of these analyses have been included in the revised manuscript from page 10 (line 15) to page 11 (line 11).

The method is described in Materials and Methods section under the heading of “Quantitative analyses of ScMreB filaments” (page 30 line 24 to page 32 line 2).

It would also be interesting to test the various mutants of this study in the yeast system, perhaps as a way to characterize their filament-forming properties in a quantitative way.

Authors' response:

We do plan to test the yeast system for characterization of the various mutants. However, we would like to maintain the focus of this manuscript on the E134A mutation for which we have data for a complete mechanism because of a structural explanation, biochemical data on ATPase activity and liposome binding experiments. Characterization of the other mutants are ongoing experiments, along with testing these mutants in their ability to complement motility and shape defects in *Spiroplasma*.

Why does the yeast system not capture the expected MreB membrane interactions?

Authors' response:

The main objective of the yeast expression in this study was to qualitatively observe the polymerization dynamics independent of membrane binding, which was technically challenging to do in vitro due to reasons mentioned above.

We (Srinivasan, et al, 2007) and others (Karczmarek A, et al, 2007) have earlier found that the N-terminal GFP fusion in *E. coli* MreB (EcMreB) abolishes membrane binding and induces bundling, likely because the GFP is linked to the N-terminal amphipathic helix of EcMreB. However, an EcMreB sandwich construct localizes to the membranes in yeast and orients perpendicular to the long axis, as expected for MreBs (Srinivasan, unpublished results; **Response Fig. 1A**).

Response Fig. 1. GFP/mVenus fusion constructs of MreB

- A. EcMreB sandwich construct (mVenus inserted after residue 228) orients perpendicular to the long axis upon yeast expression. Scale bar represents 5 μm .
- B. Pelleting assay showing that N-terminal GFP-ScMreB5 construct binds liposomes. Liposome preparation used here is 600 μM of 80% DOPG and 20% DOPC

For ScMreB5, we have checked the ATPase activity and liposome binding features of N-terminal GFP fusion, and observe that it is as active as the wild type, and is able to bind liposomes (**Response Fig. 1B**). Compared to EcMreB, where the N-terminal amphipathic helix is the sole membrane anchor, our results show that ScMreB5 binds membranes through an extensive surface. From the structure, we observe that a GFP fusion at the N-terminal end could possibly be directed away from the membrane interface. This is consistent with our observation of the ability of N-terminal GFP fusion to bind liposomes (**Response Fig. 1B**) and the functionality of N-terminal GFP fusion constructs of BsMreB, which also lacks an amphipathic helix at its N-terminus (Jones et al., 2001; Garner et al., 2011).

Since this is a novel mode of membrane binding, the properties of alignment of ScMreB5 filaments in yeast might be different compared to EcMreB. This might be due to lipid composition differences and/or membrane curvature.

We are currently cloning a sandwich construct for ScMreB, similar to the one constructed for *E. coli* MreB and plan to express the same in fission yeast and test its localization and assembly properties as well as its ability to function in *Spiroplasma*. These are ongoing experiments with a focus on understanding membrane curvature sensing and orientation by ScMreB5.

- The sedimentation assay is an interesting first attempt to look at MreB membrane interactions, but it is rather low resolution and not free of potential artefacts. This could be especially worrisome if the tested MreB mutants have slower filament forming dynamics or form aggregates which might alter their own sedimentation properties.

Authors' response:

ScMreB5 filaments do not sediment upon polymerization (neither without addition of nucleotide nor upon ADP, ATP or AMP-PNP addition), except in the presence of liposomes (Supplementary Fig. S4 A, E).

Aggregation of wildtype and the mutants does not occur at protein concentrations used in the assay. The experiments shown in Supplementary Fig. S3 B – D are typically at higher concentrations than in the pelleting assay experiments. The respective control runs (without the liposomes) have been performed for all repeats of the pelleting assay experiments (representative data in Supplementary Fig. S4 A, E). Hence, observation of protein in the pellet fraction is indeed driven by liposome-protein interactions.

We also demonstrate that the pelleting effect is dependent on the liposome composition and concentration (no protein in pellet fraction in the presence of 100% DOPC liposomes; Fig. 7 A), which can be fitted to a binding curve (Supplementary Fig. S5 in original manuscript and Supplementary Fig. S4 C in the revised manuscript). This experiment also supports that the observed sedimentation is a feature of lipid-protein interaction, and not due to aggregation of the protein.

In addition, we have confirmed these observations using an alternate approach of exploring protein-liposome interactions (Jose, GP; Pucadyil, TJ, PLiMAP: Proximity-Based Labeling of Membrane-Associated Proteins, *Current Protocols in Protein Science*, 2020), which also give results consistent with the observations in liposome pelleting assays. A representative data for the binding assay for ScMreB5 with DOPC and DOPG is shown in Response Fig. 2. This has been performed for most of the liposome binding experiments reported in this work and confirmed that it is not an artefact of protein-protein aggregation. This data will be part of a future manuscript on further mechanistic understanding of the membrane interactions of ScMreB5.

Response Fig. 2. PLiMAP (Proximity Based Labeling of Membrane-Associated Protein) experiment demonstrating that the ScMreB5 specifically binds to DOPG liposomes. DOPG and DOPC liposomes were prepared by mixing 99% DOPG/DOPC with 1% bi-functional lipid, BDPE (BODIPY-diazirine phosphatidylethanolamine) which has diazirine group as a UV cross-linker and BODIPY as the fluorophore. Addition of 2 μ M protein to this liposome mix is followed by 15 mins incubation at 25 °C and UV crosslinking. The UV cross-linked and non-cross-linked samples are run on 12 % SDS-PAGE. The bands are imaged under GFP channel to detect the cross-linked protein, followed by Coomassie staining. The presence of a band in the GFP channel indicates that lipids were attached to protein during the UV exposure.

(<https://currentprotocols.onlinelibrary.wiley.com/doi/abs/10.1002/cpps.110>)

The control runs demonstrating that ScMreB5 and mutants do not sediment in the absence of liposomes are included in Supplementary Fig. S4 A, E of the revised manuscript.

- The authors use this assay to propose their conformational change hypothesis, but again not knowing the exact polymerization properties of the mutants is problematic. For example, if the I and W mutant tend to have slower polymerization than the WT, this could explain why they are found less in the pellet in Figure 4 E-F.

Authors' response:

The lower amount of protein in the pellet for the mutants could be for any of the following three reasons:

- a) The mutation of a membrane-binding interface residue that directly interacts with the liposome will affect binding.
- b) Lower levels of polymerization could lead to less binding to the liposomes.
- c) Geometry of the filament and their higher organization such as bundles could affect liposome binding in response to the curvature.

For I95A, W96A and Δ C10 mutants, since the mutations are expected to be at the membrane binding interface, possibly, the direct binding interface might be affected (point (a) above). A decrease in polymer content for these constructs (point (b) above) could imply an allosteric effect of membrane binding on polymerization or vice versa because the membrane-binding interface is away from the polymerization interface.

For the E134A mutant, the decrease in binding could be due to the reduced polymerization and/or bundling, leading to reduced binding to the liposomes. Effect of the mutation leading to reduced polymerization or bundling indeed implies an allosteric effect because E134 is located in the ATP-binding pocket of the MreB fold and away from both polymerization interface or membrane-binding interface.

Hence, the results of the assay indeed are suggestive of the involvement of conformational changes in driving efficient membrane binding, irrespective of the polymer content.

The point that ScMreB5 and its mutants do not sediment in the absence of liposomes is further emphasized in the revised manuscript as follows:

“Prior to the addition of liposomes, the protein samples were spun at 21,500 g to ensure that any protein aggregates were removed. Pelleting assays of the reaction mix without liposomes served as negative controls for the liposome-binding experiments (Fig. S4 A). The control runs showed that the protein does not pellet on its own in the absence of liposome, irrespective of its polymerization state.”

[page 12, lines 12 - 17]

- Page 12, the authors suggest "an allosteric communication between the nucleotide pocket and the membrane binding interface of MreB". This is an attractive idea but how is the membrane interaction affected with various nucleotides?

Authors' response:

We have carried out liposome binding assays upon addition of various nucleotides namely ADP, ATP and AMP-PNP. Our results show us that there is a differential

binding for the wild type protein too based on the nucleotide state, similar to the effect of E134A mutation. This data has now been included in **Fig. 7 G-H** of the revised manuscript.

This has been included in the last section of the results, as follows:

“The effect of E134A mutation on liposome binding prompted us to explore the interdependence between nucleotide state and liposome binding. We carried out liposome pelleting assays of ScMreB5^{WT} upon addition of either ADP or ATP or AMP-PNP in the reaction mix. Pelleting assays of the reaction mix without liposomes showed that the protein does not pellet upon addition of nucleotide in the absence of liposome (Fig. S4 E). The observations from the pelleting assays with liposomes suggest that there is a differential binding for ScMreB5^{WT} based on the nucleotide state (Fig. 7, G and H), similar to the effect of E134A mutation.”

[page 14, lines 10 - 17]

The interpretations are included in the discussion as follows:

“Different nucleotide states might possess distinct conformations or differential capabilities to form bundles, or bind efficiently to specific curvatures, thus contributing to the sensing of membrane curvature by the MreB filaments. The effect of liposome binding and the accompanied conformational changes on the MreB filaments is not known – this interaction might also stimulate nucleotide exchange and/or hydrolysis. It is possible that the filament conformations or bundling features of the different nucleotide states can either i) match the curvature of the liposomes, ii) remodel the liposomes to match the filament curvature or, iii) fall off in case of a curvature mismatch. The effects observed in this study is based on liposomes with protein added onto its exterior convex surface. The binding dependence on nucleotide state might have a different effect from a concave surface.”

[page 16, line 17 - page 17, line 3]

These experiments have led us to questions related to the effect of liposomes on hydrolysis rate and ADP release/exchange from MreB filaments. Preliminary experiments appear to suggest that liposomes indeed stimulate ATP hydrolysis, the mechanism of which is currently being investigated. Further experiments with reconstitution of MreB on liposomes and the effect of mutations and nucleotide states are currently under progress. We plan to include this data as part of a future manuscript focusing on the mechanistic basis of membrane binding by MreB.

Observations from liposome binding assays for ScMreB5 with ADP, ATP and AMP-PNP addition are included in Fig. 7G, H in the revised manuscript.

The control runs for pelleting assay in the absence of liposome, showing that the protein does not sediment upon nucleotide addition, are included in Supplementary Fig. S4E.

- As presented the Figure 6 model is difficult to understand and it is not presented in details in the discussion section.

Authors' response:

This has been rephrased in the revised submission of the manuscript to clarify the point further by including references to the relevant figure panels. To quote:

“The entire network of interactions (labeled as i – vii in Fig. 8, inset) with all the 4 subdomains was observed only in the double filament conformation (PDB ID 4CZJ; Fig. 8) and not in the single protofilament or monomeric states (PDB IDs 4CZI, 4CZF or 4CZM in Fig. 8).”

[page 15, lines 12 - 15]

****Minor comments:****

- Could the authors provide in the SI an SDS-PAGE figure of the purified protein

Authors' response:

SDS-PAGE profiles of all purified proteins used in this study are shown in Supplementary **Fig S3 A**) of the revised manuscript.

- Could the authors indicate what the arrows point to in Figure 3E.

Authors' response:

This information has now been included in the figure legend of the revised version of the manuscript (**Fig. 5E** in the revised manuscript).

“While white arrows indicate bundling events in ScMreB5, white arrows point to the site of septation and highlight the bundling events that happen at the time of cell division for ScMreB5^{E134A} (in 8 cells out of 41).”

[page 46, lines 18-21]

- Page 12, the authors mention "Figure 5 F-G". These panels are absent from the Figure.

Authors' response:

The error has been corrected in the revised submission of the manuscript.

- Page 27, the reference "12" is mentioned twice (12 and 13)

Authors' response:

The error has been corrected in the reference list of the revised submission of the manuscript.

Reviewer #1 (Significance (Required)):

****Significance:****

In recent years a number of important discoveries have been made with regards to the in vivo function of the bacterial MreB cytoskeleton, the main function of which is now thought to form a dynamic polymeric scaffold to orient cell wall complexes around the cell cylinder. However, the exact mechanisms driving MreB polymer dynamics and in particular the function of ATP hydrolysis have remained a major black box due to difficulties in developing robust biochemical approaches to study MreB in vitro. In this context, demonstrating that nucleotide binding affects the

conformation of the polymers in such a way that they will tend to bind the bacterial membrane would be a major step toward understanding how MreB polymers form spontaneously against curved membrane as has been described by a number of laboratories.

While the data presented goes in this direction, it does not formally prove this mechanism which limits its current significance for people working on the bacterial cell envelope and more generally on bacterial cell biology.

Authors' response:

We thank the reviewer for providing inputs to improve the manuscript. We believe that the modifications to the manuscript help addressing the concerns raised and provides support for the mechanism.

Reviewer #2 (Evidence, reproducibility and clarity (Required)):

****Summary:****

MreB is a critical protein for rod shape in many bacteria species. MreB is thought to regulate cell shape through the organization of the cell wall synthesis enzymes, yet wall-less Spiroplasma depend on MreB to regulate their shape. MreB forms polymers on the cytoplasmic face of the plasma membrane and can hydrolyze ATP. Although the exact role of ATP hydrolysis is not known it has been suggested to be involved in structural reorganization of MreB monomers which could affect polymerization and membrane binding. This study uses Spiroplasma to study the role of MreB ATP hydrolysis on filament organization and membrane binding through crystallization, cryo-EM, lipid-binding assays, and mutational analysis.

****Major Comments:****

Overall the work is well-done and the authors conclusions are grounded. However, some additional controls and quantification would be useful.

1) Do the authors know that their mutant MreB proteins are properly folded?

Authors' response:

Analytical size exclusion chromatography runs for the mutants have been performed which shows that all mutants elute as monomer, indicating they are well folded.

This data is included as Supplementary Fig. S3 B, C in the revised manuscript.

2) Figure 3: Since the authors expressed E. coli MreB in yeast with an N-term fusion it has been shown that the observed structures in E. coli with this fusion are artifacts (Swulius and Jensen, J Bac, 2012) as well as being non-functional, which is why most researchers have switched to a functional sandwich fusion. Do the authors have any evidence that their N-terminal fusion is functional?

Authors' response:

To ascertain the activity and functionality of N-terminal GFP fusion of ScMreB5, we have purified the N-terminal GFP tagged ScMreB5 and estimated its ATPase activity (**Table S2**) and liposome binding *in vitro* as well (**Response Fig. 1B**). It is ATPase active and it binds liposomes. The value of k_{obs} is $0.12 \pm 0.01 \text{ min}^{-1}$, is similar to the k_{obs} for the wildtype ($0.15 \pm 0.007 \text{ min}^{-1}$).

Based on our yeast imaging experiments, the N-terminal fusion construct appears to be functional for polymerization, and depolymerization too. Comparison with the E134A mutant suggests that the WT protein can undergo depolymerization while the E134A (ATPase deficient mutant) cannot. This also suggests that the fusion protein might be a functional ATPase.

The N-terminal fusion in *E. coli* MreB is likely non-functional in vivo due to the loss of the essential membrane binding activity, because the N-terminal amphipathic helix of *E. coli* MreB is directly tagged to the EGFP. N-terminal GFP fusions of BsMreB, which lacks an amphipathic helix at its N-terminus are functional (Jones et al., 2001; Garner et al., 2011). Nonetheless, the N-terminal fusion in *E. coli* MreB is fully functional for polymerization and the polymerization is inhibited by A22) (Srinivasan, 2007). Further, our unpublished results using the functional sandwich fusion of EcMreB (mVenus between aa 228-229) in yeast show that it localizes efficiently on membranes and assembles into short polymers or patches on membranes (**Response Fig. 1A**). Hence, the difference between the N-terminal fusion and a sandwich fusion is probably due to the membrane-binding features.

Thus, in the heterologous yeast expression studies, while the functional sandwich construct captures membrane binding of MreB effectively, the N-terminal fusion turns out to be a better indicator for polymerization and bundling. Since, the main objective of the yeast expression in this study was to qualitatively observe the polymerization dynamics independent of membrane binding, we resorted to the use of the N-terminal GFP fusion to ScMreB for our expression studies in fission yeast and assess polymerization and bundling.

We are currently cloning a sandwich construct for ScMreB, similar to the one constructed for *E. coli* MreB and plan to express the same in fission yeast and test its localization and assembly properties as well as its ability to function in *Spiroplasma*. These are ongoing experiments with a focus on understanding membrane curvature sensing and orientation by ScMreB5.

ATPase activity of N-terminal fusion construct is included in Table S2 of the revised manuscript and liposome binding assay in Response Fig. 1.

Do the authors know that their mutant protein is being expressed in the yeast at the same levels? Perhaps a western blot or total cell fluorescence can be measured. This could explain why less cells form structures.

Authors' response:

A western-blot analysis using cultures at the same OD₆₀₀ indicate that the expression levels of WT ScMreB5 and E134A mutant are similar. This has been performed from multiple cultures and the expression levels quantitated as a fraction of intensity of tubulin as loading control. This indeed shows that the mutant and wild type proteins are expressed to the same levels.

The data has been included as **Fig. 5 D** in the revised manuscript.

Additionally, we have also provided quantification of the time lag for observation of polymerization for the yeast cells expressing WT and mutant MreB5, and a comparison of the percentage of cells forming filaments with increasing time periods after induction (**Fig. 5 B, C**).

Revised text related these figure panels have been included in the revised manuscript as follows:

“Estimation of time taken to initiate polymerization (from the time cells were placed on agarose pads) showed that while cells expressing ScMreB5^{WT} started to form polymers in 26.6 minutes (n=7, 95% CI = 15.0), ScMreB5^{E134A}

started to assemble filaments much later at 69.0 minutes (n=9, 95% CI = 11.5) (Fig. 5, A and B). A time-course experiment and quantification of the percentage of cells exhibiting polymers in ScMreB5^{WT} and ScMreB5^{E134A}, further confirmed the time lag in polymerization of ScMreB5^{E134A} (Fig. 5 C). This was not due to differences in the expression levels of the mutant. Western blotting with anti-GFP antibodies and quantification of proteins levels with tubulin as internal control shows that both ScMreB5^{WT} and ScMreB5^{E134A} expressed at similar levels (Fig. 5 D). Taken together, these results and the observation that filaments of ScMreB5^{E134A} were seen in fewer cells compared to ScMreB5^{WT} (Fig. 4 E) suggests a requirement of a higher concentration of monomers for polymerization of ScMreB5^{E134A}.”

[page 10, line 23 - page 11, line 11]

Western blot analysis which demonstrates that the mutant protein expresses at the same levels as the wild type is included as Fig. 5 D in the revised manuscript.

It would also be nice to know how many cells were observed (or percent of cells) with the phenotypes in E, F, and G. Also what is the time lag in 3D between the formation of polymers in WT and mutant expressing cells.

Authors' response:

We have counted the cells in which we observed the phenotypes reported in E, F and G and have included this information in the manuscript text and the concerned figure (**Fig. 5**) of the revised manuscript.

For the time lag between the assembly of polymers in WT and mutant, we have performed a time-course experiment and represented the percentage of cells having polymers as a function of time after induction of expression.

Accordingly, the manuscript text has been revised as follows:

“Estimation of time taken to initiate polymerization (from the time cells were placed on agarose pads) showed that while cells expressing ScMreB5^{WT} started to form polymers in 26.6 minutes (n=7, 95% CI = 15.0), ScMreB5^{E134A} started to assemble filaments much later at 69.0 minutes (n=9, 95% CI = 11.5) (Fig. 5, A and B). A time-course experiment and quantification of the percentage of cells exhibiting polymers in ScMreB5^{WT} and ScMreB5^{E134A}, further confirmed the time lag in polymerization of ScMreB5^{E134A} (Fig. 5 C).”

[page 10, line 23 - page 11, line 4]

The relevant numbers are included in the figure labels and new figure panels included for time lag (Fig. 5 B, C) in the revised manuscript.

The authors suggest that higher concentration of E134A monomers is needed to get polymerization in yeast. This could be easily tested by overexpressing their constructs. Again quantification would be needed.

Authors' response:

The quantification of the percentage of cells having polymers as a function of time after induction of expression as described above in the time course experiment (**Fig.**

5C) provides us with a comparison of filaments formed by ScMreB5 WT or E134A at the same levels of expression, and serves as an indicator that higher concentration of E134A monomers is needed to get polymerization in yeast.

The quantification data has been included in the revised version of the manuscript (Fig. 5 B-D).

Overexpression of ScMreB5 using a stronger promoter such as *nmt1* is a possible experiment. Earlier attempts of using *nmt1* promoter for bacterial cytoskeletal filaments such as FtsZ have not been successful due to aggregation driven by the strong expression. Hence, we preferred a tunable expression using the medium promoter *nmt41* for ScMreB5 expression, and the current experiments were performed in the absence of thiamine, which is the highest expression level. We have repeated the experiments with partial repression in the presence of 0.05 μ M thiamine, which leads to a further decrease in the percentage of cells exhibiting filaments in ScMreB5^{WT} as well (**Response Fig. 3**).

Response Fig. 3. Effect of decrease in protein expression by partial repression by thiamine addition

The number of cells exhibiting filaments decreased upon reducing the expression levels of ScMreB5 by addition of 0.05 μ M thiamine. Error bars represent 95 % CI.

We hope that the quantification of the protein levels and the filament content under partial repression (lower expression levels) addresses the reviewer's concerns.

3) The authors should explain the differing results in fig 4-5 more. Quantification of Fig. 5A would be useful. It appears that W96A has a much higher concentration in the pellet fraction when mixed with 100% DOPG as opposed to fig 4. Also why do the authors believe that the difference in I95A in fig 4 is not statistically significant but is in fig 5 D is important. What about the fact that W96A was less than WT in Fig 4 but more in both fig 5?

Authors' response:

Fig. 6 in the revised manuscript (*Fig. 4E, F in the original manuscript*) shows the binding of wildtype and the mutants with the liposomes that mimic the whole cell membrane composition of *Spiroplasma citri*. Here, binding of the wild type and the membrane binding mutants appear to be similar. However, we had used a liposome concentration of 1 mM in these experiments, and we wanted to ensure that we were not comparing the binding curves at the saturated phase. Hence, we shifted to

carrying out liposome binding assays at a constant concentration of liposomes, but varying the content (percentage) of DOPG, which is a better comparison of the effect of lipid composition on binding without changing the liposome amounts.

For Fig. 5 (**Fig. 7** in the revised manuscript), to decipher the lipid contributions for MreB5 binding to the membrane, we compared the binding with the negatively charged lipid DOPG (38% of negatively charged lipids are present in *Spiroplasma* lipid composition) and a neutral lipid DOPC (14% in *Spiroplasma* lipid composition). Thus, we took lipids representing these two lipids in terms of charge and percentage to see the binding effect.

Quantification of Fig. 5A (original manuscript numbering; **Fig. 7A** in the revised manuscript) was provided as a Supplementary figure in the original manuscript. We have shifted the quantification to the main manuscript figure (**Fig. 7B**) in the revised version.

The difference in W96A in Fig 4 (**Fig. 6** of the revised manuscript) and Fig. 5A (**Fig. 7** of the revised manuscript) arose because both these experiments are performed at different liposome compositions, as explained above. A similar effect has been observed for I95A mutant as well. In the Fig. 5D experiment, the composition of liposome (80% DOPG and 20% DOPC) brought out the difference in terms of liposome binding, which was earlier masked when 100% DOPG (**Fig. 7A, B**) and/or *Spiroplasma* composition (**Fig. 6E**) was used.

We have included the effect of I95A also in the Results section of the revised manuscript text. However, the double mutant ScMreB5^{IWA} did not affect the binding significantly, reasons for which are currently not understood.

- ***In the revised version, we have highlighted the differences clearly by mentioning the liposome composition below the relevant gels.***
- ***We have also highlighted the P-values corresponding to the statistical differences between the mutants and the wild type, and revised the text accordingly. This provides an explanation of the statistical significance of the effect of I95A mutation.***

There is no fig 5F-G

Authors' response:

This error has been corrected in the revised version of the manuscript, according to the new figure numbers.

4) The authors use of language is too harsh in the discussion. Please remove proves from the end of the first paragraph. Please make statements that are hypothesis or models known. For ex: 'Thus, the residue MAY act as the sensor..', 'Thus, WE PROPOSE Glu140 drives the conformational switch...'

Authors' response:

These sentences have been rephrased accordingly in the revised submission of the manuscript.

“Thus, the residue may act as the sensor for the ATP-bound state and trigger transition to the double protofilament conformation”

[page 15, lines 19 - 20]

The authors state that ATP hydrolysis is essential for a conformational switch yet the crystal structures of MreB with ADP or AMP-PNP look remarkably similar and there is no crystal structure for the E134A mutant, so it seems unclear how they can make such a direct statement about the role of ATP binding.

Authors' response:

The crystal structures for ScMreB5 complexed with ADP and AMP-PNP, respectively, are both single protofilament states, and were not captured in the double protofilament conformation. Hence all the structural basis has been explained by carefully analysing the CcMreB structures, the only MreB where double protofilament, single protofilament and monomeric conformations have been captured.

A crystal structure of E134A may not be as informative because the side chain will be missing. It can be explained best only based on a structure with intact glutamate side chain and bound ATP.

****Minor comments:****

In the future it would be beneficial to put line numbers even if it is not required. Also either indent new paragraphs or add an extra space

Authors' response:

We have made these changes in the revised submission of the manuscript.

Line 3: 'non-spherical bacteria' is confusion here. Do you mean rod shaped bacteria or bacteria that did not become round when MreB is lost? Please reword.

Authors' response:

We have deleted the words in the revised submission of the manuscript.

Gram should be capitalized.

Authors' response:

The required changes have been made in the revised submission of the manuscript. Please refer **page 3, lines 20, 21**.

Ouzounov et al. Biophys J 2016 is a missing reference for MreB and cell width

Authors' response:

This reference has been cited appropriately in the revised submission of the manuscript. To quote the first instance of citing:

“Mutations in MreB can result in cells of varying width (Ouzounov et al., 2016; Shi et al., 2017).”

[page 5, lines 15 - 16]

Figure 11 and table 3 do not seem to match. For example, E134A says its Kobs is .05 in the table and D70A is .02, yet in the figure D70 is clearly above E134A

Authors' response:

We thank the reviewer for pointing out this error. We have corrected this and updated **Table S2** in the revised submission of the manuscript.

Can these MreB mutants be expressed in vivo?

Authors' response:

We are in the process of expressing these mutants in *Spiroplasma* ASP-I (a strain deficient in MreB5, refer our previous work Harne, et al, 2020, Current Biology).

However, this work will not be part of the current manuscript because it will take about a year or more to generate the mutant strains, complete the experiments, and characterize the mutants to obtain mechanistic insights. We look forward to these results and plan to include these in a separate manuscript in future, along with characterizing all these mutants in yeast for their polymerization dynamics.

Reviewer #2 (Significance (Required)):

****Significance****

MreB is found in many rod-shaped organisms. It is thought to control rod shape through the organization of cell wall synthesis enzymes. It is therefore quite interesting that MreB is found in wall-less organisms and is essential for their shape. *Spiroplasma* makes an interesting model organism to study MreB assembly and membrane binding as it allows one to ignore cell wall synthesis.

A more detailed discussion on how these results mesh with results from walled organisms would be useful and help expand the interest of this study to a wider audience. For example, the authors reference the molecular dynamic simulations done by Colavin et al but do not really explain how the author's results help to interpret or modify the computational results.

Authors' response:

We have rewritten the text in the discussion to make the interpretation more clear in the context of our results, in the revised submission of the manuscript. To quote:

“While there are theoretical models on how this might be achieved (Wong et al., 2019), our study based on ScMreB5^{E134A} and nucleotide-dependent liposome binding provides insights into the role of ATP-driven dynamics in polymerization and membrane binding of MreB. A hypothesis on how different nucleotide states could exhibit different modes of membrane-binding by

twisting of MreB filaments in the presence of a lipid bilayer was earlier put forward based on molecular dynamics simulations (Colavin et al., 2014; Shi et al., 2020). Impairment of membrane binding by ScMreB5^{E134A} indicates that the conformational changes facilitated by Glu134 is required for liposome interaction.”

[page 16, lines 5 - 14]

Or the authors mention RodZ but do not reference Morgenstein et al , or Bratton et al, which examine MreB in cells lacking RodZ and therefore MreB-wall communication. I believe there will be interest for those who study MreB or cell shape, as well as Spiroplasma

Authors' response:

We have rewritten the text in the discussion to make the interpretation more clear in the context of our results, in the revised submission of the manuscript. To quote:

“Interestingly, RodZ plays an important role in circumferential movement of MreB by linking with the peptidoglycan synthesis machinery, and also in curvature dependent localization of MreB (Morgenstein et al., 2015; Bratton et al., 2018). In the absence of RodZ and peptidoglycan synthesis in *Spiroplasma*, a novel mode of membrane binding involving an increased surface might be important for curvature sensing.”

[page 17, line 24 – page 18, line 3]

We thank the reviewer for the suggestions for improvement. We have included these suggestions and rewritten the discussion accordingly in the revised manuscript. We hope that the revisions will address the concerns of the reviewer.

Reviewer #3 (Evidence, reproducibility and clarity (Required)):

****Summary:****

The manuscript "Filament dynamics driven by ATP hydrolysis modulates membrane binding of the bacterial actin MreB" gives several important and novel insights into the mechanism of polymerization of MreB polymerizes onto membranes. This work examine how the nucleotide state of the polymer affects this association. To gain these insights, this work uses structural biology, liposome association assays, and expression of MreB inside orthologous (eukaryotic) hosts to gain insights into the underlying mechanism of these membrane-associated polymers. While these studies were conducted on a MreB from a cell wall-less bacteria, I expect many of the observations found in this work to be applicable to other MreB filaments examined in more standard model systems.

To highlight some of the (many) findings in this work:

1) This work remedies an important, long-standing deficit in the MreB field, how the nucleotide affects MreBs ability to polymerize onto membranes. Not only do they demonstrate that ScMreB5 filaments assemble independent of ATP hydrolysis, but they also show that the ability of MreB to hydrolyze ATP controls the rate of filament formation, the lateral association of filaments, and filament disassembly. This result itself that hydrolysis affects filament disassembly is a giant leap forward in our understanding of this polymer, one that will influence many future studies of MreB in many organisms.

2) This careful structural work finally nails down the role of the E134 residue in attacking the gamma phosphate on ATP, a long-standing hypothesis in the actin field, one that was not testable with eukaryotic actin. Furthermore, they very nicely show that this residue serves as an "interaction hub," connecting the nucleotide, catalytic water, and residues from MreB sub-domains, thereby communicating the nucleotide state to the rest of the monomer, and thus affecting filament structure.

3) Unlike the MreB of *E. coli* and *C. crescentus*, the MreB of *Spiroplasma citri* charged surface on one side, and thereby are dependent on the charged nature of the membrane. Similarly, this work examines what residues help this MreB bind to the membrane, as it lacks an N-terminal amphipathic helix. Surprisingly, residues within the "hydrophobic loop" do not appear to be involved in membrane association, an interesting point of data for other groups studying MreB inside gram-positive bacteria.

****Major comments:****

Most of the critical conclusions of the paper are very convincing and well backed by the data, but a few minor points in the discussion require some re-evaluation, as detailed below. Throughout this work, the data and methods are very clear, and this work would be easily reproducible. Likewise, the key experiments are well replicated, including using protein from multiple purifications to validate the lipid-binding assays. There are a few statements in the text and discussion that, in light of past data, invoke questionable models and thus require rephrasing. On some points, showing a bit more data (if data is available) would help bolster their arguments.

1) The authors often state "modulating the membrane curvature." It must be noted

that, thus far, MreB has not been observed to modulate membrane curvature in vivo, and this conclusion might be premature. As little is known about the: A) energetics of MreB binding to membranes, B) rigidity of MreB filaments or the membrane, and C) pressure drop across the membrane, it is not clear if, inside the cell, if 1) MreB filaments deform to the membrane, or 2) if the membrane deforms to the filaments. Notably, no freeze-fracture study of bacterial membranes in *E. coli* or *B. subtilis* has seen any local membrane deformations, so it is likely the filaments deform to the membrane.

Authors' response:

As the reviewer rightly points out, there is no evidence of MreB modulating the membrane curvature in vivo, while there are instances of MreB changing the membrane curvature of liposomes (Hussain, et al, 2018; Salje, et al, 2011). Hence, we have rephrased the relevant sentences throughout in the revised submission of the manuscript, and mentioned about localization to a definite membrane curvature within the cell, and modulation of membrane curvature only in the context of liposomes.

2) In the discussion, they state - "We envisage a mechanism in which bundles of ATP-bound MreB filaments sense an optimal curvature for binding, remodel the membrane and hydrolyze ATP, and then exchange ADP with ATP and bind to the adjacent region with favorable curvature for binding, thus resulting in a processive motion."

This model is very suspect given the existing data in the field, as all experiments thus far indicate MreB processive motion is not driven by polymer dynamics, but the activity of the associated cell wall synthesis enzymes: 1) When MreB motion is halted by antibiotics in *B. subtilis*, little to no filament polymerization dynamics are observed even at long timescales (Domínguez-Escobar et al., 2011)., and 2) GFP-MreB containing the E148A mutation moves around the cell width at the same rate as WT MreB (Garner et al., 2011).

Authors' response:

Though the processive motion is not driven by the polymer dynamics of MreB and their relationship has not been fully established. The experiments carried out with ATPase mutants (E134 equivalent) in MreBs of *Bacillus*, *E. coli* and *Caulobacter* indeed show that there are shape defects. Though the rate of processive movement remains the same, the localization of MreB filaments is probably defective, leading to overall shape defects.

This has been elaborated in the revised submission of the manuscript, as follows:

"We envisage a mechanism in which bundles of ATP-bound MreB filaments sense an optimal curvature for binding, remodel the membrane, which probably gets reinforced by recruiting the peptidoglycan machinery in cell-walled bacteria. Possibly ATP gets hydrolysed within the filaments during the process, and ADP to ATP exchange might also occur. Remodelling might result in a change of curvature, leading to changes in affinity during the cycle. This could drive filaments to an adjacent region with favourable curvature for binding, thus resulting in a processive motion. Though the speed of processive movement of MreB filaments has been demonstrated to be

independent of ATP hydrolysis (Garner et al., 2011), ATPase mutants of MreB possess localization defects in vivo, finally resulting in shape defects as demonstrated for *Bacillus subtilis* (Defeu Soufo and Graumann, 2006) and *Caulobacter crescentus* (Dye et al., 2011). ”

[page 18, lines 7 - 19]

3) In the discussion, the authors state: "Thus, ATP hydrolysis can modulate filament length and bundling, and consequently the orientation of MreB filaments on the cell membrane depending on the curvature." Given they have not yet examined if ScMreB5 is curved when bound to liposomes, much less seen that the filaments orient around rod-shaped cells (in fact, they see the opposite, as noted below), this statement appears to be highly speculative and should be rephrased or removed.

Authors' response:

We agree that this is highly speculative based on the existing data in the manuscript. Hence, we have deleted this sentence from the abstract in the revised submission.

[page 2]

4) Similar to (3) above, the expression of ScMreB5 inside pombe cells gives an unexpected result for MreB, all the images provided suggest that ScMreB5 prefers to orient along the long axis of rods, more similar to what the Gladfelter lab observed with septins (Bridges et al., 2016), rather than what Hussain et al. observed with MreB. While this in no way impacts the findings in the paper, the authors may want to address this discrepancy in the discussion and perhaps revise any statements regarding orientation along curvature to be more cautious given their data.

Authors' response:

As mentioned above in the response to Reviewers 1 and 2, experiments with N-terminal fusion of EcMreB resulted in orientation along the long axis, while a sandwich construct of GFP insertion of EcMreB resulted in a change in the orientation of the filaments. Hence, the data presented in our manuscript does not address the question of orientational preference of the filaments.

The construct used in the yeast experiments may not reflect any of the properties related to orientation along the axis, which could also be due to differences in cell diameter and lipid composition. The mode of binding of ScMreB5 also might contribute to differences in how the filaments are oriented, which is unknown in *Spiroplasma*.

We have mentioned regarding the orientational preference in the revised manuscript text, as follows:

“Although the N-terminal GFP-fusion did not adopt an orientational preference perpendicular to the long axis as observed for *E. coli* or *Bacillus* MreBs in vivo (Domínguez-Escobar et al., 2011; Garner et al., 2011), it was functional for ATP hydrolysis (Table S2). Thus, it was a useful system to observe the effect of the mutation on filament dynamics and bundling.”

[page 10, lines 5 - 9]

5) While this paper nicely demonstrates that 1) ScMreB5 filament structure when bound to flat membranes and 2) that ScMreB5 binds to liposomes, it leaves the reader wondering if ScMreB5 filaments also are curved when bound to deformable liposomes. While not necessary, having EM images of ScMreB5 bound to liposomes would resolve this question and perhaps give further insight into the "long axis" filament alignment seen inside pombe.

Authors' response:

As mentioned above, the N-terminal GFP construct does not provide insights regarding the orientational preference of MreBs.

We have not performed EM experiments with liposomes with the ScMreB5. Orientation of MreB filaments on liposomes will definitely be interesting and informative. Unfortunately, we will not be able to include these experiments in the current manuscript because of restricted access to the electron microscope, especially due to the current pandemic situation when visits and access to national facilities will be available only after an indefinite period. We propose to carry out these experiments in future and include it in a follow-up study focussed on curvature preference/sensing by ScMreB5.

****Minor comments:****

I found the text and figures incredibly clear and concise, and the methods well composed and detailed. If I may offer a few minor corrections and suggestions:

1) The authors state, "Orientation of MreB filaments within cells has been proposed to be dependent on the cell diameter," citing Hussain et al. Importantly, this was not the conclusion of that study, Hussain observed that MreB filaments orient inside rods of any width (even up to 5µM). Rather, the conclusion of that study was not that filaments orient dependent of cell diameter, but rather the "difference between principle curvatures."

Authors' response:

We have rephrased the relevant sentence in the revised submission of the manuscript, as follows:

"Orientation of MreB filaments within cells has been proposed to be dependent on the differences between the principle curvatures, with a more ordered arrangement when the difference is higher as in a narrow rod (Hussain et al., 2018)."

[page 5, lines 12 - 14]

2) The authors state, "Assisting in conformational changes during polymerization is an additional novel role proposed for the catalytic residue, which has always been implicated only in stimulating hydrolysis in most actin family members such as actin,

ParM, and MamK." Given this paper very nicely elucidates the role of E148 in coordinating the water that attacks the gamma phosphate, it seems critical to cite the original study that proposed the equivalent residue in eukaryotic actin is responsible for ATP hydrolysis - "The structure of nonvertebrate actin: Implications for the ATP hydrolytic mechanism" by Vorobiev (Vorobiev et al., 2003)

Authors' response:

This reference has been included in the revised manuscript as below:

"We propose an additional novel role for the catalytic residue Glu134, which has earlier been implicated mostly in stimulating hydrolysis in most actin family members such as actin (Vorobiev et al., 2003), ParM (Gayathri et al., 2013) and MamK (Löwe et al., 2016)."

[page 6, lines 1 – 3]

We had cited the relevant reference in the original manuscript in the Results section while mentioning the role of Glu134 equivalent residues.

"However, Glu134 and/or Thr161 might interact with the catalytic water, a hypothesis based on structure superpositions with other MreBs and actin structures (Fig. 1, G and H; and Fig. 2, A and B) (van den Ent et al., 2014; Vorobiev et al., 2003; Merino et al., 2018)."

[page 8, lines 5 – 8]

3) It might help assist the reader in understanding the experiments in 4E-F if the authors added a small note to the legend or text that these experiments had a "pre-clearing" step (spinning the protein alone to remove any aggregates). Currently, this important detail is only mentioned in the methods.

Authors' response:

We have included this detail in the revised submission of the manuscript in the results section also. To quote:

"Prior to the addition of liposomes, the protein samples were spun at 21,500 g to ensure that any protein aggregates were removed. Pelleting assays of the reaction mix without liposomes served as negative controls for the liposome-binding experiments (Fig. S4 A). The control runs showed that the protein does not pellet on its own in the absence of liposome, irrespective of its polymerization state."

[page 12, lines 12 – 17]

Reviewer #3 (Significance (Required)):

****Section B.****

While MreB has been studied in different bacteria for over 20 years, the field, thus far has lacked an understanding of how the different nucleotide states of MreB polymer affect its dynamics and membrane association. This work gives large advances in our understanding of how the associated nucleotide not only affects membrane binding but also filament dynamics. Impressively, this work approaches these problems using elegant structural studies, identifying key residues involved not only in nucleotide hydrolysis but how these residues communicate the identity of the

bound nucleotide to the rest of the filament, affecting how filaments associate with the membrane. As noted above, while these studies were conducted on a MreB from a cell wall-less bacteria, I expect many of the observations found in this work to be applicable to other MreB filaments examined in more standard model systems. Generally, I expect this work to be of interest to not only the bacterial cell biology field but also researchers in the eukaryotic actin community.

My expertise lies in bacterial cell biology and biophysics. Specifically, I focus on studying bacterial polymers, examining not only their polymerization dynamics in vitro but also their in vivo motions and how these dynamics affect their associated biological function inside the cell.

****Referee Cross-commenting****

I think this is a great paper, and have no huge issues (save the need to rephrase a couple of factual errors in citing other papers).

Authors' response:

We thank the reviewer for the suggestions for improvement. We hope that the revisions incorporated will address the concerns raised by the reviewer.

December 16, 2021

Re: JCB manuscript #202106092R

Dr. Pananghat Gayathri
Indian Institute of Science and Research
Pune 411008
India

Dear Dr. Gayathri,

Thank you for submitting your revised manuscript entitled "Filament dynamics driven by ATP hydrolysis modulates membrane binding of the bacterial actin MreB" to the Journal of Cell Biology. The manuscript has now been assessed by the original three expert reviewers from Review Commons, whose reports are appended below. As you can see from the reviews, they are positive but request some additional revisions.

Please consider carefully the following points: Are the title, conclusions about ATP hydrolysis driving a switch leading to disassembly, results not due to possible loss of magnesium coordination, and other points of data interpretation identified by the expert reviewers fully proven by the data presented? Besides directly answering the concerns, we note that terms such as "suggesting" or "indicating" or especially "consistent with" are much safer. Please also clarify that the light-scattering results are not just due to concentration.

Our general policy is that papers are considered through only one revision cycle; however, given that the suggested changes are relatively minor we are open to one additional short round of revision. Please submit the final revision along with a cover letter that includes a point by point response to the remaining reviewer comments.

Thank you for your interest in the Journal of Cell Biology. We look forward to receiving a revised manuscript from you, which will be given a final review to determine whether the concerns are resolved within reasonable limits. You can contact me or the scientific editor listed below at the journal office with any questions, cellbio@rockefeller.edu or call (212) 327-8588.

With kind regards,

Kenneth Yamada, MD, PhD
Editor
Journal of Cell Biology

Dan Simon, PhD
Scientific Editor
Journal of Cell Biology

Reviewer #1 (Comments to the Authors (Required)):

I have previously reviewed this manuscript. With the added control experiments and discussion provided, the authors convincingly addressed most of the major concerns that were initially raised. However, several points should either be addressed or toned-down before the manuscript could be considered for publication. There are two main concerns:

- 1) Claims made regarding the role of hydrolysis in filament disassembly
- 2) Proving the role of hydrolysis in membrane binding

- Although MreB binding to the membrane could be modulated by filament dynamics driven by ATP hydrolysis, the data does not prove it. The title is therefore exaggerated. The data with the sedimentation assays suggests a plausible new mode of membrane binding involving a charge specific interaction and provides some hints on nucleotide dependency on membrane binding but still does not prove the role of hydrolysis and filament dynamics in this phenomenon of membrane binding.

- The conclusions from the in vivo experiments on filament disassembly, fragmentation and reannealing remain unclear and the different statement suggested by the authors related to this phenomenon are not supported by the experimental data.

The author state:

"Our results emphasize a crucial role for ATP hydrolysis in driving a conformational switch leading to filament dissociation and disassembly."

The experiment showing filament disassembly are not enough to state such a strong hypothesis, especially given that disassembling of the filament was seen in a limited number of cells (4 out of 43 cells). Considering that MreB was expressed in a

heterologous another system, these events could be just artefacts or maybe due to the membrane binding deficiency generated by the E134 mutant.

"The electron microscopy images also appeared to suggest decreased polymerization and bundling for ScMreB5WT in the presence of ADP and for ScMreB5E134A irrespective of the nucleotide present"

The authors must rephrase the sentence to explain what they mean by decreased polymerization and bundling? Without proper quantification this statement is currently unjustified.

OTHER COMMENTS:

- Supplementary Figure 3 D-E:

Is it possible that the polymerization difference observed between the MreB5wt and MreBk57A in the light scattering data are due to the concentration difference used to perform this experiment (35 μ M for MreBwt and 5.1 μ M for MreBK57A)?

- Figure 3D :

Since the Glu 134 is implicated in magnesium coordination and since magnesium is crucial for MreB polymerization, is it possible that the decrease in filament bundling observed in cryo-EM is due to a defect in magnesium coordination generated by the E134 mutant leading to a defect bundling rather than a defective bundling per se?

- Figure 3 C-D:

1) The filament seems to be oriented in the same direction for WT+ATP and wt+AMP-PNP, and different orientation of the filaments seems to occur in wt+ADP or E134A+AMP-PNP. Could the author comment on this? Could it be due to higher filament crowding in the first two cases?

2) The author state that the few sheet-like bundles observed might be due to "lower filament density or defective bundling or both".

Why did the author retract the first hypothesis and did not explore this possibility?

It may be inappropriate to use the term "defective bundling". More precisely, it's the spatial organization of the filaments which is affected, in the wt, long bundles and short bundles in the E134 mutant. Also one should note that the rigidity of the filament could be affected in the mutant leading to a difference in bundling

- Figure 5 A:

Could the author precise in the text what happens at $t=0s$? how did they induce polymerization and why it takes 26 mn for MreBwt to start polymerizing filaments in the fission yeast.

Supplementary Figure 2:

Could the author include the % of a.a. sequence homology and identity between MreB1/2/3/4/5 and the different MreBs from different species. It is not straightforward to see how close the different proteins are at the aa sequence level.

Supplementary Figure 3B:

Could the authors add the molecular standards above the pic

Supplementary Figure 4:

Could the author remove the added "E"

Reviewer #2 (Comments to the Authors (Required)):

This paper is very well written, with the experiments laid out in a clear and organized manner. The authors have addressed most my comments.

In Fig 5 Do filaments reform in cells that were observed to initially have filaments and have those filaments disassemble?

The authors postulate that MreBE143A requires a higher concentration to polymerize. I would like thank them for doing Western blots on these cells to compare expression levels to cells expression WT MreB. These blots show that on the population level there are no changes in expression. The authors could still test that filaments in individual cells are only formed if the MreB concentration passes a critical threshold by measuring the fluorescent intensity in each cell and correlating this to filament formation. This information would be very informative in the model.

Fig. 7E Why does the double mutant (I95A/W96A) bind with a similar affinity to WT when I95A binds worse. Also as there are 2 gels in D, E should be separated into 2 graphs. As of now it is unclear where the WT fraction comes from, gel 1, gel 2, or the average of the cell. To control for difference in the gels E143A should be graphed with its own WT.

Line 13-15 page 14: Neither ADP or ATP cause binding. These lines are confusing as it sounds like there is a difference "differential binding...based on nucleotide state". I think this could be clarified.

The line numbers are not continuous but...

Minor

Line 3 page 7: same should be similar
Figure 3. The white lines are not visible when printed

Reviewer #3 (Comments to the Authors (Required)):

The authors have addressed most of my concerns, and I feel the paper is of sufficient quality for publication, but there is one issue I still feel requires rectification, detailed below.

Regarding the point that hydrolysis might lead to processive motion. The authors state - "We envisage a mechanism in which bundles of ATP-bound MreB filaments sense an optimal curvature for binding, remodel the membrane, which probably gets reinforced by recruiting the peptidoglycan machinery in cell-walled bacteria. Possibly ATP gets hydrolysed within the filaments during the process, and ADP to ATP exchange might also occur. Remodelling might result in a change of curvature, leading to changes in affinity during the cycle. This could drive filaments to an adjacent region with favourable curvature for binding, thus resulting in a processive motion. Though the speed of processive movement of MreB filaments has been demonstrated to be independent of ATP hydrolysis (Garner et al., 2011), ATPase mutants of MreB possess localization defects in vivo, finally resulting in shape defects as demonstrated for *Bacillus subtilis* (Defeu Soufo and Graumann, 2006) and *Caulobacter crescentus* (Dye et al., 2011)."

A. The authors note that 1) hydrolysis dead mutants show no difference in speed, yet 2) Lead to shape defects. These two points are not in opposition, most especially with the insights provided in this paper - they nicely demonstrate that hydrolysis affects the lateral association of filaments. Thus, hydrolysis defective MreB, inducing more bundling of MreB filaments leads to filaments "clumping" in given areas of the cell (as seen in *Caulobacter* by Dye, and *bacillus* by Garner), causing shape defects as the regions the filaments are bundled would become thinner, while the rest of the cell has less filaments, and would become fatter. Similar observations in the supplemental data in the Garner paper, who I think noted that whether the thin regions of *Caulobacter* with clumps of MreB mutants reported by dye may not reflect filaments being attracted to curvature, but rather highly localized filaments causing cells to thin at that given place, as they saw in *Bacillus*.

B. It should also be noted that no possible mechanism can be envisioned for processive movement of a filament as it binds and unbinds to a membrane, much less when the filament has no structural or kinetic polarity.

C. in regards to "could drive filaments to an adjacent region with favourable curvature for binding" I suggest the authors look closely at the Wang 2019 paper in eLife, which, combined with the work from Hussain, indicates the "localization" of MreB to negative curvature viewed in snapshots is not driven by a preferential "localization", but rather that MreB filaments, as they move around the cell, reorient to curvatures as they move. This constant movement, and reorientation, causes on average, filaments to moving through negative curvature more, and positive curvature less. Basically, MreB is never "localized", as the filaments are always processively moving, and thus "localization" is technically an incorrect term.

D. it must be noted that no group has seen a difference of MreB speeds in cells with different curvatures.

E. Finally, the authors should note that NO processive movement is seen when cell wall synthesis is stopped, even in high resolution examinations of filaments. Thus, the idea of membrane binding driving processive motion is purely speculative, and importantly, goes against all evidence in the field.

Response to the Editor's comments:

Are the title, conclusions about ATP hydrolysis driving a switch leading to disassembly, results not due to possible loss of magnesium coordination, and other points of data interpretation identified by the expert reviewers fully proven by the data presented?

Authors' response:

The title has been rephrased in the revised version as follows:

"Filament organization of the bacterial actin MreB is dependent on the nucleotide state"

We have rephrased the references to hydrolysis leading to disassembly in the manuscript text.

Loss of magnesium coordination leads to a complete loss of hydrolysis. Hence, the effects due to loss of Mg^{2+} coordination alone are not easily discernible. We have therefore not implicated a separate role for magnesium coordination, but indicated the effect as a loss of a network of interactions, which is supported by the structural data.

A point-by-point response to the reviewer's concerns is given below, which explains the above points further.

Besides directly answering the concerns, we note that terms such as "suggesting" or "indicating" or especially "consistent with" are much safer.

Authors' response:

These changes have been included in the manuscript. Specific instances of changes have been elaborated in the point-by-point response below.

Please also clarify that the light-scattering results are not just due to concentration.

Authors' response:

We have limited our interpretation of the light-scattering experiment to suggest that both the wild type MreB and K57A mutant exhibit polymerization. We base this interpretation on the increase in light scattering seen with the K57A mutant, which suggests polymerization despite the experiment being conducted at a lower concentration compared to that of the wild type. Thus, our interpretations and the conclusion are not limited by the concentrations at which the experiments were conducted.

We have now revised the relevant sentence in the text such that this aspect of the experiment is clear to the readers. To quote:

"Light scattering measurements, although performed at a lower concentration of protein compared to the wild type, show that ScMreB5^{K57A} might indeed polymerize (Fig. S3, D and E)."

[page 9, lines 5 - 7]

To remove ambiguity for the readers, we also deleted the reference to a decrease in polymerized content. To quote:

“This could lead to a sub-optimal conformation of the ATPase active site within the polymers, thereby leading to a decrease in ATPase activity.”

[page 9, lines 9 - 11]

Reviewer #1 (Comments to the Authors (Required)):

I have previously reviewed this manuscript. With the added control experiments and discussion provided, the authors convincingly addressed most of the major concerns that were initially raised. However, several points should either be addressed or toned-down before the manuscript could be considered for publication. There are two main concerns:

- 1) Claims made regarding the role of hydrolysis in filament disassembly
- 2) Proving the role of hydrolysis in membrane binding

- Although MreB binding to the membrane could be modulated by filament dynamics driven by ATP hydrolysis, the data does not prove it. The title is therefore exaggerated.

Authors' response:

We have modified the title to include only the effect of nucleotide dependency and ATP hydrolysis is not part of the title. The revised title is as follows:

““Filament organization of the bacterial actin MreB is dependent on the nucleotide state”

The data with the sedimentation assays suggests a plausible new mode of membrane binding involving a charge specific interaction and provides some hints on nucleotide dependency on membrane binding but still does not prove the role of hydrolysis and filament dynamics in this phenomenon of membrane binding.

Authors' response:

We have edited references to filament dynamics and role of hydrolysis, and restricted the text to suggest nucleotide dependence on membrane binding but not ATP hydrolysis. All references to the role of ATP hydrolysis has been toned down as suggestions, and not as conclusions.

To quote from the revised text:

“Hence, the catalytic glutamate functions as a switch – i) by sensing the ATP-bound state for filament assembly, and ii) by assisting hydrolysis, thereby potentially triggering disassembly as observed in other actins.”

[page 2, lines 10 – 13]

“Although filament stabilization is a characteristic feature of ATPase defective mutants of the actin family, we cannot completely rule out that disassembly events were not observed in ScMreB5^{E134A} due to experimental artefacts.”

[page 12, lines 10 – 14]

“our study based on ScMreB5^{E134A} and nucleotide-dependence of liposome binding is indicative of the role of ATP-driven dynamics in polymerization and membrane binding of MreB.”

[page 16, lines 22 -24]

The conclusions from the in vivo experiments on filament disassembly, fragmentation and reannealing remain unclear and the different statement suggested by the authors related to this phenomenon are not supported by the experimental data.

The author state:

"Our results emphasize a crucial role for ATP hydrolysis in driving a conformational switch leading to filament dissociation and disassembly."

Authors' response:

We have deleted the above sentence from the discussion section of the manuscript (deleted from page 18, last paragraph).

The experiment showing filament disassembly are not enough to state such a strong hypothesis, especially given that disassembling of the filament was seen in a limited number of cells (4 out of 43 cells). Considering that MreB was expressed in a heterologous another system, these events could be just artefacts or maybe due to the membrane binding deficiency generated by the E134 mutant.

Authors' response:

We have rephrased the reference to the role of ATP hydrolysis as follows:

“Although filament stabilization is a characteristic feature of ATPase defective mutants of the actin family, we cannot completely rule out that disassembly events were not observed in ScMreB5^{E134A} due to experimental artefacts.”

[page 12, lines 10 – 14]

"The electron microscopy images also appeared to suggest decreased polymerization and bundling for ScMreB5WT in the presence of ADP and for ScMreB5E134A irrespective of the nucleotide present"

The authors must rephrase the sentence to explain what they mean by decreased polymerization and bundling? Without proper quantification this statement is currently unjustified.

Authors' response:

This sentence has been deleted in the revised version of the manuscript since we do not make any quantitative comparisons based on the electron micrographs in the manuscript. Further quantification has been included for the yeast microscopy data (**new figure panels Fig. 4E and 4G**) to justify the statement.

OTHER COMMENTS:

- Supplementary Figure 3 D-E:

Is it possible that the polymerization difference observed between the MreB5wt and MreBk57A

in the light scattering data are due to the concentration difference used to perform this experiment (35 μM for MreBwt and 5.1 μM for MreBK57A)?

Authors' response:

We have limited our interpretation of the light-scattering experiment to suggest that both the wild type MreB5 and K57A mutant exhibit polymerization. We base this interpretation on the increase in light scattering seen with the K57A mutant, which suggests polymerization despite the experiment being conducted at a lower concentration compared to that of the wild type. Thus, our interpretations and the conclusion are not limited by the concentrations at which the experiments were conducted.

We have now revised the relevant sentence in the text such that this aspect of the experiment is clear to the readers. To quote:

“Light scattering measurements, although performed at a lower concentration of protein compared to the wild type, show that ScMreB5^{K57A} might indeed polymerize (Fig. S3, D and E).”

[page 9, lines 5 - 7]

To remove ambiguity for the readers, we also deleted the reference to a decrease in polymerized content. To quote:

“This could lead to a sub-optimal conformation of the ATPase active site within the polymers, thereby leading to a decrease in ATPase activity.”

[page 9, lines 9 - 11]

- Figure 3D :

Since the Glu 134 is implicated in magnesium coordination and since magnesium is crucial for MreB polymerization, is it possible that the decrease in filament bundling observed in cryo-EM is due to a defect in magnesium coordination generated by the E134 mutant leading to a defect bundling rather than a defective bundling per se?

Authors' response:

Based on our structural analysis, E134 coordinates the catalytic water, interacts with the gamma-phosphate of ATP, and is within hydrogen-bonding distance to one of the waters coordinated to Mg^{2+} . Hence, we have proposed that E134 functions as an interaction hub for driving conformational changes mediated by sensing the presence of gamma phosphate. The interaction hub may not exist if any of the links are disrupted such as in the absence of the gamma phosphate (ADP-bound state) or in the absence of Mg^{2+} , as evidenced by the different stages of the conformational cycle shown in Figure 8.

Since it is not possible to segregate between the role of E134 in coordinating the catalytic water vs interaction with the Mg^{2+} coordinating water molecule, we have not delineated these differences in the manuscript text and omitted any reference to quantitative comparisons based on the electron micrographs (see point above).

- Figure 3 C-D:

1) The filament seems to be oriented in the same direction for WT+ATP and wt+AMP-PNP, and

different orientation of the filaments seems to occur in wt+ADP or E134A+AMP-PNP. Could the author comment on this? Could it be due to higher filament crowding in the first two cases?

Authors' response:

The images shown are representative sections of multiple micrographs for each sample. The different orientation of filaments in WT+ADP and E134A+AMPPNP could be due the smaller length of the filaments and the less density of filaments in these samples compared to WT+ATP and WT+AMP-PNP. However, we have not quantified the density or lengths because the ends cannot be unambiguously identified for all the filaments. Hence, instead of further quantification of the EM images, we have proceeded with characterizing the filament organization using yeast expression.

2) The author state that the few sheet-like bundles observed might be due to "lower filament density or defective bundling or both".

Why did the author retract the first hypothesis and did not explore this possibility?

Authors' response:

We have not retracted the first hypothesis. However, since we have not quantified the electron micrographs further, we had addressed these possibilities using the heterologous yeast expression. We have now included a graph (Fig. 4 E) showing that the density (polymer content per unit area of the cell) of filaments is not significantly different for ScMreB^{WT} and the ATPase mutant E134A.

This point is now included in the text, as follows:

"However, the density of filaments (MreB polymer content per unit area of the cell) was not significantly different (Fig. 4 E)."

[page 10, lines 15 – 16]

We have also mentioned about the fluorescence intensities for the threshold for polymerization as suggested by Reviewer 2 in the revised version of the manuscript (Fig. 4 G).

It may be inappropriate to use the term "defective bundling". More precisely, it's the spatial organization of the filaments which is affected, in the wt, long bundles and short bundles in the E134 mutant. Also one should note that the rigidity of the filament could be affected in the mutant leading to a difference in bundling.

Authors' response:

We have rephrased the term 'defective bundling' to 'differences in filament orientation or organization' in the manuscript.

To quote from the revised text:

"However, unlike ScMreB5^{WT} filaments, the spatial organization of ScMreB5^{E134A} filaments appeared to be different in yeast cells (Fig. 4 A). Differences in organization of filaments were more clearly visible by super-resolution imaging (3D-SIM) of ScMreB5 filaments (Fig. 4 B and Video 1)."

[page 10, lines 12 – 16]

“Quantification of the spatial organization, by measuring anisotropy using FibrilTool (Boudaoud et al., 2014) and coefficient of variation (Higaki et al., 2020) which is an indicator of cytoskeleton bundling, further confirmed that ScMreB5^{E134A} exhibited differences in lateral association and organization of filaments (Fig. 4, C and D).”

[page 10, lines 16 – 19]

“Time-lapse imaging of polymerization of ScMreB5^{WT} and ScMreB5^{E134A} within yeast cells confirmed that polymerization and lateral association of filaments were efficient in ScMreB5^{WT} compared to ScMreB5^{E134A} (Fig. 5 A and Video 2).”

[page 11, lines 3 – 5]

“Lateral association of filaments in ScMreB5^{WT} was often promoted by cell septation as the ingressing septa brought the filaments in close proximity (25 out of 43 cells). The difference in the spatial organization of ScMreB5^{E134A} filaments was clearly seen in yeast cells undergoing cell division (Fig. 5 E and Video 3).”

[page 12, lines 1 – 4]

“It is possible that the filament conformations, spatial orientations or bundling features of the different nucleotide states can either i) match the curvature of the liposomes, ii) remodel the liposomes to match the filament curvature or, iii) fall off in case of a curvature mismatch.”

[page 17, lines 16 – 19]

We have refrained from commenting on rigidity of the filaments because we have not quantified features pertaining to rigidity.

- Figure 5 A:

Could the author precise in the text what happens at t=0s? how did they induce polymerization and why it takes 26 mn for MreBwt to start polymerizing filaments in the fission yeast.

Authors' response:

While this information was briefly included in the Materials and Methods section earlier, we have added the following statements for clarity in the text and figure legends.

“To visualize initiation of polymerization, cells were grown in the absence of thiamine for upto 28 – 30 hours, placed on agarose pads and random fields with cells exhibiting diffuse fluorescence were imaged. The time at which the cells were placed on agarose pads and first imaged was taken as t = 0. Polymerization happens spontaneously presumably within the cells that have sufficient monomers beyond the critical concentration for polymerization.”

[page 11, lines 5 – 12]

“Cells were grown for 28 – 30 hours in the absence of thiamine, placed on an agarose pad as mentioned in Methods section and imaged at every 3 minutes time-interval. The time at which the cells were placed on agarose pads and first imaged was taken as t = 0.”

[page 46, lines 21 – 24]

Supplementary Figure 2:

Could the author include the % of a.a. sequence homology and identity between

MreB1/2/3/4/5 and the different MreBs from different species. It is not straightforward to see how close the different proteins are at the aa sequence level.

Authors' response:

In the revised manuscript, we have included the % of sequence identity of all the sequences included in the multiple sequence alignment, with respect to ScMreB5 at the end of the alignment in the revised Supplementary Figure 2.

Supplementary Figure 3B:

Could the authors add the molecular standards above the pic

Authors' response:

We have included the molecular standards in Supplementary Figure 3B as insets to panels B and C.

Supplementary Figure 4:

Could the author remove the added "E"

Authors' response:

We have deleted the extra 'E'.

Reviewer #2 (Comments to the Authors (Required)):

This paper is very well written, with the experiments laid out in a clear and organized manner. The authors have addressed most my comments.

In Fig 5 Do filaments reform in cells that were observed to initially have filaments and have those filaments disassemble?

Authors' response:

We have not observed filaments reforming in these cells in the duration of imaging. However, we cannot comment on the same as we have images for only for a short duration (10 – 20 minutes) after disassembly. Thus, it is possible that we might not have imaged those cells long enough to observe reassembly of filaments after disassembly. Hence, we have not commented on the same in the manuscript.

The authors postulate that MreBE143A requires a higher concentration to polymerize. I would like thank them for doing Western blots on these cells to compare expression levels to cells expression WT MreB. These blots show that on the population level there are no changes in expression. The authors could still test that filaments in individual cells are only formed if the

MreB concentration passes a critical threshold by measuring the fluorescent intensity in each cell and correlating this to filament formation. This information would be very informative in the model.

Authors' response:

We have quantified this information by quantifying the fluorescence intensity in cells with diffused fluorescence and included a plot of the mean fluorescence intensity as a figure panel in Figure 4G. This data suggests that the critical concentration of E134A mutant might be slightly higher than that of the wild type. We have also mentioned this in the text in the revised manuscript, as follows:

“A quantification of the fluorescence intensity in cells with diffused fluorescence indicated that the average fluorescence intensity for ScMreB5^{WT} is slightly lower than that of ScMreB5^{E134A}, suggestive of a lower critical concentration for the wild type (Fig. 4, F and G).”

[page 10, 23 – 24]

Fig. 7E Why does the double mutant (I95A/W96A) bind with a similar affinity to WT when I95A binds worse.

Authors' response:

Assuming that all the mutants are folded and since they do not show evident or measurable differences in their tendency to aggregate, this anomaly could be explained if the side chain orientations of I95 and W96 together might have a steric role that could negatively affect membrane binding. This might explain why mutation of a single residue (I95A) could lead to a decrease in binding, but a double mutant shows similar binding to that of wild type. Mutation of I95 in this loop might affect the loop conformation leading to a negative effect on membrane binding.

In the AMPPNP-bound structure, W96 in the loop is facing inwards, and is surrounded by Ile and Leu residues, while in the ADP-bound structure, the loop is disordered. However, the role of crystal packing for the observed differences cannot be ruled out. Hence, we have not commented on this observation since any explanation provided will be speculation.

Also as there are 2 gels in D, E should be separated into 2 graphs. As of now it is unclear where the WT fraction comes from, gel 1, gel 2, or the average of the cell. To control for difference in the gels E143A should be graphed with its own WT.

Authors' response:

WT and E134A samples have been repeated in multiple gels, and the values averaged for plotting the graphs. Hence, the graph represents values from multiple gels to compare statistically, and a single graph corresponding to the data is shown.

Line 13-15 page 14: Neither ADP or ATP cause binding. These lines are confusing as it sounds like there is a difference "differential binding...based on nucleotide state". I think this could be clarified.

Authors' response:

The following clarification has been included in the revised manuscript:

“there is a differential binding for ScMreB5^{WT} based on the nucleotide state (in the presence of ATP or ADP addition compared to AMP-PNP addition or in the absence of any nucleotide; Fig. 7, G and H), similar to the effect of E134A mutation.”

[page 15, lines 4 – 7]

The line numbers are not continuous but...

Minor

Line 3 page 7: same should be similar

Authors' response:

We have changed as suggested.

Figure 3. The white lines are not visible when printed

Authors' response:

The white lines are visible in the pdf versions in our manuscript.

Reviewer #3 (Comments to the Authors (Required)):

The authors have addressed most of my concerns, and I feel the paper is of sufficient quality for publication, but there is one issue I still feel requires rectification, detailed below.

Regarding the point that hydrolysis might lead to processive motion. The authors state - "We envisage a mechanism in which bundles of ATP-bound MreB filaments sense an optimal curvature for binding, remodel the membrane, which probably gets reinforced by recruiting the peptidoglycan machinery in cell-walled bacteria. Possibly ATP gets hydrolysed within the filaments during the process, and ADP to ATP exchange might also occur. Remodelling might result in a change of curvature, leading to changes in affinity during the cycle. This could drive filaments to an adjacent region with favourable curvature for binding, thus resulting in a processive motion. Though the speed of processive movement of MreB filaments has been demonstrated to be independent of ATP hydrolysis (Garner et al., 2011), ATPase mutants of MreB possess localization defects in vivo, finally resulting in shape defects as demonstrated for *Bacillus subtilis* (Defeu Soufo and Graumann, 2006) and *Caulobacter crescentus* (Dye et al., 2011). "

Authors' response:

We have deleted these sentences in the revised version of our manuscript. The paragraph in the revised version is as follows:

“Our studies are suggestive of an allosteric effect of ATP binding and hydrolysis for efficient filament formation and membrane binding. The speed of processive movement of MreB filaments has been demonstrated to be independent of ATP hydrolysis (Garner et al., 2011). ATPase mutants of MreB possess localization defects in

vivo, with highly localized filaments in certain areas of the cell, finally resulting in shape defects as demonstrated for *Bacillus subtilis* (Defeu Soufo and Graumann, 2006) and *Caulobacter crescentus* (Dye et al., 2011). “

[page 18, line 23 – page 19, line 3]

A. The authors note that 1) hydrolysis dead mutants show no difference in speed, yet 2) Lead to shape defects. These two points are not in opposition, most especially with the insights provided in this paper - they nicely demonstrate that hydrolysis affects the lateral association of filaments. Thus, hydrolysis defective MreB, inducing more bundling of MreB filaments leads to filaments "clumping" in given areas of the cell (as seen in *Caulobacter* by Dye, and *Bacillus* by Garner), causing shape defects as the regions the filaments are bundled would become thinner, while the rest of the cell has less filaments, and would become fatter. Similar observations in the supplemental data in the Garner paper, who I think noted that whether the thin regions of *Caulobacter* with clumps of MreB mutants reported by dye may not reflect filaments being attracted to curvature, but rather highly localized filaments causing cells to thin at that given place, as they saw in *Bacillus*.

Authors' response:

The sentence has been rewritten as follows to address the concern.

“The speed of processive movement of MreB filaments has been demonstrated to be independent of ATP hydrolysis (Garner et al., 2011). ATPase mutants of MreB possess localization defects in vivo, with highly localized filaments in certain areas of the cell, finally resulting in shape defects as demonstrated for *Bacillus subtilis* (Defeu Soufo and Graumann, 2006) and *Caulobacter crescentus* (Dye et al., 2011).”

[page 18, line 24 – page 19, line 3]

B. It should also be noted that no possible mechanism can be envisioned for processive movement of a filament as it binds and unbinds to a membrane, much less when the filament has no structural or kinetic polarity.

Authors' response:

As suggested by Reviewer #1, since the references to effects of ATP hydrolysis on membrane binding have been deleted, the concerned paragraph has also been deleted from the revised version of the manuscript.

C. in regards to "could drive filaments to an adjacent region with favourable curvature for binding" I suggest the authors look closely at the Wang 2019 paper in eLife, which, combined with the work from Hussain, indicates the "localization" of MreB to negative curvature viewed in snapshots is not driven by a preferential "localization", but rather that MreB filaments, as they move around the cell, reorient to curvatures as they move. This constant movement, and reorientation, causes on average, filaments to moving through negative curvature more, and positive curvature less. Basically, MreB is never "localized", as the filaments are always processively moving, and thus "localization" is technically an incorrect term.

Authors' response:

The sentence has been deleted in the final revised version of the manuscript.

D. it must be noted that no group has seen a difference of MreB speeds in cells with different curvatures.

Authors' response:

We have not mentioned in the manuscript that the speed of MreB movement is dependent on the cell curvature, and references to processive movement and relationship to membrane curvature have been deleted in the revised version.

E. Finally, the authors should note that NO processive movement is seen when cell wall synthesis is stopped, even in high resolution examinations of filaments. Thus, the idea of membrane binding driving processive motion is purely speculative, and importantly, goes against all evidence in the field.

Authors' response:

The sentence and references to membrane binding leading to processive movement has been deleted in the final revised version of the manuscript.

February 3, 2022

RE: JCB Manuscript #202106092RR

Dr. Pananghat Gayathri
Indian Institute of Science Education and Research Pune
Indian Institute of Science and Research
Pune 411008
India

Dear Dr. Gayathri,

Thank you for resubmitting your revised manuscript entitled "Filament organization of the bacterial actin MreB is dependent on the nucleotide state." As you can see from the appended reviews, the expert reviewers now enthusiastically recommended acceptance for publication after attending to some minor text corrections and clarifications. We would be happy to publish your paper in JCB pending final revisions necessary to address these last comments and to meet our formatting guidelines (see details below).

A. MANUSCRIPT ORGANIZATION AND FORMATTING:

- 1) Text limits: Character count for Articles is < 40,000, not including spaces. Count includes title page, abstract, introduction, results, discussion, and acknowledgments. Count does not include materials and methods, figure legends, references, tables, or supplemental legends.
- 2) Figures limits: Articles may have up to 10 main text figures.
- 3) Figure formatting: Scale bars must be present on all microscopy images, including inset magnifications. Molecular weight or nucleic acid size markers must be included on all gel electrophoresis. Please add MW markers to Figure 5D.
- 4) Statistical analysis: Error bars on graphic representations of numerical data must be clearly described in the figure legend. The number of independent data points (n) represented in a graph must be indicated in the legend. Statistical methods should be explained in full in the materials and methods. For figures presenting pooled data the statistical measure should be defined in the figure legends. Please also be sure to indicate the statistical tests used in each of your experiments (both in the figure legend itself and in a separate methods section) as well as the parameters of the test (for example, if you ran a t-test, please indicate if it was one- or two-sided, etc.). Also, if you used parametric tests, please indicate if the data distribution was tested for normality (and if so, how). If not, you must state something to the effect that "Data distribution was assumed to be normal but this was not formally tested."
- 5) Materials and methods: Should be comprehensive and not simply reference a previous publication for details on how an experiment was performed. Please provide full descriptions (at least in brief) in the text for readers who may not have access to referenced manuscripts. The text should not refer to methods "...as previously described."
- 6) For all cell lines, vectors, constructs/cDNAs, etc. - all genetic material: please include database / vendor ID (e.g., Addgene, ATCC, etc.) or if unavailable, please briefly describe their basic genetic features, even if described in other published work or gifted to you by other investigators (and provide references where appropriate). Please be sure to provide the sequences for all of your oligos: primers, si/shRNA, RNAi, gRNAs, etc. in the materials and methods. You must also indicate in the methods the source, species, and catalog numbers/vendor identifiers (where appropriate) for all of your antibodies, including secondary. If antibodies are not commercial please add a reference citation if possible.
- 7) Microscope image acquisition: The following information must be provided about the acquisition and processing of images:
 - a. Make and model of microscope
 - b. Type, magnification, and numerical aperture of the objective lenses
 - c. Temperature
 - d. Imaging medium
 - e. Fluorochromes
 - f. Camera make and model

g. Acquisition software

h. Any software used for image processing subsequent to data acquisition. Please include details and types of operations involved (e.g., type of deconvolution, 3D reconstitutions, surface or volume rendering, gamma adjustments, etc.).

10) Supplemental materials: There are strict limits on the allowable amount of supplemental data. Articles may have up to 5 supplemental figures and 10 videos.

Please also note that tables, like figures, should be provided as individual, editable files. A summary of all supplemental material should appear at the end of the Materials and methods section. Please include one brief sentence per item.

11) Video legends: Should describe what is being shown, the cell type or tissue being viewed (including relevant cell treatments, concentration and duration, or transfection), the imaging method (e.g., time-lapse epifluorescence microscopy), what each color represents, how often frames were collected, the frames/second display rate, and the number of any figure that has related video stills or images.

12) eTOC summary: A ~40-50 word summary that describes the context and significance of the findings for a general readership should be included on the title page. The statement should be written in the present tense and refer to the work in the third person. It should begin with "First author name(s) et al..." to match our preferred style.

13) Conflict of interest statement: JCB requires inclusion of a statement in the acknowledgements regarding competing financial interests. If no competing financial interests exist, please include the following statement: "The authors declare no competing financial interests." If competing interests are declared, please follow your statement of these competing interests with the following statement: "The authors declare no further competing financial interests."

14) A separate author contribution section is required following the Acknowledgments in all research manuscripts. All authors should be mentioned and designated by their first and middle initials and full surnames. We encourage use of the CRediT nomenclature (<https://casrai.org/credit/>).

15) ORCID IDs: ORCID IDs are unique identifiers allowing researchers to create a record of their various scholarly contributions in a single place. At resubmission of your final files, please consider providing an ORCID ID for as many contributing authors as possible.

16) Please note that JCB now requires authors to submit Source Data used to generate figures containing gels and Western blots with all revised manuscripts. This Source Data consists of fully uncropped and unprocessed images for each gel/blot displayed in the main and supplemental figures. Since your paper includes cropped gel and/or blot images, please be sure to provide one Source Data file for each figure that contains gels and/or blots along with your revised manuscript files. File names for Source Data figures should be alphanumeric without any spaces or special characters (i.e., SourceDataF#, where F# refers to the associated main figure number or SourceDataFS# for those associated with Supplementary figures). The lanes of the gels/blots should be labeled as they are in the associated figure, the place where cropping was applied should be marked (with a box), and molecular weight/size standards should be labeled wherever possible.

B. FINAL FILES:

**It is JCB policy that if requested, original data images must be made available to the editors. Failure to provide original images

upon request will result in unavoidable delays in publication. Please ensure that you have access to all original data images prior to final submission.**

Thank you for your attention to these final processing requirements. We look forward to receiving a final manuscript from you within the next 1-2 weeks so that we can proceed to publication. If complications arising from measures taken to prevent the spread of COVID-19 will prevent you from meeting this deadline (e.g. if you cannot retrieve necessary files from your laboratory, etc.), please let us know and we can work with you to determine a suitable revision period.

We thank you for submitting this high-quality manuscript to JCB, and we hope you will agree that the rigorous reviewing has resulted in an excellent final work.

With kind regards,

Kenneth M Yamada, MD, PhD
Editor, Journal of Cell Biology

Dan Simon, PhD
Scientific Editor
Journal of Cell Biology

Reviewer #2 (Comments to the Authors (Required)):

The authors have addressed any major issues I had.

Some minor things that I think will improve readability

1) It would nice not to have to find tables or to know the result. Therefore, if the authors could include data in the text that would be useful. Example: pg8 lines 10-20 no Kabs data is given for any mutant nor is there a mention that the mutants are different than WT.

2) page 11 line 6-11. Please indicate the role of thiamine. Also i presume that pads the cells were imaged on had thiamine? please indicate as this is important to understanding the experiment.

3) Is the role of potassium important to mreB function. pg 7 line 10 notes the interesting observation but there is no discussion of the implications or importance.

Reviewer #3 (Comments to the Authors (Required)):

The reviewers have addressed all my concerns, and as long as they satisfy the other reviewers concerns (especially the important points raised of reviewer 1).

1. This paper nicely examines uses a mix of in vivo, in vitro, and structural studies to examine the effects of nucleotide hydrolysis on the assembly of MreB filaments, their lateral association, and disassembly. I do not see any prior work that conflicts with the findings in this paper.

2. The main points of the paper are all well evidenced:

a. The overall data showing that the E134 is a switch, sensing the ATP-bound state for filament assembly and stimulating assisting hydrolysis.

b. Likewise, their experiments indicating the nucleotide state and the E134 residue effects lateral associations and membrane binding also thorough.

c. Finally, their discovery of the effect the charge interactions between MreB and the membrane is a novel advance, and well done.

3. I have one small note that should be fixed, on line 18 they attribute the observation of chromosome segregation by MreB in *Caulobacter* to Dye, but the correct reference should be Z. Gitai 2005.

RE: JCB Manuscript #202106092RR

Dr. Pananghat Gayathri
Indian Institute of Science Education and Research Pune
Indian Institute of Science and Research
Pune 411008
India

Dear Dr. Gayathri,

Thank you for resubmitting your revised manuscript entitled "Filament organization of the bacterial actin MreB is dependent on the nucleotide state." As you can see from the appended reviews, the expert reviewers now enthusiastically recommended acceptance for publication after attending to some minor text corrections and clarifications. We would be happy to publish your paper in JCB pending final revisions necessary to address these last comments and to meet our formatting guidelines (see details below).

A. MANUSCRIPT ORGANIZATION AND FORMATTING:

Full guidelines are available on our Instructions for Authors page, <https://jcb.rupress.org/submission-guidelines#revised>. **Submission of a paper that does not conform to JCB guidelines will delay the acceptance of your manuscript.**

1) Text limits: Character count for Articles is < 40,000, not including spaces. Count includes title page, abstract, introduction, results, discussion, and acknowledgments. Count does not include materials and methods, figure legends, references, tables, or supplemental legends.

- The character count is < 40,000.

2) Figures limits: Articles may have up to 10 main text figures.

- The manuscript has 8 main text figures only.

3) Figure formatting: Scale bars must be present on all microscopy images, including inset magnifications. Molecular weight or nucleic acid size markers must be included on all gel electrophoresis. Please add MW markers to Figure 5D.

- The molecular weights have been indicated in the Figure 5D now. However, the molecular weight marker lane for Figure 5D is not shown in the main figure because this is a Western Blot, and the marker bands are visible only at a different contrast. The source data including the marker lanes have been submitted separately.

4) Statistical analysis: Error bars on graphic representations of numerical data must be clearly described in the figure legend. The number of independent data points (n) represented in a graph must be indicated in the legend. Statistical methods should be explained in full in the materials and methods. For figures presenting pooled data the statistical measure should be defined in the figure legends. Please also be sure

to indicate the statistical tests used in each of your experiments (both in the figure legend itself and in a separate methods section) as well as the parameters of the test (for example, if you ran a t-test, please indicate if it was one- or two-sided, etc.). Also, if you used parametric tests, please indicate if the data distribution was tested for normality (and if so, how). If not, you must state something to the effect that "Data distribution was assumed to be normal but this was not formally tested."

All these have been stated as necessary in the figure legends.

5) Materials and methods: Should be comprehensive and not simply reference a previous publication for details on how an experiment was performed. Please provide full descriptions (at least in brief) in the text for readers who may not have access to referenced manuscripts. The text should not refer to methods "...as previously described."

All methods have been described in the current manuscript, and the original reference cited.

6) For all cell lines, vectors, constructs/cDNAs, etc. - all genetic material: please include database / vendor ID (e.g., Addgene, ATCC, etc.) or if unavailable, please briefly describe their basic genetic features, even if described in other published work or gifted to you by other investigators (and provide references where appropriate). Please be sure to provide the sequences for all of your oligos: primers, si/shRNA, RNAi, gRNAs, etc. in the materials and methods. You must also indicate in the methods the source, species, and catalog numbers/vendor identifiers (where appropriate) for all of your antibodies, including secondary. If antibodies are not commercial please add a reference citation if possible.

All these have been provided as required in the materials and methods or provided as supplementary tables.

7) Microscope image acquisition: The following information must be provided about the acquisition and processing of images:

- a. Make and model of microscope
- b. Type, magnification, and numerical aperture of the objective lenses
- c. Temperature
- d. Imaging medium
- e. Fluorochromes
- f. Camera make and model
- g. Acquisition software
- h. Any software used for image processing subsequent to data acquisition. Please include details and types of operations involved (e.g., type of deconvolution, 3D reconstitutions, surface or volume rendering, gamma adjustments, etc.).

All these have been stated under the materials and methods section.

References have been included in the required format.

10) Supplemental materials: There are strict limits on the allowable amount of supplemental data. Articles may have up to 5 supplemental figures and 10 videos. Please also note that tables, like figures, should be provided as individual, editable

files. A summary of all supplemental material should appear at the end of the Materials and methods section. Please include one brief sentence per item.

A brief sentence on each supplemental material has been included after the Materials and methods section.

11) Video legends: Should describe what is being shown, the cell type or tissue being viewed (including relevant cell treatments, concentration and duration, or transfection), the imaging method (e.g., time-lapse epifluorescence microscopy), what each color represents, how often frames were collected, the frames/second display rate, and the number of any figure that has related video stills or images.

12) eTOC summary: A ~40-50 word summary that describes the context and significance of the findings for a general readership should be included on the title page. The statement should be written in the present tense and refer to the work in the third person. It should begin with "First author name(s) et al..." to match our preferred style.

eTOC summary has been included in the title page in the required format.

13) Conflict of interest statement: JCB requires inclusion of a statement in the acknowledgements regarding competing financial interests. If no competing financial interests exist, please include the following statement: "The authors declare no competing financial interests." If competing interests are declared, please follow your statement of these competing interests with the following statement: "The authors declare no further competing financial interests."

This has been included as required.

14) A separate author contribution section is required following the Acknowledgments in all research manuscripts. All authors should be mentioned and designated by their first and middle initials and full surnames. We encourage use of the CRediT nomenclature (<https://casrai.org/credit/>).

This has been included as required.

15) ORCID IDs: ORCID IDs are unique identifiers allowing researchers to create a record of their various scholarly contributions in a single place. At resubmission of your final files, please consider providing an ORCID ID for as many contributing authors as possible.

16) Please note that JCB now requires authors to submit Source Data used to generate figures containing gels and Western blots with all revised manuscripts. This Source Data consists of fully uncropped and unprocessed images for each gel/blot displayed in the main and supplemental figures. Since your paper includes cropped gel and/or blot images, please be sure to provide one Source Data file for each figure that contains gels and/or blots along with your revised manuscript files. File names for Source Data figures should be alphanumeric without any spaces or special characters (i.e., SourceDataF#, where F# refers to the associated main figure number or SourceDataFS# for those associated with Supplementary figures). The lanes of the gels/blots should be labeled as they are in the associated figure, the place where cropping was applied should be marked (with a box), and molecular weight/size standards should be labeled wherever possible. Source Data files will be made available to reviewers during evaluation of revised manuscripts and, if your paper is eventually published in JCB, the files will be directly

linked to specific figures in the published article.

This has been included as required.

B. FINAL FILES:

Thank you for your attention to these final processing requirements. We look forward to receiving a final manuscript from you within the next 1-2 weeks so that we can proceed to publication. If complications arising from measures taken to prevent the spread of COVID-19 will prevent you from meeting this deadline (e.g. if you cannot retrieve necessary files from your laboratory, etc.), please let us know and we can work with you to determine a suitable revision period.

Please contact the journal office with any questions, cellbio@rockefeller.edu or call [\(212\) 327-8588](tel:(212)327-8588).

We thank you for submitting this high-quality manuscript to JCB, and we hope you will agree that the rigorous reviewing has resulted in an excellent final work.

We are grateful to the Editor and the reviewers for the time taken out for critically evaluating the manuscript, and we agree that it indeed did contribute towards improving the manuscript.

With kind regards,

Kenneth M Yamada, MD, PhD
Editor, Journal of Cell Biology

Dan Simon, PhD
Scientific Editor
Journal of Cell Biology

Reviewer #2 (Comments to the Authors (Required)):

The authors have addressed any major issues I had.

Some minor things that I think will improve readability

1) It would nice not to have to find tables or to know the result. Therefore, if the authors could include data in the text that would be useful. Example: pg8 lines 10-20 no Kabs data is given for any mutant nor is there a mention that the mutants are different than WT.

We have included k_{obs} within brackets in the main text, in addition to the values in the supplementary table. [page 8, lines 18 - 21]

2) page 11 line 6-11. Please indicate the role of thiamine. Also i presume that pads the cells were imaged on had thiamine? please indicate as this is important to understanding the experiment.

This part has been edited as follows to clarify the point:

“To visualize initiation of polymerization, cells were grown in the absence of the repressor (thiamine) for upto 28 – 30 hours, placed on agarose pads lacking thiamine and random fields with cells exhibiting diffuse fluorescence were imaged.” [page 11, lines 6 - 9]

3) Is the role of potassium important to mreB function. pg 7 line 10 notes the interesting observation but there is no discussion of the implications or importance.

A sentence related to this has been included in the discussion as follows:

“An interesting observation from our biochemical and structural characterization is the identification of a potassium ion at the interface of the nucleotide and the protein, which probably stabilizes the bound nucleotide conformation of ScMreB5.” [page 16, lines 1 -4]

Reviewer #3 (Comments to the Authors (Required)):

The reviewers have addressed all my concerns, and as long as they satisfy the other reviewers concerns (especially the important points raised of reviewer 1).

1. This paper nicely examines uses a mix of in vivo, in vitro, and structural studies to examine the effects of nucleotide hydrolysis on the assembly of MreB filaments, their lateral association, and disassembly. I do not see any prior work that conflicts with the findings in this paper.

2. The main points of the paper are all well evidenced:

a. The overall data showing that the E134 is a switch, sensing the ATP-bound state for filament assembly and stimulating assisting hydrolysis.

b. Likewise, their experiments indicating the nucleotide state and the E134 residue effects lateral associations and membrane binding also thorough.

c. Finally, their discovery of the effect the charge interactions between MreB and the membrane is a novel advance, and well done.

3. I have one small note that should be fixed, on line 18 they attribute the observation of chromosome segregation by MreB in Caulobacter to Dye, but the correct reference should be Z. Gitai 2005.

We thank the reviewer for complementing the work. The reference has been corrected now to the above, as suggested.